# Spatiotemporal establishment of dense bacterial colonies growing on hard agar

**Mya R Warren[1†], Hui Sun[1,2,3†], Yue Yan[2,4‡], Jonas Cremer[1§], Bo Li[2]\*, Terence Hwa[1]\***

[1]Department of Physics, University of California, San Diego, La Jolla, United States; [2]Department of Mathematics, University of California, San Diego, La Jolla, United States; [3]Department of Mathematics and Statistics, California State University, Long Beach, Long Beach, United States; [4]School of Mathematical Sciences, Fudan University, Shanghai, China

**Abstract** The physical interactions of growing bacterial cells with each other and with their surroundings significantly affect the structure and dynamics of biofilms. Here a 3D agent-based model is formulated to describe the establishment of simple bacterial colonies expanding by the physical force of their growth. With a single set of parameters, the model captures key dynamical features of colony growth by non-motile, non EPS-producing *E. coli* cells on hard agar. The model, supported by experiment on colony growth in different types and concentrations of nutrients, suggests that radial colony expansion is not limited by nutrients as commonly believed, but by mechanical forces. Nutrient penetration instead governs vertical colony growth, through thin layers of vertically oriented cells lifting up their ancestors from the bottom. Overall, the model provides a versatile platform to investigate the influences of metabolic and environmental factors on the growth and morphology of bacterial colonies.
DOI: https://doi.org/10.7554/eLife.41093.001

## Introduction

Bacteria often form dense biofilms with complex spatiotemporal structures (*Costerton et al., 1995*; *Nadell et al., 2016*; *O'Toole et al., 2000*; *Stoodley et al., 2002*). Mechanical and biochemical inter-actions, together with cell growth, motility, and signaling, are some of the common elements under-lying the rich variety of patterns and behaviors observed. Biofilms often play important roles in diverse settings ranging from environment to human health (*Costerton et al., 1999*; *Jayaraman and Wood, 2008*; *Potera, 1999*). But they are notoriously difficult to study experimentally because of their opaqueness, high heterogeneity and complex organization, involving multiple spatial and tem-poral scales (*Roberts et al., 2015*; *Stewart and Franklin, 2008*). In addition, biofilm-bound bacteria alter their micro-environment by secreting various polysaccharides, forming heterogeneous matrices of filaments that bind cells together within biofilms (*Branda et al., 2005*; *Flemming and Wingender, 2010*).

Over the years, various computational models have been constructed to capture different aspects of biofilm development (*Alpkvist et al., 2006*; *Espeso et al., 2015*; *Ginovart et al., 2002*; *Klapper and Dockery, 2002*; *Kreft et al., 2001*; *Kreft et al., 1998*; *Picioreanu et al., 2004*; *Seminara et al., 2012*; *Tierra et al., 2015*). However, most of these models are 'descriptive' in nature – the complexity of the biofilms makes it difficult to make quantitative comparison between experimental data and model predictions. In recent years, an increasing body of literature has been devoted to simpler, stripped down versions of the biofilm which can be more readily compared to experimental studies. The simplest among these is the growth of a simple bacterial colony on hard agar surface, with cells pushing against each other by the force of their own physical growth, without

**\*For correspondence:**
bli@math.ucsd.edu (BL);
hwa@ucsd.edu (TH)

[†]These authors contributed equally to this work

**Present address:** [‡]School of Mathematics, Shanghai University of Finance and Economics, Shanghai, China; [§]Groningen Biomolecular Sciences and Biotechnology Institute, University of Groningen, Groningen, Netherlands

**Competing interests:** The authors declare that no competing interests exist.

motility and without extracellular polysaccharides (*Boyer et al., 2011*; *Cole et al., 2015*; *Farrell et al., 2013*; *Ghosh et al., 2015*; *Grant et al., 2014*; *Jayathilake et al., 2017*; *Rudge et al., 2013*; *Rudge et al., 2012*; *Volfson et al., 2008*) In addition to serving as simpler models of biofilms, the growth of such colonies has been increasingly used in recent years as a model of microbial range expansion in studies of population genetics and ecology (*Hallatschek et al., 2007*; *Hallatschek and Nelson, 2010*; *Korolev et al., 2012*). Although the growth of such simple colonies has been investigated experimentally many decades ago (*Cooper et al., 1968*; *Lewis and Wimpenny, 1981*; *Mitchell and Wimpenny, 1997*; *Palumbo et al., 1971*; *Pirt, 1967*; *Reyrolle and Letellier, 1979*; *Wimpenny, 1979*), surprisingly, there has not yet been a common quantitative understanding of the basic elements controlling their growth, for example what factors determine the radial and vertical expansion speeds.

In this study, we develop a conceptually simple, yet physically realistic three-dimensional computational model, incorporating the elements of nutrient diffusion, cell-cell and cell-agar mechanical interactions, and introducing a unique cell-level model of surface tension. Our model is efficiently implemented with a parallel algorithm, enabling the simulation of a colony comprising a few million cells within 24 hr. The model is able to capture many observed features of the growing colonies, including the conic shape, the linear growth of the colony radius and height, and their dependence on the cell growth rate. Extensive analysis of the results reveals key driving forces underlying these observations, especially on the role of surface tension and the dynamic form of cell-agar friction, allowing us to make distinct predictions on how various biochemical and mechanical effects alter physiological features of the colony and generate macroscopic spatiotemporal patterns of the growing colony. To guide the construction of our model and validate our simulations, we conducted a series of experiments on the growth of colonies on agar using non-motile *E. coli*. A set of minimum media with various carbon sources was used to vary the cell growth rate.

## Results

### Experimental results

Experiments were performed using *E. coli* K12 strain EQ59, which is non-motile and harbors constitutive GFP expression; see 'Experimental Methods'. Each colony was inoculated as a single cell from batch culture growing in mid-log phase on 1.5% (w/v) agar with glucose minimal media, and incubated, covered, at 37°C for up to 1.5 days. The colony height profile was periodically monitored using a confocal microscope (see 'Experimental Methods'), and the result was highly repeatable; see *Figure 1—figure supplement 1*. Starting with a single cell, the colony remained a single layer through the first 13 hours (*Figure 1AB*), buckling into a second layer at around $t = 14$ h at a radius of ~75 $\mu$m (*Figure 1C–E and F*). It then developed into a 3D colony over time, maintaining an approximate conic shape through the ensuing 10-15 hours after buckling (*Figure 1G*). During this period which we refer to as the 'establishment phase', the colony radius increased linearly in time with a constant radial speed $V_R \approx 45.2$ $\mu$m/h and the colony height increased also linearly at a vertical speed $V_H \approx 12.4$ $\mu$m/h (*Figure 1H*), reaching a radius of ~500 $\mu$m and a height of ~150 $\mu$m by $t = 24$ h. As the colony grew further, the gain in height slowed down while radial expansion continued at the same speed (*Figure 1H* and *Figure 1—figure supplement 1*), leading to a significant flattening of colony morphology. In this study, we focus on the relatively simple establishment phase defined by $14 \leq t \leq 24$ h, where both the radial and vertical growth are linear.

We further probed the growth of colony using saturating amounts of different carbon sources, each supporting a different batch culture growth rate, spanning the range $0.5$ h$^{-1}$ to $1$ h$^{-1}$; see *Supplementary file 1*-Table S1. The radial and vertical expansion speeds obtained in the linear growth regime are plotted in *Figure 1I* against the batch culture growth rate in the respective medium. Our findings of vertical linear growth disagree with earlier finding by Pirt (*Pirt, 1967*) which was first questioned by Wimpenny (*Lewis and Wimpenny, 1981*; *Wimpenny, 1979*). However, the latter reported much larger radial expansion speeds than ours, suggesting that their study might be in a very different regime dominated by swarming motility (*Wu et al., 2011*).

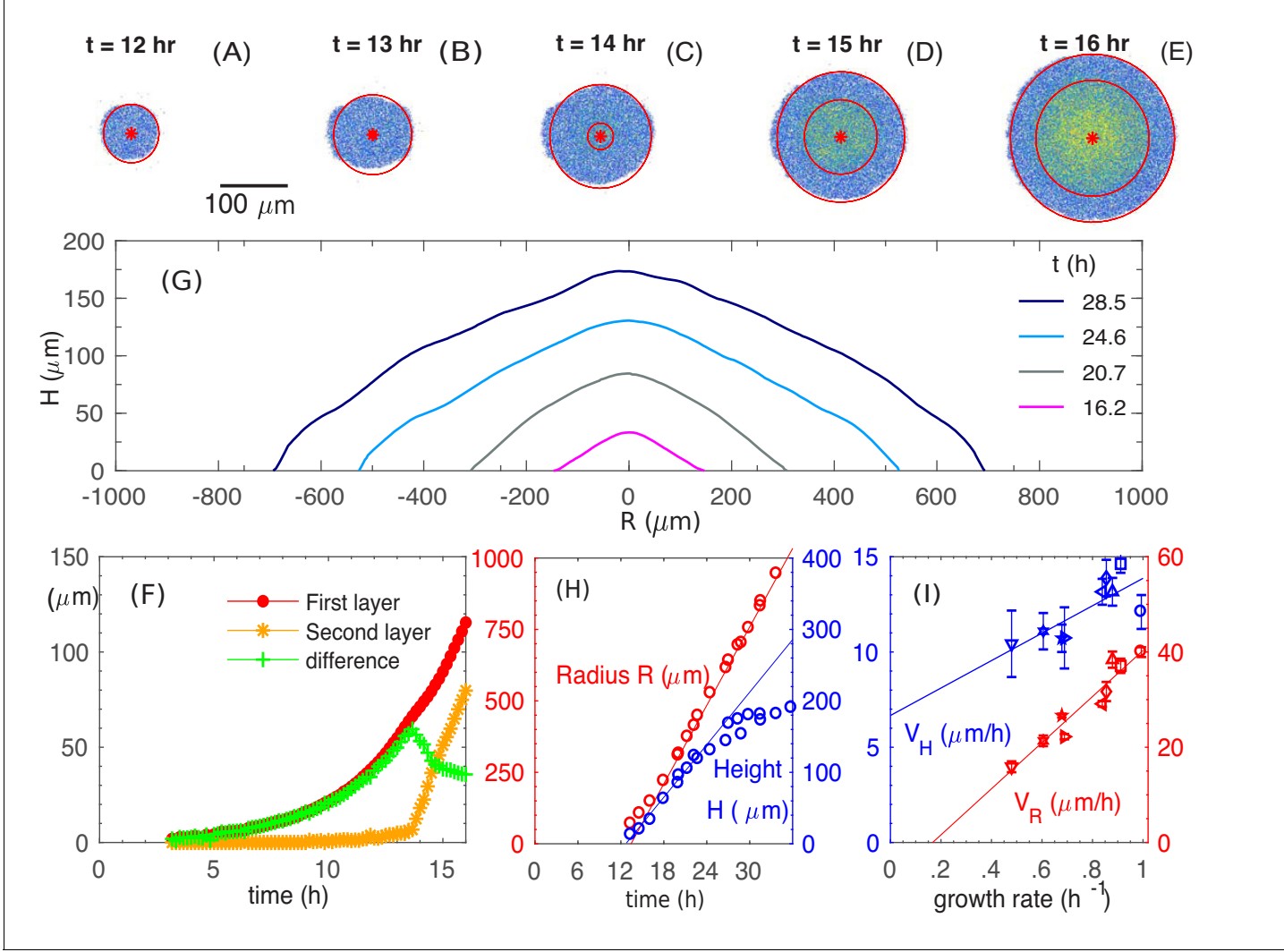

**Figure 1.** Experimental observations of the growth and morphology of a bacterial colony. (**A–E**) Confocal images of an *E. coli* colony harboring GFP expression growing on 1.5% agar (glucose minimal medium) taken at various time after seeding ($t = 0$). The center of the colony is indicated by the red dot. Single- and multi-layer regions are distinguished by red circles based on fluorescence intensity; see 'Experimental Methods'. (**F**) The radius of the first (red) and second layer (orange) of the colony, as well as their difference (green), versus time. (**G**) The cross-sectional profile of the growing bacterial colony at indicated time after single-cell inoculation. (**H**) After the buckling at around $t = 13$ h, the colony radius (red symbols) increased at a constant speed $V_R = 45.2$ $\mu$m/h (red line), while the colony height (blue symbols) increased linearly with speed $V_H = 12.4$ $\mu$m/h (blue line). The latter slowed down some time after $t = 24$ h. (**I**) The dependence of the radial speed $V_R$ (red symbols) and the vertical speed $V_H$ (blue symbols) on cell growth rate (x-axis), for colonies grown in minimal medium with 8 different carbon sources (***Supplementary file 1***-Table S1): glucose (O); arabinose (□); mannitol (△); maltose (◇); fructose (◁); melibiose (▷); sorbitol (▽); mannose (*I*). The lines are best linear fit of the data.

DOI: https://doi.org/10.7554/eLife.41093.002

The following source data and figure supplements are available for figure 1:

**Source data 1.** Experimental data for the temporal development of colony profiles and velocities.
DOI: https://doi.org/10.7554/eLife.41093.005

**Figure supplement 1.** Data for five repeats of *E.coli* EQ59 grown on 1.5% (w/v) agar in minimal medium with 0.2% glucose (11 mM), and incubated, covered, at 37°C for up to 3 days; cf. 'Experimental Methods'.
DOI: https://doi.org/10.7554/eLife.41093.003

**Figure supplement 1—source data 1.** Repetitions for the temporal development of colony height and radius.
DOI: https://doi.org/10.7554/eLife.41093.004

## Simulation results and analysis

To describe the morphology and dynamics of these growing colonies in the linear regime (the establishment phase), we focus on several main elements in the process: the supply of nutrient and interaction driven by the physical growth of cells. We construct a minimal, multiscale, three-dimensional model consisting of the diffusion of nutrient through the agar and the colony; the growth, division, and movement of individual cells; and the cell-cell, cell-agar, cell-surface mechanical interactions that generate forces driving cell movement; see *Figure 2*. A salient summary of the model is provided in Materials and methods. As will be described, a unique aspect of this model is the implementation of the surface tension, which enables us to capture bulk as well as single layer effects. We use the data from our experiments and literature to estimate the range of key parameters in the model, and implement our model using various numerical techniques. Details of the model and numerical methods are given in Appendix 1. Through the bulk of the study described below, a standard set of parameters were used (*Supplementary file 1*-Tables S2-S4); effects due to variation of parameter values are discussed towards the end.

### Radial and vertical growth of the colony

We start by examining how fast the colony expands radially and vertically. We run a simulation with the batch culture growth rate $\lambda_S = 1.0 \text{ h}^{-1}$, which corresponds approximately to the growth of *E. coli* in glucose minimal medium (*Supplementary file 1*-Table S1). We use a substrate concentration $C_s = 0.5 \text{ mM}$ here and will vary this parameter later. From *Figure 3A*, we see that the number of cells in a colony increases exponentially for approximately 10 hours before it slows down. From *Figure 3B*, we see that the cross-sectional profiles of the colony preserve their shapes and are evenly separated at equal time intervals for $t \geq 12 \text{ h}$, suggesting a constant expansion of the colony in the radial and vertical directions by $t = 12 \text{ h}$, similar to the experimental profiles in *Figure 1G*. (The spatial cell density inside the colony is constant, ~0.68 $\rho_{\text{cell}}$, throughout the interior of the colony; see *Figure 3—figure supplement 1*.) Detail of the profile at the colony periphery appears to be different. This is due to an approximate height assignment based simply on thresholding the fluorescence intensity to obtain the global height profile. This thresholding procedure does not capture height at

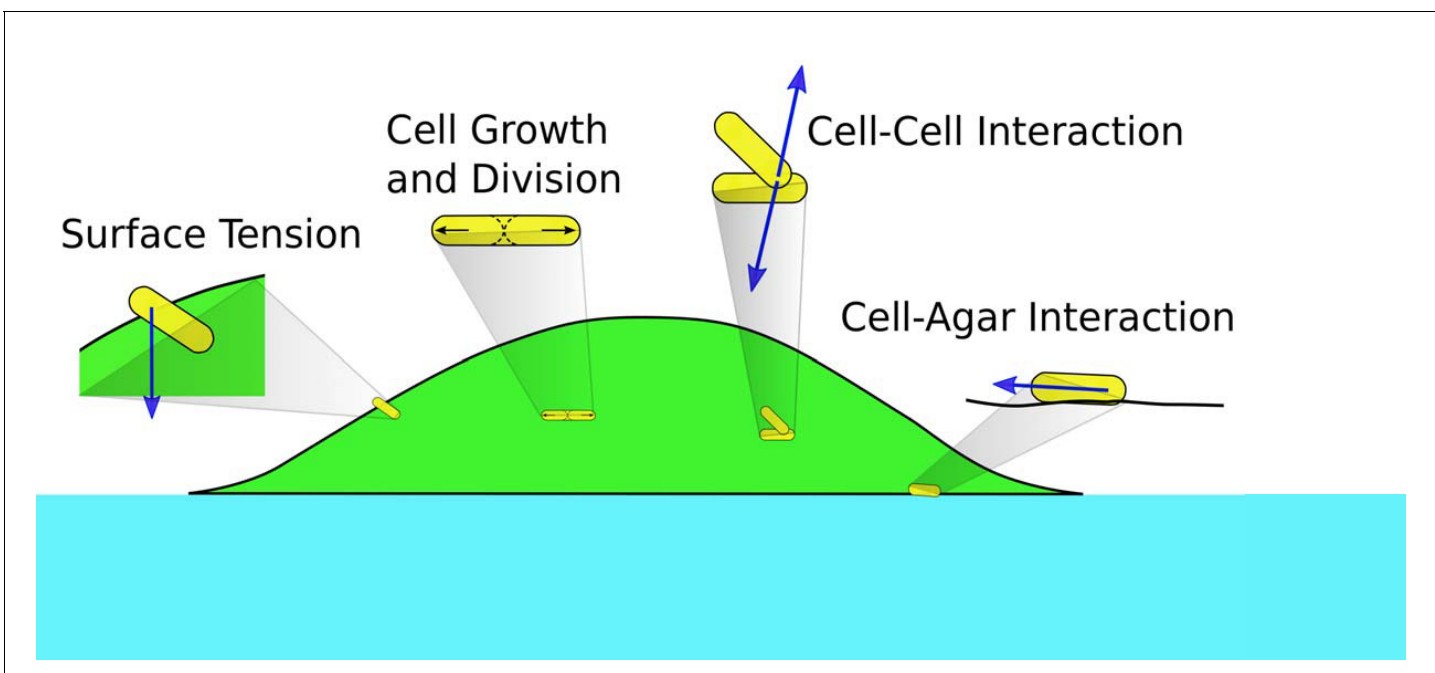

**Figure 2.** Schematics of cell-cell, cell-agar, cell-fluid, and surface tension forces investigated in this study. Green area indicates the colony (with cells in yellow). Blue area indicates the agar.
DOI: https://doi.org/10.7554/eLife.41093.006

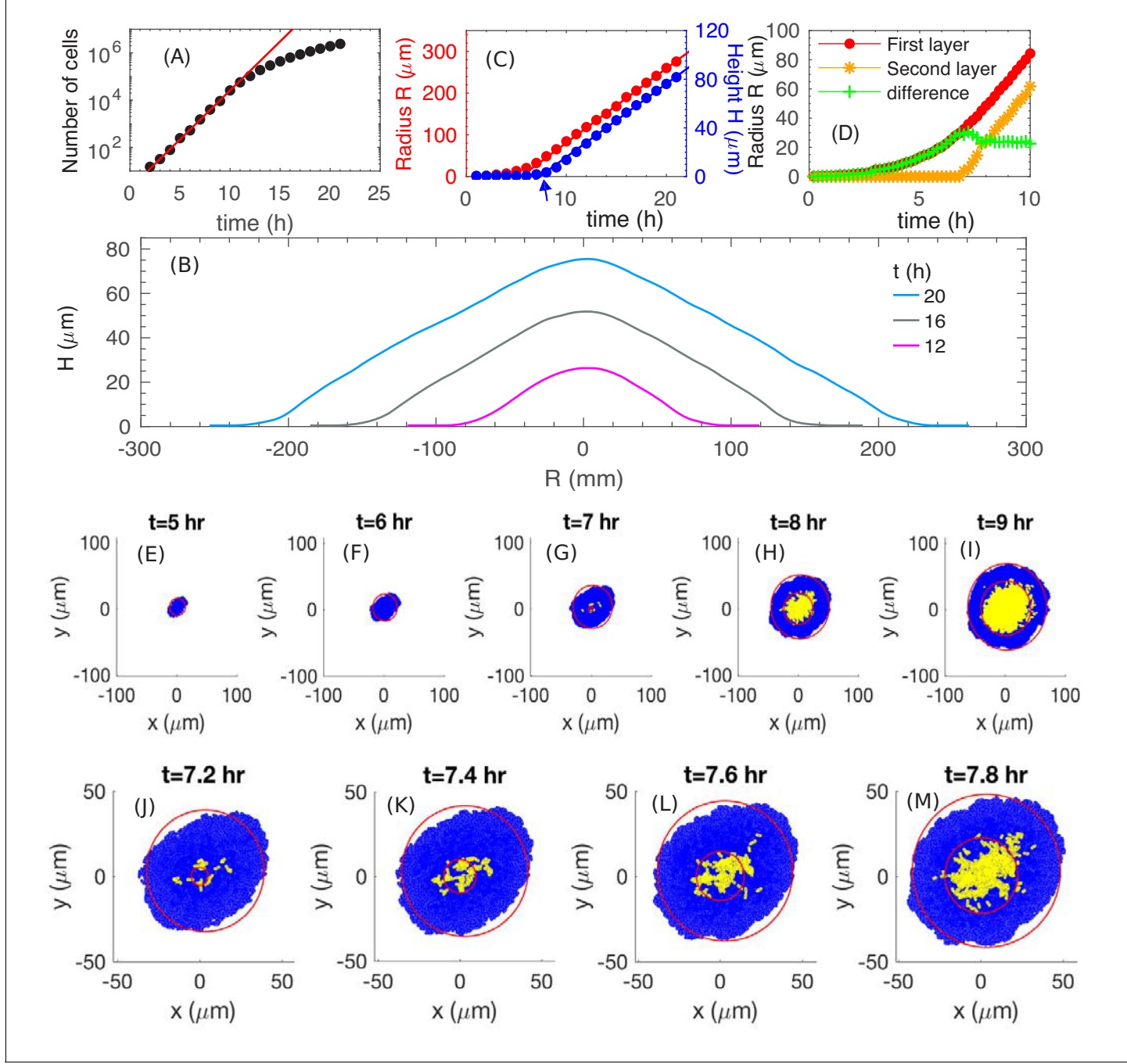

**Figure 3.** The simulated growth and morphology of a bacterial colony. (A) A semi-log plot of number of cells vs. time showing the exponential growth of the population starting from a single cell at $t = 0$. Red line shows exponential growth rate of $0.96 \text{ h}^{-1}$. (B) Cross sections of the growing colony at various times after seeding of a single cell. (C) Plots of the radius (red) and height (blue) vs. time up to $t = 20$ h, showing that, after an initial transient period of ~10 h, the growing colony increases linearly at the radial speed $V_R \approx 18 \ \mu\text{m/h}$ (red line) and vertical speed $V_H \approx 6.0 \ \mu\text{m/h}$. The blue arrow at $t = 6$ h indicates the time when the colony height starts to increase. (D) The radius of the first (red) and second (orange) layer of the colony as well as their difference (green) vs. time. (E–M) Top view of the colony at various time. Cells in the bottom layer (blue) and upper layers (yellow) are fitted into red circles. The time evolution of buckling phenomenon is captured in detail in (J–M).

DOI: https://doi.org/10.7554/eLife.41093.007

The following figure supplement is available for figure 3:

**Figure supplement 1.** The spatially varying cell density $\rho$ (per unit volume of colony) is related to the spatially varying cell volume fraction $\phi$ by $\rho = \phi \rho_{\text{cell}}$, where the volume fraction $\phi$ is defined as the volume of all cells in a unit volume of the colony and $\rho_{\text{cell}}$ is the constant mass density of a typical mature cell; cf. Appendix 1.2 on nutrient update.

*Figure 3 continued on next page*

*Figure 3 continued*

DOI: https://doi.org/10.7554/eLife.41093.008

the periphery where it is one to a few layers in thickness. *Figure 3C* provides a quantitative picture of the colony radius (*R*, defined as the average radius of the bottom layer of the colony) and colony height (*H*, defined as the height at the center of the colony). At early time, $t \leq 6\,\mathrm{h}$, the colony expands radially, while the height remains close to zero, indicating that the colony is comprised of a thin layer (see discussion in 'Radial expansion – quantitative analysis'). At around $t = 7\,\mathrm{h}$ (indicated by the blue arrow in *Figure 3C*), the height starts to increase, indicating the occurrence of 'buckling'. Details of this transition is shown in *Figure 3D and E–I*; they correspond well to the experimental patterns observed in *Figure 1F and A–E*. In particular, the model generates a constant width for the single-layer annulus region at the periphery, recapitulating report of a constant monolayer region by earlier mechanical study (*Su et al., 2012*). Moreover, the model captures the dynamical details around the buckling transition (compare *Figures 3D* and *1F*), which exhibits an initial fast increase of the annulus width resulting from the initial non-compact nature of the cells forming the second layer; see *Figure 3J–M*. After that point, both the colony radius and height increase linearly with time, with radial expansion speed $V_\mathrm{R} \approx 18\,\mu\mathrm{m/h}$ and vertical ascending speed $V_\mathrm{H} \approx 6\,\mu\mathrm{m/h}$; see *Figure 3C*. Thus, our model captures the linear increase of both the colony radius and height observed experimentally (*Figure 1H*). To understand the origin of these behaviors, we will analyze below the model output, first pictorially and then quantitatively. The lower numerical values of the speeds obtained from simulations are due to parameter settings chosen to limit computational time; this will be discussed in 'Parameter dependence'.

## Vertical rise – a pictorial view

We first focus on factors driving the linear vertical rise of the colony. We start with a pictorial view of the cell configuration and motion inside the colony. *Figure 4A* shows a snapshot of cell configuration in a vertical slice through the center of the colony, taken at time $t = 20\,\mathrm{h}$ which is well in the steady linear growth regime. The colors distinguish the gross orientations of the cells. The model shows that cells near the top surface are oriented parallel to the colony surface (shown in cyan), while cells away from the top surface are mostly oriented vertically (shown in yellow). A detailed view of the top surface of the colony generated from the simulation is shown in *Figure 4—figure supplement 1A*. This prediction is validated by confocal scan of the colony in experiment as shown in *Figure 4—figure supplement 1B*.

The model shows a thin region at the periphery of the colony in which all cells are oriented in plane. This region governs radial growth and will be discussed more in the next section. Away from the periphery into the colony interior, more and more cells stand up vertically. The azimuthally averaged angle from the agar surface is plotted against the radial position in *Figure 4B*. However, the internal verticalization took some time to develop (*Figure 4—figure supplement 2*); appreciable fraction of cells (50%) picked up vertical orientation only when the radius reached 250 μm.

To characterize the spatial variation in cell orientation more quantitatively, we coarse-grain the local director fields $\vec{n}\left(\vec{r}, t\right)$ (as described in Appendix A1.5) for the snapshot of *Figure 4A*. In *Figure 4C*, we plot the orientation of the azimuthally averaged director field, coarse-grained over boxes of size 4 μm × 4 μm over the *rz*-plane. We see that the orientation is vertical in the colony interior, but changes to be parallel to the colony surface in a transition zone of $\sim 50\,\mu\mathrm{m}$ into the surface along the radial direction.

Next, we examine the coarse-grained velocity field $\vec{v}\left(\vec{r}\right) = \left(v_x\left(\vec{r}\right),\ v_y\left(\vec{r}\right),\ v_z\left(\vec{r}\right)\right)$ whose azimuthal average is shown as arrows in *Figure 4D*. The velocity field points in the vertical direction throughout most of the colony, even at the top surface where cells are oriented parallel to the colony surface according to *Figure 4C*. Very close to the periphery in the bottom layer, the velocity field turns sideway; it is oriented planarly there and will be discussed below in the context of radial growth. As indicated by the length of the arrows, the vertically oriented velocity increases in magnitude away from the agar. This is illustrated by the plot of vertical velocity at different height z at the center of the colony, that is $V_z(z) = v_z(0, 0, z)$, in *Figure 4E*. We see that $V_z$ increases through a thin

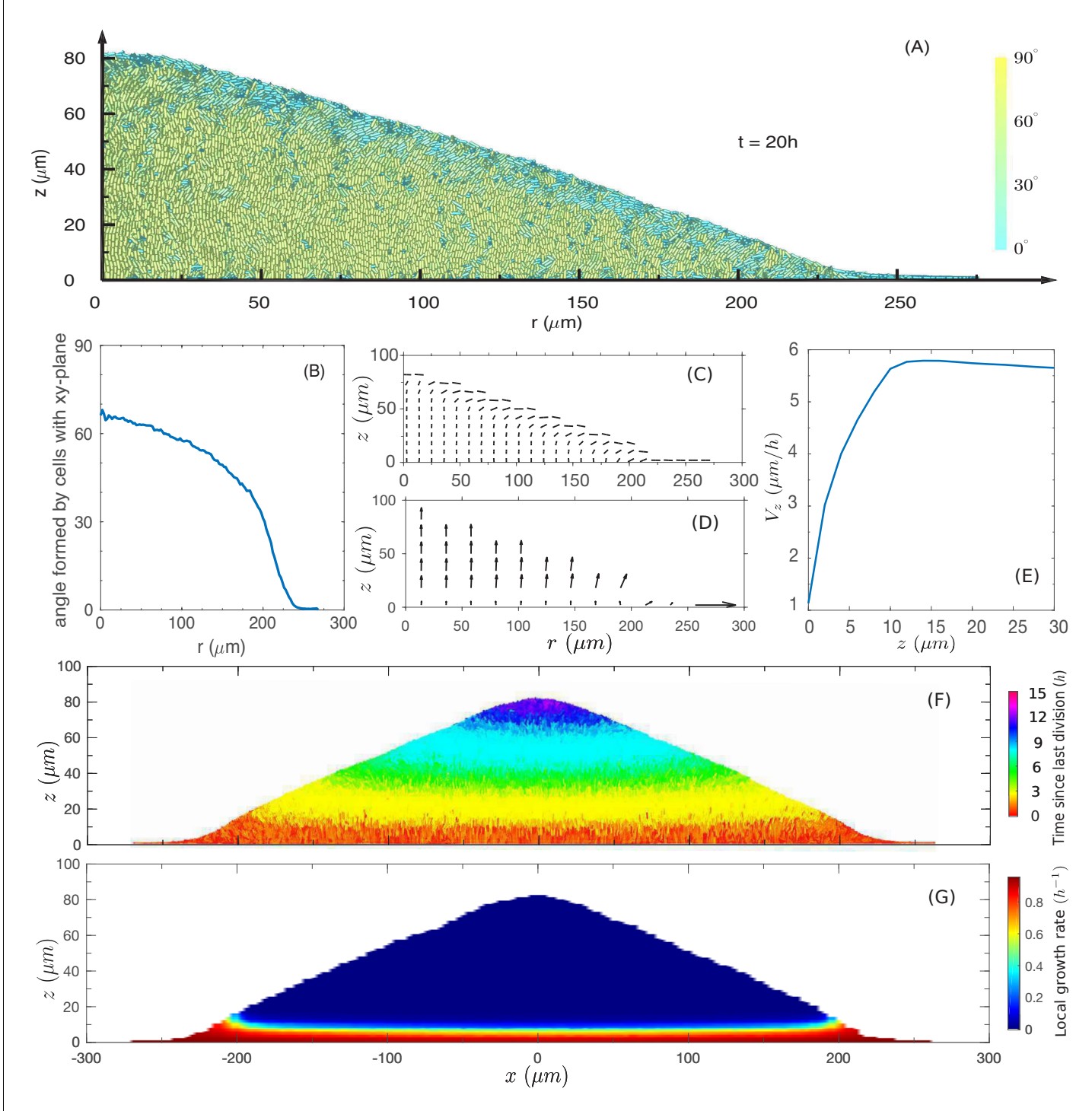

**Figure 4.** The cross-sectional anatomy of a simulated colony. (**A**) Snapshot of cross-sectional view of the colony at $t = 20$ h. Cyan represents horizontally oriented cells ($\geq 45^o$ with z-axis); Yellow represents vertically oriented cells ($\leq 45^o$ with z-axis). (**B**) Fraction of vertically oriented cells averaged over $z$ vs radius. (**C**) A side view of the azimuthally averaged director field, indicating the orientation of the rod-like cells. (**D**) A side view of the azimuthally averaged velocity field. (**E**) Vertical component of velocity, $V_z$, at various values of $z$ along the center of the colony. Increase in vertical speed is seen only for the bottom 10 μm (**F**) A cross-sectional view of the colony, color representing the time since last division. Purple and blue represent cells that have not divided for the past 10 h, and red represents the actively dividing cells. (**G**) A cross-sectional view of the local growth rate in the colony, with the color bar showing the values of local growth rate. A disc-shaped 'growth zone' is revealed by the red color at the bottom of the colony.

DOI: https://doi.org/10.7554/eLife.41093.009

*Figure 4 continued on next page*

*Figure 4 continued*

The following figure supplements are available for figure 4:

**Figure supplement 1.** A few top layers of cells in the colony visualized using simulation data.

DOI: https://doi.org/10.7554/eLife.41093.010

**Figure supplement 2.** Fraction of verticalized cells increase for large colonies.

DOI: https://doi.org/10.7554/eLife.41093.011

region of height $H_S \approx 10~\mu m$. The vertical ascension speed is saturated for $z > H_S$, meaning that above this thickness $H_S$, cells move up steadily.

Another way to visualize the vertical growth of the colony is to show the 'age' of cells in a cross-sectional view (*Figure 4F*). In this plot, the age of a cell is defined as the time duration since the last division of the cell, with red being the youngest and purple being the oldest. We see that cells at the bottom of the colony are all young (red), indicating that the bottom layer is constantly dividing. In contrast, the oldest cells (purple) occupy the top/center region of the colony, and the next oldest age groups (blue, green, etc.) are located in different layers below the purple top region.

Together, the above results suggest a simple picture of the vertical colony growth: The cells are oriented vertically (except those close to the surface) and are pushed up by growing cells located within a 10 μm thick growth zone at the bottom; they stop dividing once pushed out of the growth zone. This picture is verified in *Figure 4G*, where the cross-sectional plot of the local growth rate shows a clear growth zone of $\sim 10~\mu m$ (red region) confined to the bottom of the colony.

## Vertical rise – quantitative analysis

This disc-shaped growth zone at the bottom of the colony may be intuitive, since cells at the bottom of the colony are in direct contact with the agar and hence have the best access to the nutrients. A planar growth zone is in fact required to support the observed linear increase of colony radius and height (during the period $t$ = 12-24 hours in *Figure 1*): As the colony has the shape of a cone (*Figures 1G* and *3B*), its volume is given by $V_{\mathrm{colony}} \propto R^2 H \propto R^3$. Assuming that the increase of the colony size is due to a portion of cells growing at the maximal possible rate ($\lambda_S$) in a growth zone of volume $V_{\mathrm{growth}}(t)$, then $\frac{d}{dt} V_{\mathrm{colony}} \propto V_{\mathrm{growth}}(t)$ leads to $V_{\mathrm{growth}} \propto R^2$, that is a disc. The thickness of this growth zone is of interest because it controls the vertical ascension speed. As the local growth rate is merely a 'readout' of the nutrient concentration according to *Equation 3* in Materials and methods, we look into the penetration of nutrients into the colony, which determines the thickness of the growth zone. In *Figure 5A*, we plot the vertical nutrient concentration profile at the center of the colony, $C_{\mathrm{ctr}}(z) \equiv C(0,~0,~z)$, at various times $t$ during colony growth. In the linear growth regime (for $t \geq 12~h$), the profile $C_{\mathrm{ctr}}(z)$ is essentially stationary. As shown in *Figure 5—figure supplement 1* and Appendix A2.3, this stationary profile drops quadratically at small heights (i.e. close to the agar surface), and exponentially at larger heights (top of the colony), with the crossover between these two dependences occurring at the height scale $H_S$ such that $C_{\mathrm{ctr}}(H_S) = K_S$, the Monod constant appearing in *Equation 3*; see Appendix A2.3. Since the local growth rate drops substantially where the nutrient concentration is below $K_S$, we can take the value $H_S$ as the thickness of the vertical growth zone, leading to the vertical ascending speed: $V_H \propto H_S \lambda_S$.

Detailed analysis of *Equations 1 and 3* in Materials and methods shows that the scale of the stationary profile is set by $\sqrt{D_+/\lambda_S}$; cf. Appendix A2.3. This is verified in *Figure 5B* where the stationary profile $C_{\mathrm{ctr}}(z)$ is computed by repeating the simulation for different growth rate $\lambda_S$: the profile collapses into the same curve for different values of $\lambda_S$ when plotted against $z\sqrt{\lambda_S}$; see *Figure 5—figure supplement 2* for the same profiles without rescaling in z. Given this scaling, we expect the thickness of growth zone to decrease as $H_S \propto 1/\sqrt{\lambda_S}$ for faster growth (due to stronger nutrient depletion), leading to a sublinear dependence of the vertical ascending speed, $V_H \propto \sqrt{\lambda_S}$. Our numerical result on the growth of vertical height is shown as open blue symbols in *Figure 5C*. The results are well fitted by the square-root dependence on $\lambda_S$; see the solid line. In *Figure 1I*, we attempted to test the dependence of the vertical ascension speed on growth rate experimentally, by growing colony in different carbon sources supporting different growth rates. Unfortunately, the most distinguishing regime of the predicted square-root relation, for $\lambda_S < 0.4~h^{-1}$, is difficult to

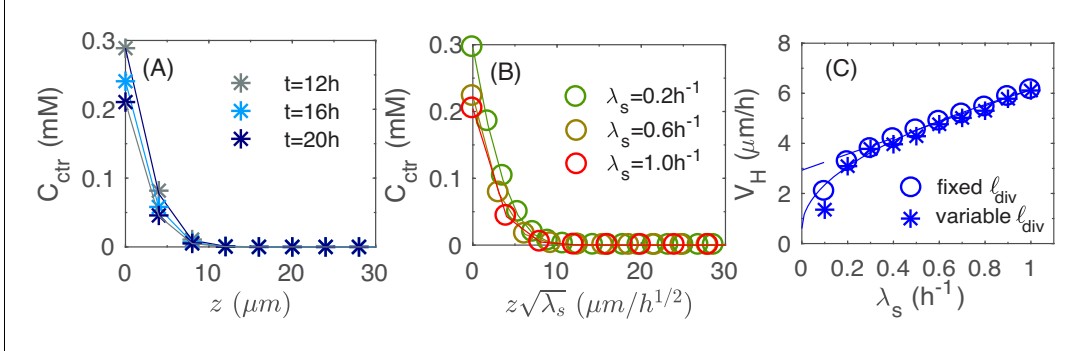

**Figure 5.** Vertical penetration of nutrients. (A) The profiles of nutrient concentration $C_{\mathrm{ctr}}(z) = C(0,0,z)$ along the z-axis at different times. (B) The profile $C_{\mathrm{ctr}}(z)$ in the uniform scale vs. that in the rescaled z-axis, $z\sqrt{\lambda_S}$. (C) The numerical results for the height velocity $V_H$ vs. the batch culture growth rate $\lambda_S$ with a fixed cell division length $l_{\mathrm{div}}$ (open circles) and variable $l_{\mathrm{div}}$ (asterisk), respectively. The square root fit for the open circles (solid line) is given by the expression $V_H = 5.5\sqrt{\lambda_S} + 0.6$; the linear fit for circles with $\lambda_S \geq 0.5\ \mathrm{h}^{-1}$ (dashed line) is given by the expression $V_H = 3.2\lambda_S + 2.9$.
DOI: https://doi.org/10.7554/eLife.41093.012
The following figure supplements are available for figure 5:

**Figure supplement 1.** Semi-log plot of the steady-state nutrient profile $C_{\mathrm{ctr}}(z) = C(0,0,z)$ : reconstructed from 3D simulations (green *); and the numerical solution to the 1D model (cf. Appendix 2.3 on nutrient penetration) with discretization $\Delta z = 4\ \mu m$ (green circles) and $\Delta z = 0.1\ \mu m$ (black line), respectively.
DOI: https://doi.org/10.7554/eLife.41093.013

**Figure supplement 2.** Profiles of nutrient concentration $C_{\mathrm{ctr}}(z) = C(0,0,z)$ versus $z$ for various values of the batch culture growth rate $\lambda_S$.
DOI: https://doi.org/10.7554/eLife.41093.014

realize by changing carbon sources. However, if we just fit the data in **Figure 5C** for $\lambda_S > 0.5\ \mathrm{h}^{-1}$, we obtain a weak linear dependence (dashed line) that resembles the experimental data in **Figure 1I** obtained. Note that the overall scale of the vertical ascending speed is 2-fold smaller in the simulation. This is attributed to the smaller nutrient concentrations used in the model compared to experiment, as will be discussed further below in the section of parameter dependence.

## Radial expansion – a pictorial view

We first study the case mimicking glucose medium, corresponding to the simulation result shown in **Figures 3** and **4**. Since cells at the bottom grow substantially (**Figure 4G**), we plot the cell configuration for the bottom layer of the colony at $t = 20\ \mathrm{h}$ in **Figure 6A**; the same color code as **Figure 4A** is used, with vertically oriented cells shown in yellow and horizontally oriented cells in cyan. The periphery is seen to be largely cyan while the interior is more yellowish, suggesting that cells at the interior of the bottom layer are already oriented vertically, consistent with the cross-sectional view shown in **Figure 4A**. We again coarse-grain the local director field $\vec{n}\left(\vec{r}, t\right)$ for the snapshot of **Figure 6A**. **Figure 6B** shows the planar projection of this director field in the bottom layer, where each bar indicates the average cellular orientation of cells in a region. We observe an annular region of $\sim 100\ \mu\mathrm{m}$ in width near the periphery, where the director field has a significant in-plane component, directed mostly along the radial direction, except at the outermost boundary, where the director field has a great azimuthal component. Towards the inner boundary of the annulus, the in-plane component becomes smaller in magnitude. Interior to the annulus, the in-plane projection of the director field vanishes, confirming that they are largely oriented vertically.

Next, we examine the coarse-grained velocity field $\vec{v}\left(\vec{r}, t\right)$ for the bottom layer of cells shown in **Figure 6A**, with the x-y projection of $\vec{v}\left(\vec{r}, t\right)$ shown as arrows in **Figure 6C**. We observe a narrow annulus of non-vanishing velocity field (arrows with finite length) at the outermost edge pointing radially outward; see also the side view provided in **Figure 4D**. Since the in-plane component of velocity vanishes inside the annulus (turning to vertical speed as shown already in **Figure 4D**), the driving force for radial colony expansion reside solely in the narrow annulus where cells are oriented

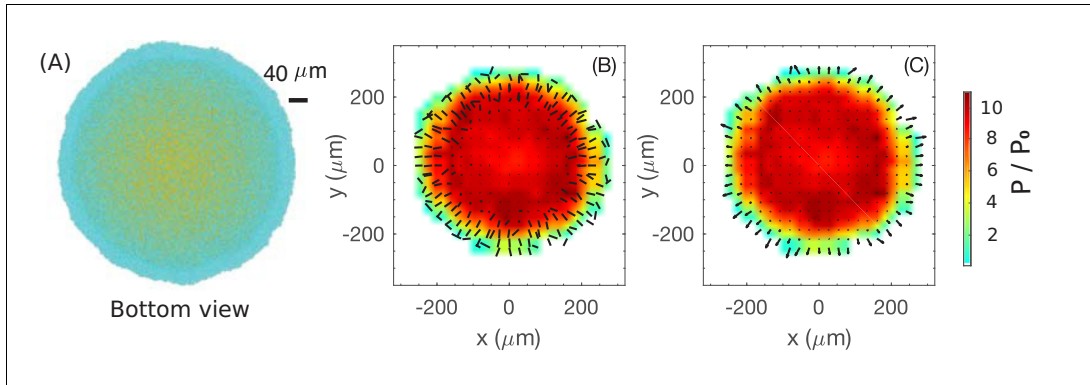

**Figure 6.** Coarse-grained view of director, velocity and pressure fields in the bottom layer of colony. (**A**) The bottom view of a simulated colony. Color scheme is the same as in **Figure 4A**. (**B**) Bars show the planar component of the coarse-grained director field at the bottom layer. (**C**) Arrows show the planar component of the coarse-grained velocity field at the bottom layer. Colors in (**B**) and (**C**) indicate the local pressure; see scale bar on the far right. The pressure is expressed in unit of $P_0 = \gamma_{\mathrm{surf}}/w_0$ where $\gamma_{\mathrm{surf}}$ is the surface tension, the main force underlying pressure build-up in our model.
DOI: https://doi.org/10.7554/eLife.41093.015

planarly (**Figure 6B**). Just as the thickness of the growth zone determines the vertical ascension speed, here the width of the annulus determines the colony's radial expansion speed.

So, what controls the annulus width? Or, equivalently, what determines the transition of velocity to the vertical orientation in the interior? Qualitatively, the difference between the peripheral and interior regions can be appreciated by looking at the coarse-grained pressure field $P(\vec{r}, t)$ experienced by the bottom layer, indicated by the color in **Figure 6BC**. This pressure is zero at the colony outer edge, and gradually builds up in the interior due to the physical growth of cells inside the closely packed colony. Where pressure is high, cells are oriented vertically and move vertically. This analysis thus suggests that increased pressure due to the physical growth of cells, which itself results from friction exerted by the substrate on the expanding cells, eventually forces cells to buckle and flow upward, manifested by the reorientation of cell directors in the vertical direction. Once the flow turns upward, pressure does not build up further due to the lack of friction with the agar surface. Since the upward flow is resisted by the surface tension, we conclude that pressure maxes out in this case at a level that is mostly determined by the surface tension. Below, we investigate quantitatively this buckling phenomenon.

### Radial expansion – quantitative analysis

First, we examine the nutrient profile at the colony agar interface for growth on glucose. As can be seen from **Figure 7A**, the nutrient concentration is reduced underneath the colony. However, the actual concentration (**Figure 7BC**) is still much larger than $K_S$ of glucose uptake (dashed line in **Figure 7BC**), so that cells at the bottom do not experience nutrient depletion. In fact, at the colony periphery, nutrient concentration is close to the bulk level (**Figure 7D**).

To elucidate the determinants of buckling, we plot in **Figure 8A** the azimuthal-averaged radial velocity profile $V_r(\Delta r)$ for the bottom layer of cells, for a range of (signed) distances $\Delta r$ into the edge of the colony; see Appendix 1 **Equations (A1.5.1 and A1.5.2)** for the definitions of $V_r$ and $\Delta r$. This radial velocity profile, which is stationary for $t \geq 12$ h, is nearly zero in the colony interior, but increases almost linearly within a $\sim 20$ $\mu$m region at the outermost periphery of the colony. Since the radial expansion speed of the colony $V_R$ is simply $V_r$ at $\Delta r = 0$, we see that the width of this annulus together with the slope of $V_r(\Delta r)$ set the radial expansion speed of the entire colony.

To understand what goes on in this peripheral region, we examine in **Figure 8B** the azimuthal-averaged height profile of the colony, $h(\Delta r)$, which is also stationary for $t \geq 12$ h, with $h(\Delta r) \approx 1$ $\mu$m in the $\sim 20$ $\mu$m periphery region. This indicates that this periphery region is comprised of a single layer of cells lying horizontally on agar. In this monolayer region, the increase of $V_r$ can be understood

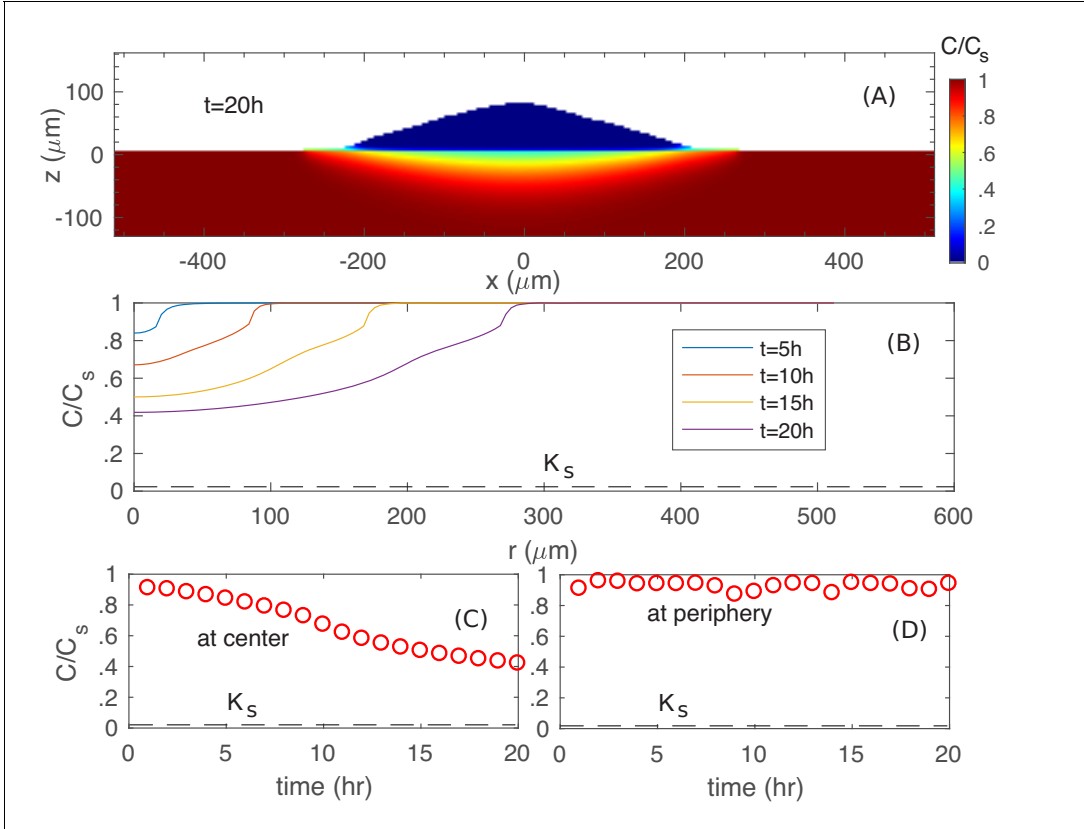

**Figure 7.** Spatiotemporal nutrient profiles. (**A**) The xz cross sectional view of the nutrient concentration inside the colony and in the agar, at time $t = 20\ h$. (**B**) The nutrient profile at the agar surface for different times. The nutrient concentration vs. time at the center (**C**) and at the periphery (**D**) of the colony at the agar surface.

DOI: https://doi.org/10.7554/eLife.41093.016

analytically, as we explain now. By mass conservation, the rate of local cell volume increase is balanced by the divergence of the velocity field, that is $\vec{\nabla} \cdot \vec{V} = \lambda$, where $\lambda$ is the local mass growth rate (**Klapper and Dockery, 2002**). Through most of the monolayer region (except close to the inner edge), the vertical velocity $V_z$ is negligible (**Figure 8C**). Hence, $V_r$ satisfies

$$\frac{1}{r}(rV_r)' = \lambda.$$

Throughout the periphery region, the growth rate is essentially the maximal growth rate, that is $\lambda \approx \lambda_S$, since the nutrient concentration in this region is set by the boundary value $C_s$ which well exceeds the Monod constant $K_S$; cf. **Figure 8—figure supplement 1**. Solving the above equation in the annulus in the limit $|\Delta r| \ll R$ yields a linear dependence,

$$V_r(\Delta r) \approx V_R + \lambda_S \cdot \Delta r$$

where we used the definition of the radial expansion speed $V_R = V_r(\Delta r = 0)$. This solution is indicated by the red line in **Figure 8A**, which is in agreement with the numerical data, with a small discrepancy for small $V_r$ attributed to the neglected vertical velocity at the inner periphery.

Given the linear radial velocity profile (cf. the previous equation) in the peripheral monolayer region, the width of this region $W_b$, defined as the largest value of $|\Delta r|$ where $V_r(\Delta r) = 0$, sets the radial expansion speed since $V_R \propto \lambda_S \cdot W_b$. We call this width the 'buckling width' since in the outer most ring region of the colony of size being this buckling width, cells form a monolayer, expanding with the speed $V_R$, while the interior cells that are away from the colony edge by this buckling width flow up vertically; cf. **Figure 8—figure supplement 2**. The magnitude of the buckling width is set by

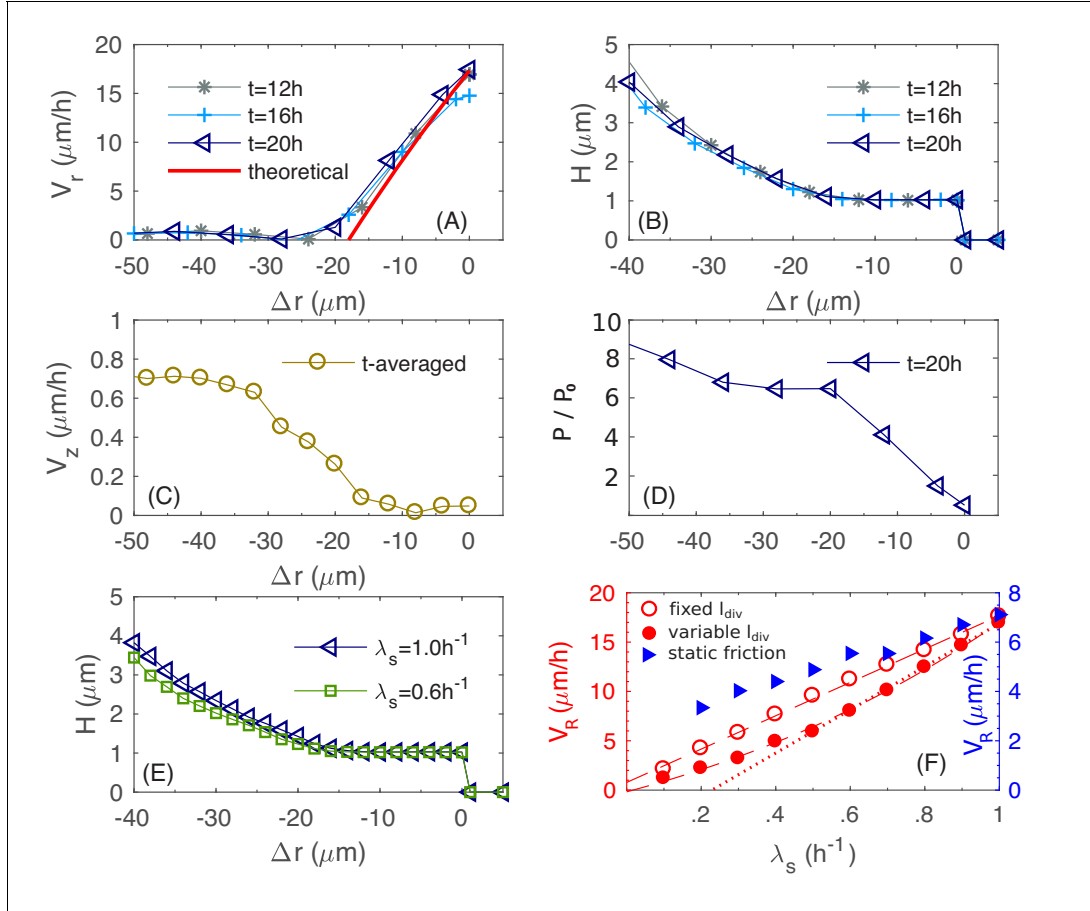

**Figure 8.** Physical characteristics near the outer periphery of the colony. (A) The azimuthally averaged local speed of radial expansion $V_r$ vs. the signed distance $\Delta r$ from the edge of the colony (cf. *Equations (A1.5.1)* in Appendix 1) at various times. (B) The azimuthally averaged local height $H$ vs. $\Delta r$ at various times. (C) The azimuthally averaged local speed of vertical expansion $V_z$ vs. $\Delta r$, averaging over $t = 12,\ 16,\ 20$ hr. (D) The azimuthally averaged local pressure $P$ vs. $\Delta r$ at time $t = 20$ h. As in *Figure 6*, pressure is expressed in unit of $P_0 = \gamma_{\text{surf}}/w_0$. (E) The azimuthally averaged height $H$ vs. $\Delta r$ at growth rate $\lambda_S = 0.5\ \text{h}^{-1}$ and $1.0\ \text{h}^{-1}$. (F) The simulated colony horizontal expansion speed $V_R$ vs. the batch culture growth rate $\lambda_S$ with a fixed $l_{\text{div}}$ (red open circles) and a growth-rate dependent $l_{\text{div}}$ (red closed circles) using dynamic friction. The dashed line that fits the open circles is given by $V_R = 16.9\lambda_S + 0.8$; the solid line that fits the closed circles for $\lambda_S \geq 0.5\ \text{h}^{-1}$ is $V_R = 5.7\lambda_S l_{\text{div}} - 0.2$; the dash dotted line fits the closed circles with $V_R = 22.1\lambda_S - 5.2$. In these expressions, the speeds are in unit of $\mu\text{m/h}$ and growth rate in $\text{h}^{-1}$. For comparison, we also include simulated $V_R$ vs. $\lambda_S$ with a growth-rate dependent $l_{\text{div}}$, for a model with static friction alone (see *Equations (A1.4.6)* of Appendix 1) between cell and agar (blue triangles).

DOI: https://doi.org/10.7554/eLife.41093.017

The following figure supplements are available for figure 8:

**Figure supplement 1.** The azimuthally averaged and rescaled local growth rate $\lambda/\lambda_S$ as function of the signed distance $\Delta r$ to the colony rim with various values of the batch culture growth rate $\lambda_S$.

DOI: https://doi.org/10.7554/eLife.41093.018

**Figure supplement 2.** A zoomed in view of the periphery of the colony shown in *Figure 4A*, overlaid with coarse-grained velocity field (zoomed in view of the same periphery region in *Figure 4D*).

DOI: https://doi.org/10.7554/eLife.41093.019

the radial forces exerted on the monolayer of cells. As these cells grow outward, they experience frictions from the agar substrate as well as surface tension that holds them down flat. These two forces lead to the accumulation of pressure, which is built up from the periphery. *Figure 8D* shows the azimuthally averaged pressure $P(\Delta r)$ for the bottom layer of cells. At the inner edge of the monolayer region, pressure reaches a critical value that surface tension can no longer hold cells in a single layer. There, some cells buckle into the vertical dimension, leading to vertical flow of cells and forming multiple layers (*Figure 8—figure supplement 2*), alleviating the further build-up of

pressure. It is interesting to observe that this buckling phenomenon is already evident early on during transition from monolayer growth to multiple layers, as shown in *Figure 3D*. The 20 μm annulus of monolayer at the periphery is set soon after the initial buckling transition, at around $t = 8$ h (*Figure 3D*), setting the pace of radial expansion.

Quantitative details of the buckling transition depend on the form of the cell-agar friction. Two types of friction have been used in the cell-modeling literature, one which depends linearly on the cell-agar velocity, known as viscous or static friction (*Farrell et al., 2013*; *Ghosh et al., 2015*), and the other which saturates to a constant set by the magnitude of the normal force (in this case, resulting from the surface tension). The latter is referred to as dynamic friction; see Appendix 1.4. The two forms can be distinguished by comparing the buckling width $W_b$ at different radial expansion speed $V_R$: Static friction would increase for increased $V_R$, leading to decreased buckling width, whereas dynamic friction would not be affected by the radial expansion speed. Experimentally, we characterized $V_R$ in sugars supporting different growth rates $\lambda_S$, and $V_R$ is seen to increase linearly with $\lambda_S$ (*Figure 1I*), suggesting a constant $W_b$, and hence the applicability of dynamic friction. This dependence is tested by running simulations with the dynamic friction form (*Equations 7a and 7b* in 'Computational Model') for different growth rate $\lambda_S$. The buckling width $W_b$ is indeed not dependent on $\lambda_S$ (*Figure 8E*), and the radial expansion speed $V_R$ is indeed linear in $\lambda_S$ (open red symbols and dashed red line, *Figure 8F*). In contract, static friction leads to a much weaker dependence of $V_R$ on $\lambda_S$ (blue triangles in *Figure 8F*).

The linear dependence on $\lambda_S$ seen in the experimental data in *Figure 1I* (red symbols) however exhibits a noticeable horizontal offset. This offset likely results from an additional effect we have not included into the model so far: The size of the cells is dependent on their growth rate, with faster growth rate being longer and wider (*Jun and Taheri-Araghi, 2015*; *Nanninga and Woldringh, 1985*; *Taheri-Araghi et al., 2015*). By repeating the established dependence of cell size on growth rate (see *Equation (A2.2.3)* in Appendix 2) for different values of $\lambda_S$, we recover a nonlinear dependence of $V_R$ on $\lambda_S$ (*Figure 8F*, filled red circles and solid red line). Note that a similar horizontal offset is obtained as the experimental data in *Figure 1I* if we do a linear fit using the data with $\lambda_S > 0.5$ h$^{-1}$ (dotted red line). On the other hand, the growth-rate dependence of cell sizes has no noticeable effect on the vertical ascension speed (filled blue symbols, *Figure 5C*) since the growth zone thickness $H_S \propto 1/\sqrt{\lambda_S}$ does not depend on $l_{\mathrm{div}}$ (Appendix 2.3).

## Parameter dependance

The preceding analysis shows that the vertical expansion speed of the colony depends on the thickness of the vertical growth zone which is set by the nutrient penetration depth, while the radial expanding speed depends on the width of the monolayer annulus which is set by the onset of the buckling transition but not the nutrient. The sizes of these growth zones are therefore dependent on the magnitudes of the physical parameters in different ways: We expect changing the cell-agar friction to affect the onset of the buckling transition and hence the radial expansion speed $V_R$, but not the vertical ascension speed $V_H$. Conversely, we expect changing the nutrient concentration $C_s$ to change the vertical nutrient penetration length and hence $V_H$, but not $V_R$. These expectations are indeed reproduced by the full colony simulation using different parameter values from the ones discussed so far, with the nutrient concentraiton $C_s$ doubled in *Figure 9A* (only $V_H$ increased), and with the cell-agar friction quartered in *Figure 9B* (only $V_R$ increased). These predictions are further tested experimentally, by varying the glucose concentraton used in the agar (*Figure 9C*), and by repeating experiments in reduced agar densities (*Figure 9D*) which we expect to reduce the cell-agar friction. The observed changes are very much in line with the expectations of the model shown in *Figure 9AB*. These results serve to validate the very important qualitative results of our study, that radial grow of the colony is not limited by nutrients while the vertial growth is limited by nutrients.

We note that the actual values of radial and vertical expansion speeds obtained ($V_R = 17.2$ $\mu$m/h and $V_H = 5.8$ $\mu$m/h), for the standard parameter set used (*Supplementary file 1*-Tables S2–S4) through the bulk of the study, are approximately 2x lower than the range of values obtained in experiments. The results of *Figure 9AB* show that the experimental range could be obtained simply by adjusting the combinations of parameters. We did not do that – the parameter set giving smaller $V_R$ and $V_H$ was chosen due to computational constraints: Higher nutrient concentrations requires longer computational time to reach the linear steady state due to the larger

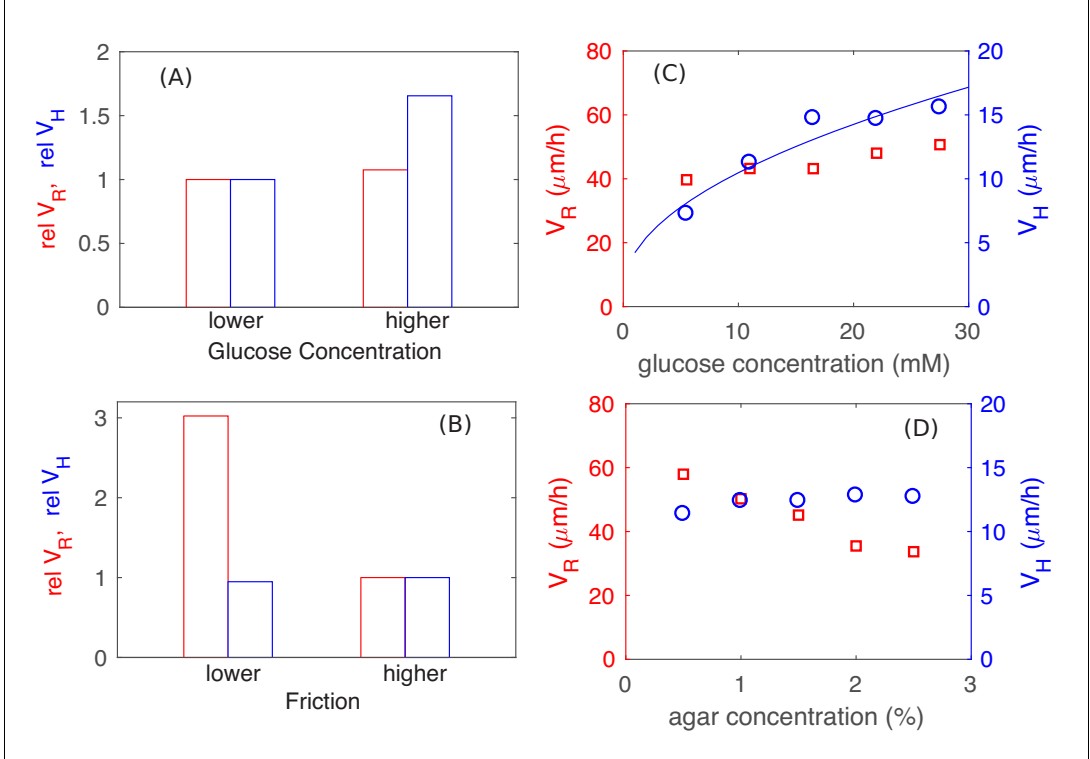

**Figure 9.** Parameter dependence of colony growth characteristics. Simulation results using the full model with 2x increase in glucose concentration (panel **A**) and 4x decrease in all frictional parameters (panel **B**) for $V_R$ (red bars) and $V_H$ (blue bars). Specifically, in panel (**A**) we fix the friction at a high level (with $\mu_{ca} = 0.8$, $\mu_{cc} = 0.1$, $\gamma_{cc,t} = 10000\ \mu m^{-1}h^{-1}$, and $h_{ran} = 0.1\ \mu m$), and use $C_s = 0.5$ mM as the lower glucose concentration, $C_s = 1.0$ mM as the higher glucose concentration. In panel (**B**), we fix the glucose concentration at the lower level ($C_s = 0.5$ mM), and vary the friction, from the higher value of $\mu_{ca} = 0.8$, $\mu_{cc} = 0.1$, $\gamma_{cc,t} = 10000\ \mu m^{-1}h^{-1}$, $h_{ran} = 0.1\ \mu m$ used in (**A**) to the lower value of $\mu_{ca} = 0.2$, $\mu_{cc} = 0.025$, $\gamma_{cc,t} = 2500\ \mu m^{-1}h^{-1}$, $h_{ran} = 0.025\ \mu m$. The corresponding experimental results are shown in panels C and D: In (**C**), glucose concentration was varied with agar density fixed at 1.5%. In (**D**), agar density was varied with glucose fixed at 0.2% (w/v). The data for $V_H$ in panel C is consistent with a square root dependence on nutrient concentration (blue line) expected from the basic analysis in *Figure 5*.

DOI: https://doi.org/10.7554/eLife.41093.020

The following source data is available for figure 9:

**Source data 1.** Experimental data on the horizontal and vertical colony expansion speeds at various glucose and agar concentrations.

DOI: https://doi.org/10.7554/eLife.41093.021

nutrient penetration depth. Similarly, lower cell-agar friction would lead to colony spreading too rapidly in the radial direction, thus requiring larger simulation sizes and hence again longer computational time. Their combination becomes difficult to investigate at the level of details done in this study. The particular values of frictional coefficients in the standard parameter set have been chosen so that the colony retains similar aspect ratio as observed in experiments, but with both $V_R$ and $V_H$ being about half of the experimentally observed values for growth on glucose medium. As computing power continues to increase, these models should soon be able to reach sizes comparable to realistic colonies with realistic parameters.

## Discussion

In this work, we presented a detailed quantitative study of the growth of a bacterial colony on hard agar surface starting from a single cell. For non-motile bacteria incapable of producing extracellular polysaccharides, the colony is driven primarily by the force of their own growth. Key factors involved are nutrient diffusion, mechanical interactions between cells, friction between cell and agar, and the surface tension holding the cells to the agar. We developed a continuum model for nutrient diffusion and implemented it with a multi-resolution numerical technique. With a discrete agent-based model,

we captured mechanical interactions, including elasticity and dynamic friction. Most importantly, the surface tension of the liquid in the colony is implemented by introducing a restoring force on cells protruding from a smoothened colony surface.

Our model is able to capture quantitatively some of the characteristic features observed for bacterial colony growth, including the conic shape of the colony, the linear expansion of colony radius and height, and both the linear and sublinear dependence of the speed of radial expansion and that of vertical expansion, respectively, on the cell growth rate. The model makes a number of important predictions on the expanding colony as summarized in *Figure 10*: The growth zone is predicted to be disc-like and extended throughout the bottom of the colony, contrary to common belief (see below). Radial growth is driven by cells at the outer perimeter of the growth zone; these cells are predicted to form a thin layer, oriented parallel to the agar due to the downward pull of surface tension, with the width of the region determined by the onset of the buckling transition (which occurs when radial compression due to cell-agar friction overwhelms the surface tension). In the colony interior, cells are predicted to orient vertically and are mainly pushed upward by elongating cells in the bottom growth zone.

Capturing all these behaviors within a single model and with a fixed set of parameters is a non-trivial task despite the seeming simplicity of this problem. Many aspects of our model are taken from what are commonly adopted in the extensive literature devoted to this class of problems over the past decade (*Boyer et al., 2011*; *Cole et al., 2015*; *Farrell et al., 2013*; *Ghosh et al., 2015*; *Grant et al., 2014*; *Jayathilake et al., 2017*; *Rudge et al., 2013*; *Rudge et al., 2012*; *Volfson et al., 2008*). These include the basic modeling of metabolism and cell growth (*Cole et al., 2015*; *Farrell et al., 2013*; *Rudge et al., 2012*), and the use of Hertzian elasticity to describe cell-cell elastic interaction (*Boyer et al., 2011*; *Farrell et al., 2013*; *Ghosh et al., 2015*; *Grant et al., 2014*; *Volfson et al., 2008*), all incorporated as computational power increases to reach ever increasing colony sizes (*Cole et al., 2015*; *Rudge et al., 2013*; *Rudge et al., 2012*). Unique to our study is the treatment of mechanical interactions, specifically friction and cell-level surface tension, which we believe are at the root of all behaviors described above, including the forms of radial and vertical colony growth. A key result of our study is that the linear radial growth is driven by the growth of a thin layer of radially oriented cells located at the colony periphery, whose width is determined by mechanical buckling. Although the linear radial expansion of bacterial colonies has been

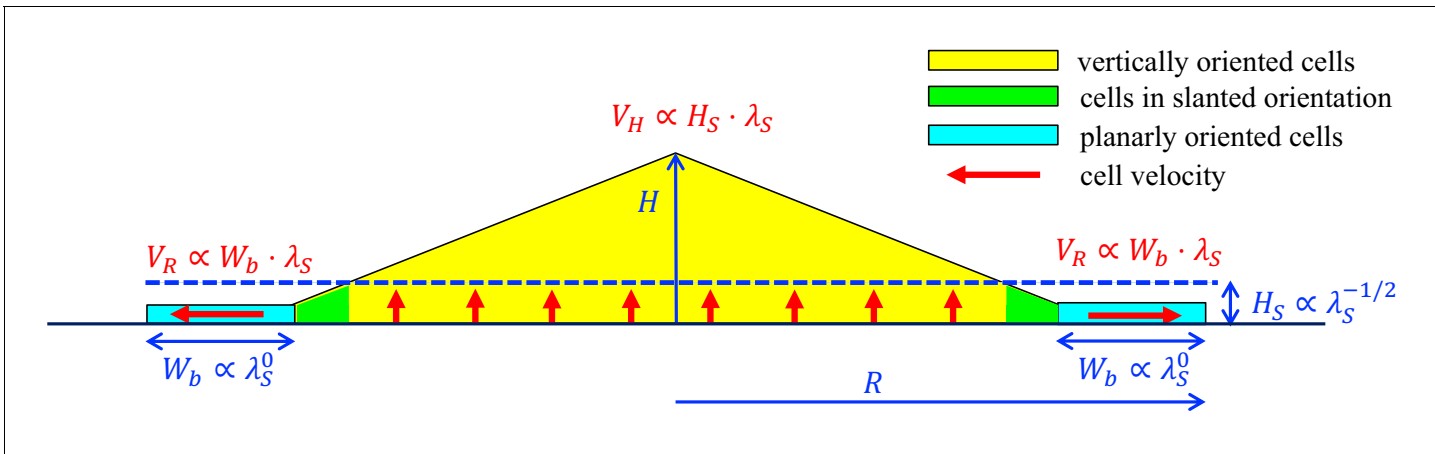

**Figure 10.** A schematic summary of key mechanisms in the growth of an *E.coli* colony. After an initial, exponential monolayer growth, buckling occurs at the center of the colony. Cells then grow actively only in the bottom layers (red vertical arrows) whose thickness ($H_S$) is determined by the nutrient penetration level (dashed blue line). Cells lying above them are passively pushed up. Throughout this yellow triangular region, cells are oriented vertically. Near the colony edge (cyan region), the cells are oriented planarly and grow outward (horizontal red arrow) continuously in a spread mode to expand the colony in the radial direction. The width of this annulus ($W_b$) is determined by mechanical effects arising from the surface tension which pulls the thin layer of cells into the agar, and cell-agar friction which builds up the pressure from the outer edge of the layer, eventually causing buckling at an inner radius where cells transition to the vertical orientation (the green region). These two characteristic parameters, $H_S$ and $W_b$, set the speeds of radial and vertical expansions, $V_R$ and $V_H$, respectively, as shown in red. The growth rate dependence of these parameters is shown in blue.
DOI: https://doi.org/10.7554/eLife.41093.022

known for about 50 years (*Pirt, 1967*), for a long time this was attributed to a ring-shaped growth zone at the outer colony periphery due to nutrient diffusion (*Lewis and Wimpenny, 1981*; *Pirt, 1967*; *Wimpenny, 1979*). Only quite recently has the notion been made that mechanical effects might also lead to linear radial growth (*Farrell et al., 2013*; *Su et al., 2012*). (*Su et al., 2012*) showed experimental results that implicated the interplay of forces in the radial expansion of colonies. (*Farrell et al., 2013*) proposed mechanical effects as a colloquial rationalization of numerical results generated by toy models with unrealistic details, for example a 'gravity-like' adhesion force acting on all cells in the colony. In our study, the adhesion of cells to the agar surface is provided by the surface tension of the liquid surrounding cells in the colony. We introduce a novel cell-level model of surface tension which acts only on cells at the colony surface, distinct from common models of surface tension which depends on the macroscopic curvature of the colony surface and cannot describe thin layers. It is this unique surface tension model that enables us to capture the dynamics from the initial single-layer cell growth, through buckling, to the growth of a macroscopic colony. This cell-level surface tension, responsible for pressing cells into the agar thereby generating friction that eventually causes buckling, cell reorientation and vertical colony growth, is thus the source of all mechanical interactions in the colony. A strong, uniform force such as the ones used in (*Farrell et al., 2013*) would lead to artificially flattened colonies, especially at the colony center where the height is the highest, since the force is proportional to the height in that model.

We regard the characterization of colony growth for different nutrients (which give rise to different cell growth rates) as a unique contribution by our study. The knowledge of the dependence of colony growth on cell growth allows us to discriminate different models of colony growth. As an example, an important component of our model that makes a quantitative difference to the outcome is the form of the friction used. Viscous drag (i.e., friction proportional to the velocity difference) is the form adopted in most models of cell dynamics (*Farrell et al., 2013*; *Ghosh et al., 2015*; *Rudge et al., 2012*). We instead adopt a form commonly used in modeling granular solids (*Brilliantov et al., 1996*; *Cundall and Strack, 1979*; *Kuwabara and Kono, 1987*; *Shäfer et al., 1996*). It involves a static friction depending on relative velocity, capped by a *dynamic friction* which is independent of the velocity. This form, introduced in one of the first models of 2D colony growth (*Volfson et al., 2008*), exerts a pressure which is independent of the speed of radial expansion, leading to a growth rate-independent buckling width and hence a radial expansion speed that is proportional to cell growth rate, in agreement with our experiments. In contrast, a model based on static friction would have the buckling width reducing with increasing cell growth rate, giving a sublinear dependence of radial expansion speed on cell growth rate which is not compatible with the data in *Figure 1I*. Indeed, in a model with static friction alone, a much weaker growth-rate dependence of radial expansion speed was obtained (*Figure 8F* blue triangles). Along a different line, Fisher-Kolmogorov (FK) dynamics has been used as a phenomenological model to describe radial colony expansion, and has been successful in describing certain spatial patterns formed in growing colonies (*Cao et al., 2016*). However, FK dynamics would predict a square-root dependence of the radial expansion speed on the cell growth rate (*Fisher, 1937*; *Kolmogorov et al., 1937*), which will need to be reformulated to conform to the observed dependences.

In addition to the well-known linear radial growth, the linear vertical growth of the colony is dissected for the first time qualitatively here since it was first reported (*Lewis and Wimpenny, 1981*; *Wimpenny, 1979*). Our analysis shows that the vertical expansion speed is limited by the depth that nutrient can penetrate upward into the colony from the agar. Accompanying our result of vertical growth is the predicted vertical orientation of cells in the colony interior, which transitions from the radial orientation at the outer periphery (i.e., the monolayer zone colored in cyan in *Figure 10*).

Cell verticalization has been observed experimentally for *Vibrio parahaemolyticus* (*Enos-Berlage and McCarter, 2000*) and for Vibrio Cholerae (*Beroz et al., 2018*; *Yan et al., 2016*). In both cases, vertical orientations could be seen already for very small bacterial colonies, possibly due to their production of extracellular polysaccharide substance (EPS). In this work, verticalization is predicted to occur for plain bacterial colonies as well, without the need of any EPS, but at much larger colony sizes. We have not been able to observe verticalization directly for our colonies due to multiple scattering associated with very dense colonies we are studying. This is left as a challenge for future studies.

In our model, verticalization results from an interplay among colony surface tension, cell-agar friction and the physical force of expansion due to cell growth. (*Beroz et al., 2018*) also introduced a

discrete model to describe cell verticalization. In their model, verticalization resulted from a similar mechanical instability due to the interplay between in-plane compression force and cell-agar adhesion. Due to the different energy barriers against verticalization, the length scales of verticalization between our model and that of (*Beroz et al., 2018*) are very different: The colonies in Beroz et al. spread very slowly radially ($\sim 3 \mu m/h$), and verticalization occurs at a colony radius of $\sim 5 - 10 \mu m$. Colonies in our model spread much faster ($\sim 14 \mu m/h$), and substantial verticalization occurs at a radius of $\sim 250 \mu m$; see *Figure 4—figure supplement 2*.

Although we have restricted our study to colonies growing in rather simple conditions, insight from our model can be readily used to make qualitative predictions in a variety of other conditions. Generally, we expect the radial expansion speed to be controlled by the buckling width and vertical expansion speed be controlled by the thickness of the growth zone. Thus, if agar hardness or ambient humidity is changed, the effect on air-liquid surface tension is expected to affect the buckling width and the ratio of the radial and vertical expansion speeds, hence changing the colony aspect ratio. Also, during later stages of colony growth when oxygen becomes limiting in the colony interior, the obligatory excretion of large amounts of fermentation product associated with anaerobiosis is predicted to lower the pH in colony interior and thereby slow down vertical colony growth while not affecting the radial growth. Our observations shown in *Figure 1H* and *Figure 1—figure supplement 1* are in qualitative agreement with the expectation. A quantitative study of this late regime ($t > 24$ h for the growth condition used in *Figure 1H*) requires a much more detailed model of anaerobic metabolism, pH effect, and growth transition kinetics, well beyond the scope of the current study, and will be reported elsewhere. Note that recent series of colony-based microbial range expansion studies (*Hallatschek et al., 2007*; *Hallatschek and Nelson, 2010*; *Korolev et al., 2012*), which involve much larger colony sizes and longer periods of colony growth, are likely in this late regime where vertical growth has ceased. Nevertheless, the radial expansion of these large colonies may still be governed by the same factors discussed in this work.

While our work is exclusively on bacterial colonies without EPS, key results we learned from this study shed light on the more complex dynamics of heterogeneous biofilms. First, we establish that the radial growth of our colonies is not limited by nutrient as commonly believed, but by the interplay of surface tension and cell-agar friction (*Figure 9*). Given that biofilms have typically much lower bacterial densities, nutrient limitation will be even less of a problem. Also, EPS secreted by the bacteria could modify both the surface tension and cell-agar friction to control the radial expansion speed. Second, nutrient supply is limiting for the vertical growth of our colonies (*Figure 9AC*). This becomes less of a problem for the loosely packed biofilms. Moreover, biofilms are said to form channels in their interior (*Wilking et al., 2013*), which would further alleviate the supply of nutrient, thereby allowing for faster vertical expansion. Finally, verticalization of cells in the interior, which is important for vertical growth but occurs at rather large colony sizes according to our model (*Figure 4—figure supplement 1*), also occurs in biofilms but at much smaller colony sizes (*Beroz et al., 2018*; *Enos-Berlage and McCarter, 2000*; *Yan et al., 2016*). While the precise nature of the forces driving verticalization may be different in the two cases, the underlying origins may be similar — mechanical instability due to in-plane compression resulting from colony expansion and cell-agar friction. In light of these comparisons, we see that the additional ingredients provided by biofilms enable the colonies to expand faster both horizontally and vertically.

The model presented here, with results quantitatively comparable to experimental data, can be used to interpret large-scale data being generated by high-throughput colony growth assays to track the growth of different strains in different conditions (*Takeuchi et al., 2014*). Our model can be used as a launching pad, not only to include the more complex effects of metabolism and cell growth mentioned here, but also other factors such as extracellular matrix to allow the simulation of biofilms, and multiple interacting species to explore microbial ecology in compact space. Finally, it will be an interesting challenge to develop coarse-grained hydrodynamic models that incorporate the unique features of surface tension and dynamic friction discussed here, and capture the radial and vertical colony growth characteristics, both their temporal behaviors and their dependences on cell growth rates and other environmental factors.

# Materials and methods

## Experimental methods

### Bacterial strain

The strain of *E. coli* K12 used in all the experiments reported in this work, EQ59, was derived from NCM3722 (*Lyons et al., 2011*), with deletion of the *motA* gene to remove bacterial motility and harboring constitutive GFP expression. We note that biofilm formation is highly suppressed in NCM3722, as acquired nonsense mutations within both the *bsg* and *csg* operons prevent the synthesis of extracellular cellulose and curli needed to support biofilm (*Lyons et al., 2011*; *Serra et al., 2013*).

To make strain EQ59, we cloned the *gfp* gene from pZE12G (*Levine et al., 2007*) into the KpnI/BamHI sites of the plasmid pKD13-rrnBT:Ptet (*Klumpp et al., 2009*), yielding the plasmid pKDT_Ptet-gfp. The fragment 'km$^r$:*rrn*BT:P*tet-gfp*' present in pKDT_Ptet-gfp was PCR amplified, gel purified and then electroporated into EQ42 cells (*Klumpp et al., 2009*), expressing the $\lambda$-Red recombinase. The cells were incubated with shaking at 37°C for 1 hour and then applied onto LB +Km agar plates. The Km$^r$ colonies were verified for the 'km$^r$:*rrn*BT:P*tet-gfp*' substitution for the 67 bp *intS/yfdG* intergenic region between 117th and 51st nucleotides relative to the start codon of *yfdG* by colony PCR and subsequently by sequencing. The chromosomal region carrying 'km$^r$:*rrn*BT:P*tet-gfp*' in EQ42 was then transferred to EQ54 (that is NCM3722$\Delta motA$) (*Kim et al., 2012*) by P1 transduction, yielding strain EQ59, in which the *gfp* gene is constitutively expressed in the absence of TetR.

### Growth medium

Phosphate-buffered media (N$^-$ C$^-$) was used for both batch culture and colony growth as described in *Csonka et al. (1994)*. Various carbon sources were used as specified in *Supplementary file 1*-Table S1. The concentration of all carbon sources used was 0.2% (w/v) unless otherwise specified. 10 mM of NH$_4$Cl was added as the sole nitrogen source. The agar concentration used was 1.5% (w/v) unless otherwise specified. 20 mL of molten agar gel was poured into 60 mm diameter dishes to a final thickness of approximately 7 mm, and allowed to cool at room temperature. Agar plates were sealed in plastic and stored at 4°C until use.

### Cell growth

Batch culture growth was performed in a 37°C water-bath shaker (220 rpm). Cells from a fresh colony in a LB plate were inoculated into LB broth and grown for several hours at 37°C as seed cultures. Seed cultures were then transferred into the desired minimal medium and grown overnight at 37°C as pre-cultures. For batch culture growth rate measurements, overnight pre-cultures were diluted to $OD_{600} \approx 0.01$ in the same minimal medium and grown at 37°C as experimental cultures. After two doublings, OD measurements were taken at various time over a 10-fold increase (i.e., from 0.04 to 0.4), and the growth rate was determined from a linear fit of ln(OD) vs. time.

Colonies were seeded on the agar gel as single cells. The pre-culture (prepared as above) was diluted to $OD_{600} \approx 10^{-6}$. 10 $\mu L$ of culture (containing approximately 10 cells) was spread over pre-warmed plates and transferred immediately to a 37°C incubator for growth. Petri dishes remained covered at all times, except during periodic measurements with a confocal microscope, in order to minimize moisture loss.

### Microscopy

Colonies were imaged with a Leica TCS SP8 inverted confocal microscope placed within an incubated box at 37°C. Samples were grown in covered petri dishes stacked on one side of the box. Each was moved to the microscope objective for periodic measurements. They were immediately covered once measurement was done. For the measurements, the dishes were uncovered and measurements were taken from the top (air) side to obtain a complete 3D image of the colony. GFP was excited with a 488 nm diode laser, and fluorescence was detected with a $10 \times /0.3$ objective and a high sensitivity HyD SP GaAsP detector. For a large colony, an xy-montage was created and stitched together to form a single 3D image using the ImageJ Grid/Collection Stitching plugin.

## Image analysis

The colony shape was obtained from the 3D confocal image using custom Matlab software. Under aerobic conditions, the bacterial fluorescence was spatially uniform near the top surface of the colony, and the surface height, h(x,y), could be reconstructed by simply thresholding the intensities: for each (x,y) position, the height was defined by the top pixel whose intensity was greater than the threshold. To account for the fact that fluorescence varied somewhat with growth conditions (sugar, agar concentration, etc.), this threshold was rescaled by the maximum fluorescence of the colony for each condition.

Furthermore, to capture the radius of the single- and multi-layers at early time of colony development (*Figure 1A–E*), we analyze the image intensity of the colony as the follows: for each stencil of $5 \times 5$ pixels centered at pixel $(i,j)$, we count the number of pixels whose intensity is above a threshold, and call it $n_{i,j}$. Pixel $(i,j)$ is assigned as type 1 if $16 > n_{i,j} \geq 3$, and as type 2 if $n_{i,j} \geq 16$, indicating the pixel belonging to single- or multi-layer region, respectively. We then estimated the inner radius $r_{inner}$ and outer radius $r_{outer}$ of the colony by the formulas $r_{inner} = r_{\mu m/px} \sqrt{N_{px2}/\pi}$ and $r_{outer} = r_{\mu m/px} \sqrt{(N_{px1} + N_{px2})/\pi}$, where $N_{px1}$ and $N_{px2}$ are the total numbers of pixels of type 1 and 2, respectively, and $r_{\mu m/px} \approx 0.84$ is the ratio of µm per pixel in our confocal image.

## Colony growth curves

Colony growth was monitored by measuring an individual colony at intervals of 1—4 hours. The radial growth curve, R(t), was extremely reproducible from colony to colony on the same agar plate and from day to day on different plates, up to a small offset in time, $t_l$, reflecting a variable lag time, of up to two hours before colony growth began. To monitor the colony growth over long periods of time, we started identical colonies at seed times $t_s$ separated by several hours. Growth curves extending over a period of multiple days could be obtained by stitching together $R(t - t_l - t_s)$ at times where they overlapped. This stitching procedure is illustrated in *Figure 1—figure supplement 1*. For example, in *Figure 1—figure supplement 1A*, there are three different symbols: triangles, squares, and circles. Each symbol represents data from one colony. They are seeded several hours apart and are plotted together with respect to their respective starting time. The data thus shows that the colony development is highly repeatable and can be put together to reconstruct the overall dynamics which spans a long period. In most cases, at least three separate colonies are measured concurrently for each (short) time span, and three separate time spans were stitched together in a series.

## Computational model

### Continuum model of nutrient dynamics

We assume that the growth of cells in the colony is limited by a single type of nutrient (the carbon source), and use a continuum scalar field $C = C\left(\vec{r}, t\right)$ to represent the nutrient concentration at a spatial location $\vec{r} = (x, y, z)$ and time $t$. Agar, which contains the nutrient and which cannot be penetrated by cells (at the dense concentrations used in out experiments), is confined to the region $z < 0$, while cells grow on top of the agar in the region $z > 0$, and bounded by the colony surface $\Gamma_{01}$ to be defined below; see *Figure 11*. Nutrient diffuses in the two compartments, agar and colony, according to the diffusion equations

$$\partial_t C = D_+ \Delta C - \rho \lambda / Y \qquad \text{for} \quad z > 0, \qquad (1)$$

$$\partial_t C = D_- \Delta C \qquad \text{for} \quad z < 0, \qquad (2)$$

with the distinct diffusion coefficients $D_+$ in the interstitial space between cells in the colony above the agar, and $D_-$ inside the agar. The second term on the right-hand side of *Equation 1* describes the rate of nutrient consumption by growing cells. Here, $\rho = \rho\left(\vec{r}, t\right)$ is the local cell mass density (total mass of cells in a unit volume of space) and $Y$ is the yield factor. For simplicity, we shall approximate the spatially and temporally varying cell mass density $\rho = \rho\left(\vec{r}, t\right)$ by a constant value $\rho_0$.

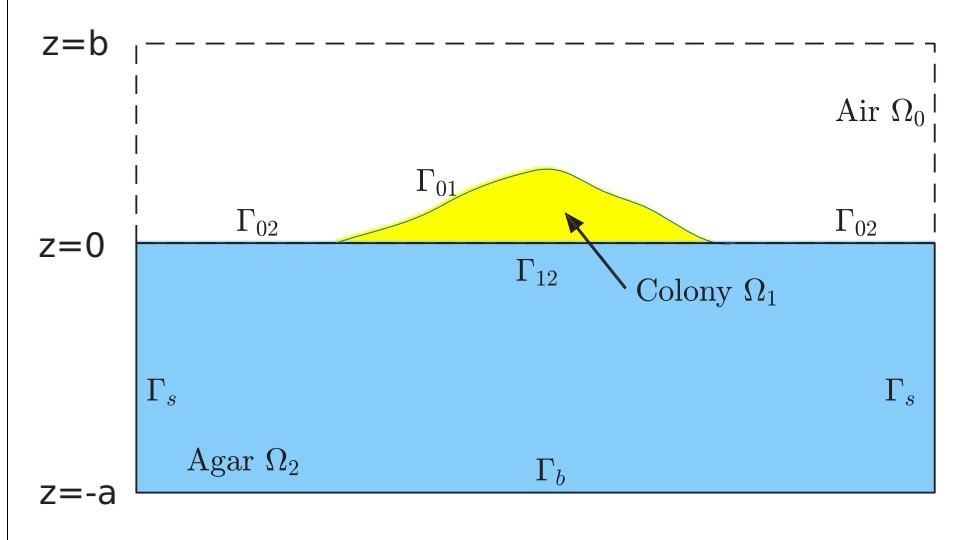

**Figure 11.** Schematic of the computational box and different regions in the model of simulation. The computational box is $\Omega = (-L,\ L) \times (-L,\ L) \times (-a,\ b)$, where all $L$, $a$, and $b$ are positive numbers in the units of length. This box is divided into the air region $\Omega_0$, colony region $\Omega_1$, and agar region $\Omega_2 = (-L,\ L) \times (-L,\ L) \times (-a,\ 0)$, respectively. See **Supplementary file 1**-Table S5 for typical values of $L$, $a$, and $b$ used in our simulations. The colony surface or colony-air interface $\Gamma_{01}$ separates the colony from air. The plane $z\ =\ 0$ in the computational box is divided into two parts. One is the interface that separates the colony from agar, and is denoted by $\Gamma_{12}$. The remaining part, denoted $\Gamma_{02}$, separates the air from agar. Note that, since the bacterial colony grows with time $t$, all the air region $\Omega_0$, the colony region $\Omega_1$, the colony-air interface $\Gamma_{01}$, and the colony-agar interface $\Gamma_{02}$ depend on time $t$.

DOI: https://doi.org/10.7554/eLife.41093.023

Above the spatial scale of a few cell lengths, the spatial variation in $\rho$ is $<5\%$ within the colony; see **Figure 3—figure supplement 1**. The upper boundary of the colony ($\Gamma_{01}$ in **Figure 11**) is defined by thresholding the density; see Appendix 1.2. The local mass growth rate $\lambda = \lambda\left(\vec{r},t\right)$ is given by Monod kinetics

$$\lambda\left(\vec{r},t\right) = \lambda_{\mathrm{S}} \frac{C\left(\vec{r},t\right)}{C\left(\vec{r},t\right) + K_{\mathrm{S}}}\ , \tag{3}$$

where $\lambda_{\mathrm{S}}$ is the batch culture growth rate for cells in a medium saturated with some sugar S, and $K_{\mathrm{S}}$ is the Monod constant for the sugar S. At the (mean) interface ($z=0$) between the colony and agar substrate, we have the continuity of the nutrient concentration and its flux:

$$C_- = C_+ \text{ and } D_- \partial_z C_- = D_+ \partial_z C_+ \quad \text{at the agar} - \text{colony interface } z = 0, \tag{4}$$

where the symbols $C_-$ and $C_+$ indicate the nutrient concentration on the agar side ($z < 0$) and colony side ($z > 0$), respectively. **Equations 1–4** are supplemented by boundary conditions imposed on the boundaries of a computational region comprising of both the colony and agar regions. We impose the flux-free boundary condition on the parts $\Gamma_{01}$, $\Gamma_{02}$, and $\Gamma_b$, and the Dirichlet boundary condition $C = C_s$ on the lateral wall of agar region $\Gamma_s$; cf. **Figure 11**. The parameter $C_s$ mimics the nutrient concentration far away from the colony. It is one of the key parameters in our study.

## Discrete model for cell growth, division and movement

In addition to modeling the nutrient as a continuum, we use a discrete, agent-based model to describe the growth, division, and movement of cells, as well as the interactions of cells with each other and with the environment. In this agent-based model, each *E. coli* cell is represented by a sphero-cylinder, comprised of a cylinder with hemispherical caps of diameter (also called cell width) $w_0$ on its two ends; see **Figure 12A**. For a given cell $i$ at a given time $t$, we use a position vector $\vec{r}_i(t)$

to denote the center-of-mass of the cell, a unit vector $\vec{n}_i(t)$ to denote its orientation (the direction along the cylindrical axis), and $l_i(t)$ to denote the length of the cylinder between the two centers of hemispherical caps. Each cell starts off with the same cylindrical length $l_0$. We assume that during cell growth, the width $w_0$ is fixed, and the cylinder length of a cell increases at a rate $\tilde{\lambda}(t)$. We call this the cell elongation rate. It is proportional to the mass growth rate $\lambda(t)$ of the cell with a geometrical proportionality factor $\sigma$: $\tilde{\lambda} = \sigma\lambda$; see Appendix 1.3.

The mass growth rate is calculated based on the nutrient concentration at the center $\vec{r}_i(t)$ of the cell $i$ at time $t$, that is $\lambda_i(t) = \lambda\left(\vec{r}_i(t), t\right)$, according to *Equation 3*. The growth of cylindrical length $l_i(t)$ is then given by the growth equation

$$\frac{d}{dt}l_i = \sigma\lambda\left(\vec{r}_i(t), t\right)l_i(t). \tag{5}$$

Once the cylindrical length $l_i(t)$ reaches a critical value $l_{\mathrm{div}}$, the cell divides into two daughter cells with cylindrical lengths being $l_0$ with small fluctuation; see *Figure 12B* and Appendix 1.3. For different growth media supporting different growth rates $\lambda_S$, the value of $l_{\mathrm{div}}$ is in general growth-rate dependent (*Jun and Taheri-Araghi, 2015*; *Taheri-Araghi et al., 2015*), the consequences of which are discussed above in 'Radial expansion – quantitative analysis'.

The position and orientation of cell $i$ change according to its velocity $\vec{v}_i$ and angular velocity $\vec{\omega}_i$, which follow Newton's second law

$$M_i\frac{\partial}{\partial t}\vec{v}_i = \vec{F}_i^{\,\mathrm{net}} \quad\text{and}\quad I_i\frac{\partial}{\partial t}\vec{\omega}_i = \vec{T}_i^{\,\mathrm{net}} \tag{6}$$

where $M_i$ and $I_i$ are the mass and moment of inertia of the cell, and $\vec{F}_i^{\,\mathrm{net}}$ and $\vec{T}_i^{\,\mathrm{net}}$ are the net force and net torque, respectively, exerted on that cell. As cells grow, divide and move, the colony region (defined by the part of boundary $\Gamma_{01}$ in *Figure 11*) expands. The nutrient concentration in the new domain requires an update by solving the boundary-value problem of *Equations 1–4* again.

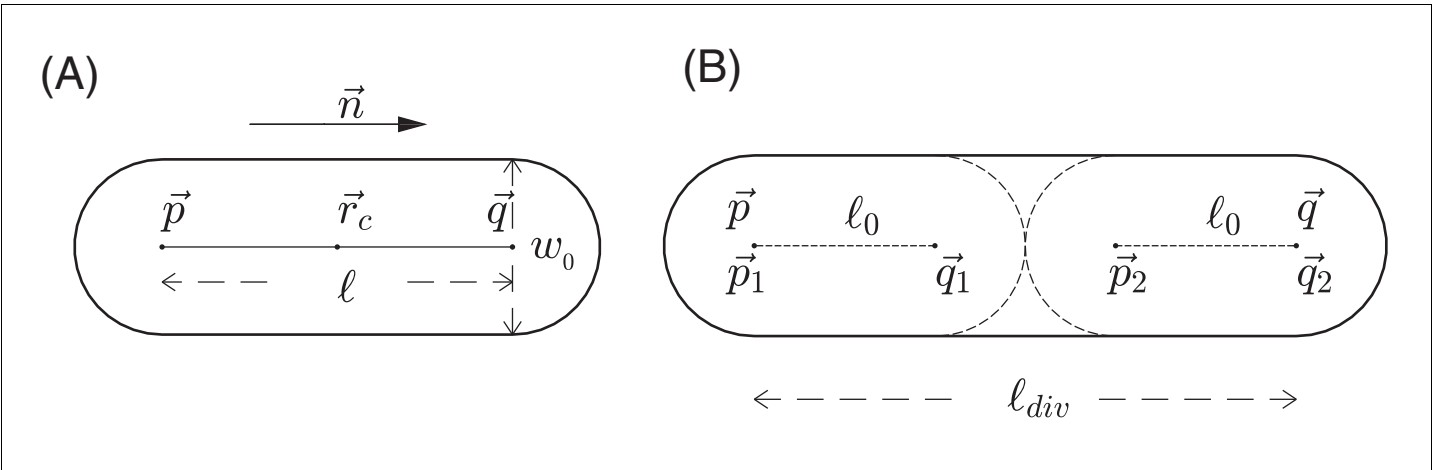

**Figure 12.** Schematic view of cell growth and division. (**A**) A sphero-cylinder model of an *E.coli* cell. Here, $w_0$ is the diameter of each of the hemispheres, $\vec{p}$ and $\vec{q}$ are the centers of these hemispheres, $l = |\vec{p} - \vec{q}|$ is the cylindrical length of the cell, $\vec{n} = \left(\vec{p} - \vec{q}\right)/l$ is the unit vector along the cylindrical axis of the cell, and $\vec{r}_c$ is the center of the cell. (**B**) Cell division. Once the cylindrical length of a cell reaches a critical value $l_{\mathrm{div}}$, the cell divides into two daughter cells. The two centers of hemispheres of the mother cell become the centers of hemispheres of the daughter cells. Each of these two daughter cells has the cylindrical length $l_0$ with fluctuations, where $l_0$ is a constant cylindrical length for any new born cell and any initial cell in the simulation. Fluctuations of angular velocities are also introduced for the daughter cells; cf. Appendix 1.3.
DOI: https://doi.org/10.7554/eLife.41093.024

## Discrete model for interaction forces

The net force $\vec{F}_i^{\,\text{net}}$ and torque $\vec{T}_i^{\,\text{net}}$ exerted on cell $i$ arise from (a) cell-cell mechanical interaction, (b) cell-agar interaction (if the cell touches the agar surface), (c) cell-fluid interaction, and (d) surface tension (if the cell is on top of the colony); cf. *Figure 2*. Below we briefly describe each component used in our model. Details are provided in Appendix 1.4.

### (a) Cell-cell interaction

In the interior of a colony, two cells interact only if they are in direct physical contact, characterized here by the overlap $\delta_{\text{cc}}$ in their sphero-cylinder cell boundaries; see *Figure 13A*. At the point of contact, the cell-cell interaction force $\vec{F}_{\text{cc}}$ is decomposed into the normal and tangential components, of magnitudes $F_{\text{cc},n}$ and $F_{\text{cc},t}$ as defined in and Appendix 1.4a. The normal force includes the Hertzian elasticity force with magnitude $F_{\text{cc,elas}} \propto \sqrt{w_0}\delta_{\text{cc}}^{3/2}$ (*Hertz, 1882*; *Johnson, 1985*). Additionally, the normal and tangential force each has a dissipation component, of magnitude $F_{\text{cc,disp},n}$ and $F_{\text{cc,disp},t}$, respectively, describing the effect of friction against cell movement. In the cell modeling literature (*Farrell et al., 2013*; *Ghosh et al., 2015*; *Rudge et al., 2013*; *Rudge et al., 2012*), these dissipation forces are often taken to be viscous. (We shall include such viscous force in (c) below.) In our model, we found it necessary to further include static and dynamic friction as described below.

We follow standard models of granular solids (*Brilliantov et al., 1996*; *Cundall and Strack, 1979*; *Kuwabara and Kono, 1987*; *Shäfer et al., 1996*), first introduced to cell modeling by Tsimring and

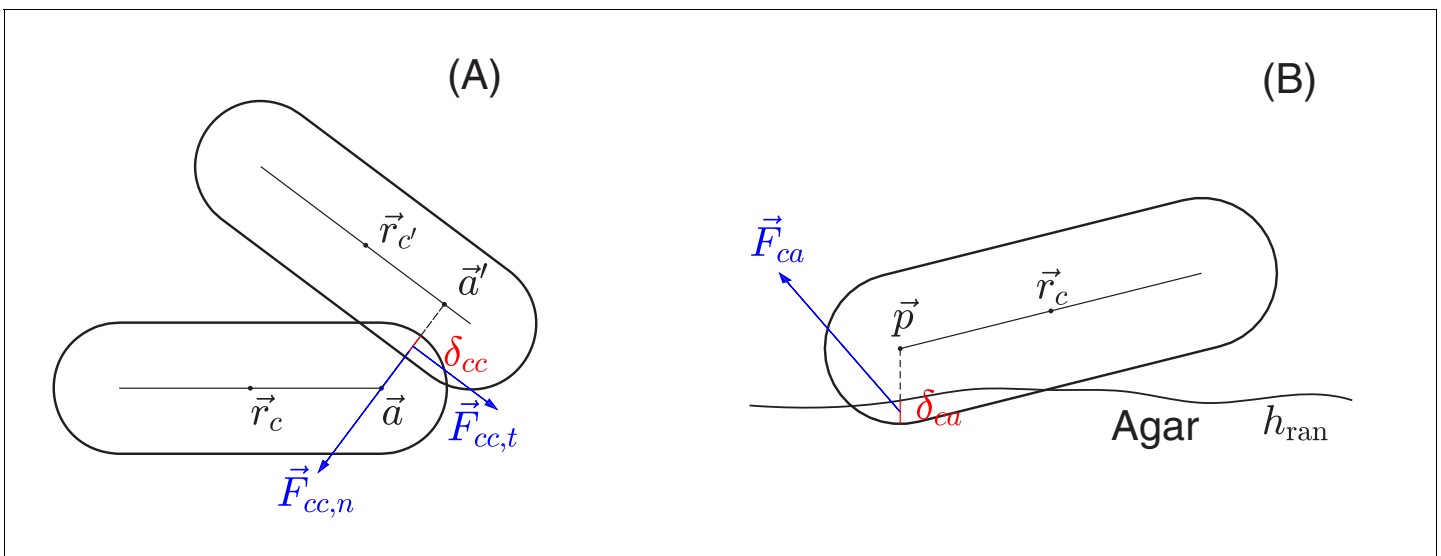

**Figure 13.** Schematic view of cell-cell and cell-agar interactions. (**A**) Cell-cell interaction. Two cells, centered at $\vec{r}_c$ and $\vec{r}_{c'}$, respectively, are in contact with each other. The shortest distance between the central cylindrical lines of the two cells is $d = \left| \vec{a} - -\vec{a}' \right|$ with $\vec{a}$ and $\vec{a}'$ two points on the cylindrical central lines of these two cells, respectively. The amount of the overlap of these two cells is $\delta_{\text{cc}} = w_0 - d$. We shall denote by $\vec{r}_{\text{cc}}$ the center of the line segment connecting $\vec{a}$ and $\vec{a}'$. The total interaction force, exerted at center $\vec{r}_{\text{cc}}$, is the sum of the normal force $\vec{F}_{\text{cc},n}$ in the normal direction $\vec{n}_{\text{cc}} = \left( \vec{a} - \vec{a}' \right)/d$ and the tangential force $\vec{F}_{\text{cc},t}$ in a direction orthogonal to $\vec{n}_{\text{cc}}$ that is determined by the relative velocities of these two cells; see the details in Appendix 1.4 on force calculations. (**B**) Cell-agar interaction. A cell, centered at $\vec{r}_c$, touches the agar surface that has the mean position at $z = 0$ and the roughness $h_{\text{ran}}$ (the maximum fluctuation around the mean), with $\delta_{\text{ca}}$ the amount of the overlap of the cell and agar. If the center of the hemispherical cap of the cell corresponding to the end that dips into the agar is $\vec{p} = (x_a, y_a, z_a)$, then $\delta_{\text{ca}} = w_0/2 - - z_a$. Denote $\vec{r}_{\text{ca}} = (x_a, y_a, z_a - w_0/2)$, which is the midpoint of the line segment along the vertical line passing through the point $\vec{p}$ between $z = 0$ and $z = -\delta_{\text{ca}}$. The total cell-agar interaction force $\vec{F}_{\text{ca}}$, exerted at the center $\vec{r}_{\text{ca}}$, is the superposition of a normal force $\vec{F}_{\text{ca},n}$ in the vertical direction and the tangential force $\vec{F}_{\text{ca},t}$ in a direction along the $xy$ plane that is determined by the velocity of the cell; cf. Appendix 1.4.
DOI: https://doi.org/10.7554/eLife.41093.025

his collaborators (**Volfson et al., 2008**). To model *static friction*, we adopt a fictitious drag force whose normal and tangential components, $F_{cc,disp,n}$ and $F_{cc,disp,t}$, respectively, are taken to be proportional to the normal and tangential components of the relative cell-cell velocity, $v_{cc,n}$ and $v_{cc,t}$. We use $F_{cc,disp,n} \propto \delta_{cc}^{1/2} v_{cc,n}$ and $F_{cc,disp,t} \propto \delta_{cc} v_{cc,t}$, where the additional dependences on the overlap $\delta_{cc}$ captures the dependence on contact area; see **Figure 13A**. To implement *dynamic friction*, we cap the tangential dissipation by the static yield criterion, that is $F_{cc,disp,t}^{max} = \mu_{cc} F_{cc,elas}$, where $\mu_{cc}$ is the dynamic frictional coefficient; see Appendix 1.4a. Thus, the full cell-cell interaction force is given by

$$F_{cc,n} = F_{cc,elas} + F_{cc,disp,n} \, , \tag{7a}$$

$$F_{cc,t} = \min\{F_{cc,disp,t} \, , \ \mu_{cc} F_{cc,elas}\}. \tag{7b}$$

As we see in 'Radial expansion – quantitative analysis', a dynamic friction form imposed by **Equation 7b** provides a natural explanation of the experimental observation that the radial velocity of the colony is independent of the cell growth rate.

## (b) Cell-agar interaction

The force exerted on a cell in contact with the agar, $\vec{F}_{ca}$, can be similarly calculated as sketched in **Figure 13B**. The same forms of the elastic and frictional forces are used as in **Equations 7a and b**. Note that to break the planar symmetry and facilitate buckling of cell layer into the vertical direction, we introduced certain roughness to the agar surface, characterized by the roughness parameter $h_{ran}$ which is the maximum fluctuation of the agar surface around its mean ($z = 0$).

## (c) Cell-fluid interaction

A cell also interacts with the surrounding fluid and experiences viscous drag. This is given by the Stokes drag force $\vec{F}_{visc}$, which is proportional to the velocity of the cell $\vec{v}$. We note that in high-density colonies such as those studied here, dissipation due to viscous drag is significantly less than the cellular friction force.

## (d) Surface tension

Surface tension is a critical factor determining the dynamics of an expanding colony. It is frequently treated as a property of a composite fluid comprising of cells plus the surrounding fluid (**Grimson and Barker, 1993**; **Zhang et al., 2008**). Alternatively, the liquid phase is ignored altogether, and surface tension is assumed to arise from attractive interactions between the cells themselves (**Ben-Jacob et al., 2000**). In both cases, surface tension reflects the curvature of the macroscopic colony profile. However, such coarse-grained treatments of surface tension cannot describe the initial layer-by-layer growth of the colony arising from buckling (**Figure 1A–E**), nor can they capture thin layers surrounding the periphery of large colonies (**Figure 14**). In order to capture these effects, we endeavor here to model the surface tension experienced by cells in a colony as a *boundary force*, that is as force experienced by discrete cells at the colony boundary due to increased surface tension of the continuum liquid these cells are immersed in.

For *E. coli* growing on hard agar, the cells themselves have no appreciable attraction to one another. They are instead held together at high densities in a colony above the agar through the surrounding liquid they share (blue color in **Figure 14A**): Liquid is pulled into and retained in the colony through the osmotic effects of hydrophilic molecules on cell surface (**Seminara et al., 2012**), wetting the surface of cells in the colony including those at the boundary. One can think of the cellular density within the colony as determined by the osmotic balance between the colony and the agar. This can be readily observed, as colonies grown on lower density agar are more liquid-like.

In the same way, the cohesion of a three-dimensional colony is maintained by the interaction of cells with the surrounding liquid at the air-liquid boundary. As shown in **Figure 14A**, the red curve indicates a smooth air-liquid boundary preferred by minimization of the liquid surface tension. Wherever a cell protrudes sharply out of the smooth surface, it drags out the liquid surrounding the cell, resulting in increased liquid surface tension, and hence a restoring force $F_{surf}$ acting on that cell. A detailed treatment of these physical effects, requiring both the cell configuration and the air-liquid boundary, is computationally untenable. Here, we do not model the liquid explicitly, but retain its

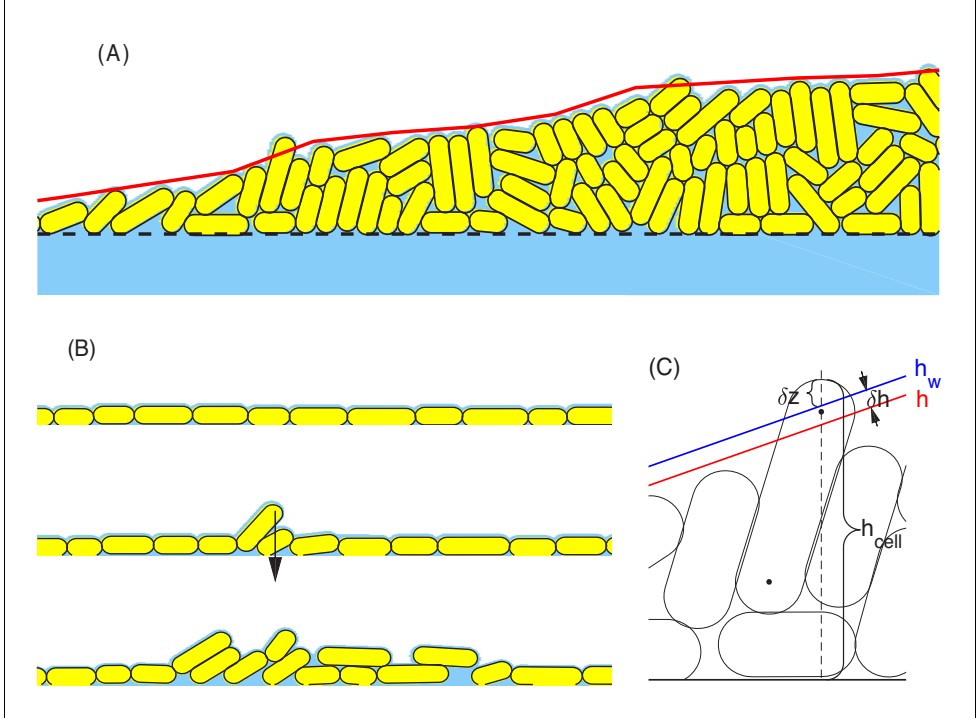

**Figure 14.** Schematic view of surface tension acting on cells at the colony boundary. (A) a snapshot of part of a growing colony from a typical computer simulation. The red curve defines the macroscopic colony-air interface which is used in updating the nutrient profile; cf. $\Gamma_{01}$ in **Figure 11** and Appendix 1.1. Cells on the top of the colony are held down by the surface tension force. (B) A sequence of configurations of colony during the early stage of growth. Top: Initial cells grow exponentially and form a monolayer. All cells in this layer are held down by the surface tension force. Middle: As more cells are born in the monolayer, the frictions between these cells and the rough agar surface increase. The competition between such frictions and the surface tension force that pulls down cells leads to an accumulation of the lateral pressure in the cells. The monolayer buckles up once that the surface tension can no longer hold down all the cells in the monolayer. Such buckling occurs at certain radial distance from the center of colony; cf. **Figure 8D**. Bottom: Once the monolayer buckles, the colony starts to grow vertically. In the meantime, the buckling region moves outward as the colony expands radially. (C) The parameters used in the definition of surface tension force. The parameter $h_{cell}$ is the height of a cell that sticks out of the macroscopic colony surface; it is measured from the mean height of the agar surface $z = 0$ to the 'tallest point' in the cell. The blue and red lines describe the macroscopic water level (indicated by $h_w$) and colony height (indicated by $h$), respectively. The parameter $\delta_h$ is used to control how tightly the surface tension holds back those cells on the top of the colony. See more details in Appendix 1.4c.

DOI: https://doi.org/10.7554/eLife.41093.026

effect on cells at the colony boundary via the restoring force. In Appendix 1.4c, we describe a toy model calculation which yields a saturating restoring force,

$$F_{\mathrm{surf},0} = \pi \gamma_{\mathrm{surf}} w_0 , \qquad (8)$$

whose magnitude is proportional to the width of the cell $w_0$ rather than the (much smaller) macroscopic curvature of the colony boundary. As shown in 'Radial expansion – quantitative analysis' and illustrated in **Figure 14B**, this large cell-level surface tension is able to hold a large group of cells in a monolayer above the agar surface, until the pressure inside the expanding monolayer (due to friction against motion on agar surface) exceeds a critical level to overcome the liquid surface tension resisting vertical protrusion, resulting in the 'buckling' of the monolayer into multiple layers.

To implement the surface tension force at the single-cell level in our model, we first compute the coarse-grained colony height $h(x, y, t)$ (red curves in **Figure 14AC**) from the cell configurations. Then we compute the height of the coated liquid (blue curve in **Figure 14C**) $h_w(x, y, t)$ by adding $\delta h$ to the colony height. This thickness $\delta h$ depends primarily on the agar hardness, being larger for softer agar where cells are less tightly bound by the liquid. For each cell whose maximum height $h_{cell}$ exceeds the liquid height $h_w$, we impose a restoring force normal to the liquid surface; see **Figure 14C**. As the magnitude of the saturating restoring force $F_{\mathrm{surf},0}$ (**Equation 8**) is independent of the height

difference $\delta z \equiv h_{\mathrm{cell}} - h_{\mathrm{w}}$ for $\delta z > 0$, the restoring force can be mathematically written as $F_{\mathrm{surf}} = F_{\mathrm{surf},0} \cdot u(\delta z)$, where $u$ is the Heaviside step function. To avoid numerical instability, we make a linear extrapolation between $F_{\mathrm{surf}} = 0$ and $F_{\mathrm{surf}} = F_{\mathrm{surf},0}$ over a narrow transition region $0 < \delta z < w_0/10$, which is $1/10$ of the cell width.

## Pressure calculation

Once all the individual forces exerted on a cell $i$ described above are calculated, the net force $\vec{F}_i^{\mathrm{net}}$ and the corresponding torque $\vec{T}_i^{\mathrm{net}}$ are calculated. Moreover, the pressure on the cell $i$ can be also calculated as

$$P_i = V_i^{-1} \sum_j \vec{F}_{ji} \cdot \vec{r}_{ji} \tag{9}$$

where $V_i$ is the volume of cell $i$, the index $j$ runs through all the different forces experienced by cell $i$, and $\vec{r}_{ji}$ are the corresponding displacement vectors from the points where the forces are exerted to the cell center.

## Coarse-grained variables

We define coarse-grained fields of cell spatial mass density $\rho\left(\vec{r},t\right)$, velocity $\vec{v}\left(\vec{r},t\right)$, directors $\vec{n}\left(\vec{r},t\right)$, and pressure $P\left(\vec{r},t\right)$ by averaging the corresponding individual quantities over small regions in the colony (e.g., a few finite-difference grid boxes); see Appendix 1.5 for details.

## Model parameters

We fix the coefficient of nutrient diffusion in the agar to be $D_- = 600 \ \mu\mathrm{m}^2/\mathrm{s}$, which is typical for the diffusion of small molecules in solution (*Beuling et al., 1996*; *Cole et al., 2015*). The diffusion coefficient in the colony is much smaller due to the fact that bacterial cells are not permeable to most sugars. We take $D_+ = 90 \ \mu\mathrm{m}^2/\mathrm{s}$ with the influence of volume fraction and tortuosity; see *Supplementary file 1*-Table S2 for details. We take the value of the yield factor for different sugars used to be that for glucose, which is $Y = 0.5 \ g_{\mathrm{CDW}}/g_{\mathrm{glucose}}$ (*Payne, 1970*). As shown above in 'Radial and vertical growth of the colony', the local cell mass density is found to vary only mildly around an average of $0.68 \ \rho_{\mathrm{cell}}$, with $\rho_{\mathrm{cell}} = 0.137 \times 10^{-12} \ g_{\mathrm{CDW}}/\mu\mathrm{m}^3$ being the cell dry weight density (*Basan et al., 2015*). The Monod constant is taken to be that for glucose, $K_{\mathrm{S}} = 20 \ \mu\mathrm{M}$ (*Monod, 1949*). The cell dividing length $l_{\mathrm{div}} = 3 \ \mu\mathrm{m}$ and the diameter $w_0 = 1 \ \mu\mathrm{m}$ are fixed in all the simulations unless otherwise indicated. The constant value $C_{\mathrm{s}}$ of nutrient concentration in the boundary conditions for the diffusion equations and the batch culture growth rate $\lambda_s$ are used as control parameters in our simulations. Other parameters that are crucial to the colony patterns and growth dynamics include various friction coefficients. See *Supplementary file 1*-Tables S2–S4 for the definitions and estimated values of all the parameters.

## Numerical implementation

We use an iteration algorithm for our simulations. It has two loops. The main loop, the 'outer loop', consists of the following 3 steps: (i) define the colony region using the local spatial cell density $\rho = \rho\left(\vec{r},t\right)$; (ii) update the nutrient concentration by solving the diffusion equations in steady state; and (iii) simulate cell growth, division, and movement over a small-time increment. The last step has its own loop, the 'inner loop', consisting of the following steps: update the local cell growth rate by *Equation 3*; simulate cell growth and division; and compute the forces and torques on cells, update the cell velocities and angular velocities, and update all the cell positions. We use the velocity Verlet algorithm, a commonly used molecular dynamics simulations of macromolecules, to update the cell velocities and positions (*Frenkel and Smit, 2002*). The inner loop is determined with a time step $\Delta t$. Usually, we run through one main loop per 100—1,000 inner loops. In updating the nutrient concentration, we use the finite difference to discretize the equations and the Jacobi or Gauss-Seidel relaxation method to solve the resulting systems of linear equations. We use multi-resolution

adaptive grids for a large computational domain, and use the OpenMP for parallelizing our code. See Appendix 1 for details. On a multi-processor (14-16 processors) computer, the simulation can reach a colony of a few million cells in 24 hours. We have placed the major and basic parts of our C++ codes in the repository GitHub (*Warren et al., 2019*; copy archived at https://github.com/elifes-ciences-publications/CellsMD3D).

## Acknowledgements

This work was partially supported by the NSF through grants DMS-1319731 and DMS-1620487 to BL, by Simons Foundation through grants #542387 to TH and #522790 to HS. MRW was supported by a Canadian Natural Sciences and Engineering Research Council post-doctoral fellowship. This work used the NSF Extreme Science and Engineering Discovery Environment (XSEDE) through grant ACI-1548562. We thank Agnese Seminara for useful discussions. We also thank the reviewers to bring to our attention the references Beroz et al. (Nat. Phys. 2018) and Enos-Berlage and McCarter (J. Bacteriol. 2000).

## Additional information

### Funding

| Funder | Grant reference number | Author |
| --- | --- | --- |
| National Science Foundation | DMS-1319731 | Bo Li |
| National Science Foundation | DMS-1620487 | Bo Li |
| Natural Sciences and Engineering Research Council of Canada | | Mya R Warren |
| Simons Foundation | 542387 | Terence Hwa |
| Simons Foundation | 522790 | Hui Sun |

The funders had no role in study design, data collection and interpretation, or the decision to submit the work for publication.

### Author contributions

Mya R Warren, Conceptualization, Investigation, Methodology; Hui Sun, Investigation, Visualization, Methodology, Writing—original draft; Yue Yan, Jonas Cremer, Investigation, Visualization, Methodology; Bo Li, Supervision, Writing—original draft, Writing—review and editing; Terence Hwa, Supervision, Methodology, Writing—original draft, Writing—review and editing

### Author ORCIDs

Hui Sun http://orcid.org/0000-0003-3137-3873
Terence Hwa http://orcid.org/0000-0003-1837-6842

### Decision letter and Author response

Decision letter https://doi.org/10.7554/eLife.41093.039
Author response https://doi.org/10.7554/eLife.41093.040

## Additional files

### Supplementary files

• Supplementary file 1. Supplemental tables S1-5.
DOI: https://doi.org/10.7554/eLife.41093.027

### Data availability

The simulation data files are large, hence we do not include it. All the simulation data can be generated from running the source code in GitHub (https://github.com/huiprobable/CellsMD3D; copy

archived at https://github.com/elifesciences-publications/CellsMD3D). Experimental source data files are provided for Figure1, Figure 1–figure supplement 1, and Figure 9.

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

# Appendix 1

DOI: https://doi.org/10.7554/eLife.41093.028

## Simulation Model and Methods

We model the colony expansion through the coupling of the growth, division, and movement of individual cells with the diffusion and reaction of nutrients and wastes. The growth of an individual cell within a short time period is described by a linear growth equation. The local growth rate varies spatially and temporally, and is determined locally by the cell density and nutrient supply. Once a cell grows into a critical size, it divides itself into two daughter cells with some randomness in their cylindrical lengths and orientations. In the meantime, growing and dividing cells push each other, generating mechanical forces. These forces, together with those arising from the cell-agar interactions, cell-liquid interactions, and surface tension, determine the movement of individual cells described by Newton's law of motion. At any instant of time, the coarse-graining of all cells through their spatial positions determines the cellular colony region. Nutrients and wastes diffuse in both the agar and colony regions, while their reactions only occur in the colony region. In our current study, we only include one species of nutrient and we do not consider any wastes.

### A1.1 Set Up and Main Algorithm

Our computational box is $\Omega = (-L, L) \times (-L, L) \times (-a, b)$, where all $L$, $a$, and $b$ are positive numbers in the units of length; cf. **Figure 11** in the main text. It is divided into the air region $\Omega_0$, colony region $\Omega_1$, and agar region $\Omega_2 = (-L, L) \times (-L, L) \times (-a, 0)$, respectively. The colony surface or colony-air interface $\Gamma_{01}$ separates the colony from air. The plane $z = 0$ in the computational box is divided into two parts. One is the interface that separates the colony from agar, and is denoted by $\Gamma_{12}$. The remaining part, denoted $\Gamma_{02}$, separates the air from agar. Note that, since the bacterial colony grows with time $t$, all the air region $\Omega_0$, the colony region $\Omega_1$, the colony-air interface $\Gamma_{01}$, and the colony-agar interface $\Gamma_{02}$ depend on time $t$. All the simulations of cell growth, division, and movement, and the force calculations are done in the colony region which expands with time. The reaction-diffusion equation for nutrient is solved in both the colony region $\Omega_1$ and the agar region $\Omega_2$ (where there is no reaction). The growth rate is also defined everywhere in the colony region.

We cover the computational box with a finite difference grid with a grid size $h_{\text{grid}}$. We generate random and small heights at the grid points on the mean agar surface ($z = 0$) to construct a rough agar surface. These random heights are in the range $[0, h_{\text{ran}}]$ with $h_{\text{ran}}$ an input number representing the possible maximum height. Such a rough surface will be used to calculate the interaction force between a cell and the agar. Initially, we set the nutrient concentration to be a nonzero constant in the agar region but zero elsewhere. We also randomly distribute a set of initial cells on the agar surface, and define their initial velocities and angular velocities to be all zero.

Our simulation continues through time iteration that consists of two loops. The main loop, or outer loop, consists of the following steps:

(1) Generate the boundary of colony;

(2) Update the nutrient concentration and cell growth rate;

(3) Simulate the cell growth, division, and movement.

The last step is carried out through an inner loop, a time iteration with time step $\Delta t$, that consists of the following steps:

(3.1) Simulate the cell growth and division;

(3.2) Compute the forces and torques on cells, and update the cell velocities and angular velocities with a half time step;

(3.3) Update all the cell positions;

(3.4) Compute again the cell forces and torques, and update the cell velocities and angular velocities with the other half time step;

(3.5) Set $t := t + \Delta t$ and continue with Step (3.1).

Note that Steps (3.2)–(3.4) are the velocity Verlet algorithm (cf. *Frenkel and Smit, 2002*) used for the simulation of cell movement. Practically, we update the nutrient concentration once every 100–1000 time steps of calculations of cell growth, division, and movement.

## A1.2 Nutrient and Growth Rate Update

### Cell density and colony boundary

Given the positions of all the cells at time $t$, we define the (local) volume fraction $\phi = \phi(\vec{r}, t)$ of the cells in each finite difference grid box above $z = 0$ by

$$\phi = \frac{\text{sum of volumes of the cells inside the grid box}}{\text{volume of grid box}}.$$

A cell is inside the grid box if the center of this cell is in this box. The volume of a cell is given by the formula in *Equation (A1.3.1)* in section A1.3 below. By averaging over nearest grid boxes, we obtain the volume fraction of cells at each grid point. We then define the cell mass density (at grid points) to be

$$\rho(\vec{r}, t) = \phi(\vec{r}, t)\rho_{\text{cell}}, \tag{A1.2.1}$$

where $\rho_{\text{cell}}$ is the constant mass density of a typical mature cell, and can be estimated from experiment; cf. *Supplementary file 1*-Table S3. We also define the colony region $\Omega_1 = \Omega_1(t)$ to be that with $\phi(\vec{r}, t) > 0$; cf. *Figure 11* in the main text.

### Nutrient update: Reaction-diffusion equations and boundary conditions

The concentration field $C = C(\vec{r}, t)$ of the nutrient is defined spatially on the colony and agar regions $\Omega_1 = \Omega_1(t)$ and $\Omega_2$, respectively; cf. *Figure 11* in the main text. It is governed by the following system of reaction-diffusion equations, interface conditions, and boundary conditions:

$$\frac{\partial C}{\partial t} = D_+ \Delta C - \frac{\lambda_S \rho}{Y} \frac{C}{C + K_S} \quad \text{in } \Omega_1(t), \tag{A1.2.2}$$

$$\frac{\partial C}{\partial t} = D_- \Delta C \quad \text{in } \Omega_2, \tag{A1.2.3}$$

$$C_- = C_+ \quad \text{on } \Gamma_{12}(t), \tag{A1.2.4}$$

$$D_- \frac{\partial C_-}{\partial z} = D_+ \frac{\partial C_+}{\partial z} \quad \text{on } \Gamma_{12}(t), \tag{A1.2.5}$$

$$\frac{\partial C}{\partial n} = 0 \quad \text{on } \Gamma_{01}(t) \cup \Gamma_{02}(t) \cup \Gamma_{\text{b}}, \tag{A1.2.6}$$

$$C = C_{\text{s}} \quad \text{on } \Gamma_{\text{s}}. \tag{A1.2.7}$$

The first two equations, *Equation (A1.2.2)* and *Equation (A1.2.3)*, are the reaction-diffusion equation for the concentration in the colony region $\Omega_1(t)$ and the diffusion equation for the concentration in the agar region $\Omega_2$, respectively, where $D_-$ and $D_+$ are the corresponding diffusion coefficients, $Y$ is the yield factor, $\lambda_S$ is the batch culture growth rate, $K_S$ is the capacity constant (the Monod constant) for sugar, and $\rho = \rho(\vec{r}, t)$ is the local cell mass density (cf. *Equation (A1.2.1)*). As the density of cells is rather uniform (cf. discussions in Appendix A2.1), we use a constant density value $\rho_0$ to approximate $\rho = \rho(\vec{r}, t)$. We take this constant $\rho_0$ to be the cell dry weight (CDW) per unit volume of the colony; cf. *Supplementary file 1*-Table S3. *Equation (A1.2.4)* and *Equation (A1.2.5)* are the interface

conditions for the concentration across the colony-agar interface $\Gamma_{12}(t)$ (i.e., $z = 0$), where $+$ and $-$ denote the colony side and agar side, respectively. The last two equations, *Equation (A1.2.6)* and *Equation (A1.2.7)*, are the boundary conditions. On the agar-air interface $\Gamma_{01}$, the colony-air interface $\Gamma_{02}$, and the bottom of agar $\Gamma_b$, we impose the flux-free boundary condition, with $\partial/\partial n$ denoting the derivative in the normal direction along the corresponding part of the boundary, pointing from the colony or agar to the air region or pointing downward from the bottom of agar. On $\Gamma_s$, the lateral faces of the agar region, we prescribe a constant value $C_s$ of nutrient concentration that represents the maximum nutrient that the system supplies.

In each time step, we update the nutrient by solving the steady-state reaction-diffusion equations with the corresponding boundary conditions, that is, *Equation (A1.2.2)–(A1.2.7)* with $\partial C/\partial t$ set to be 0. We use an iterative scheme to solve the equations with the previous nutrient concentration as the initial solution. This iterative scheme is constructed based on solving the corresponding time-dependent equations with the fixed colony region, and the time here is only a numerical parameter. We use the forward Euler's method to discretize this numerical time (cf. *Gustafsson et al., 2013*; *Morton and Mayers, 1995*). The spatial derivatives of concentration are discretized with central differencing schemes. For a grid point that is near the colony-air interface but is outside the colony region, we assign a value of nutrient concentration by interpolating the values of such concentration at nearby grid points inside the colony. To treat a large computational region $\Omega_2$ that represents the agar region, we use a nested, multi-level, finite difference grid as shown in *Appendix 1—figure 1*. Techniques of interpolation are employed to discretize the diffusion equation on grid points at the interface of grids with different levels. In each numerical time step, we sweep the grid points from top down to those at the interface $z = 0$, and further down to the bottom $\Gamma_b$, and then from grids on the bottom $\Gamma_b$ up to those at $z = 0$ and further up to the top. The numerical time iteration stops if the difference between the concentration fields of the current and previous numerical time steps is smaller than a given tolerance $err_{conv}$; cf. *Supplementary file 1*-Table S5. To ensure the numerical stability, we chose a numerical time step that approximately satisfies the CFL condition (*Gustafsson et al., 2013*; *Morton and Mayers, 1995*). We use OpenMP parallelization for both nutrient update and cell activities. Our simulations speed up more than 15 times in one 2.5 GHz Intel Xeon cluster node with 12 cores and 24 CPU processors.

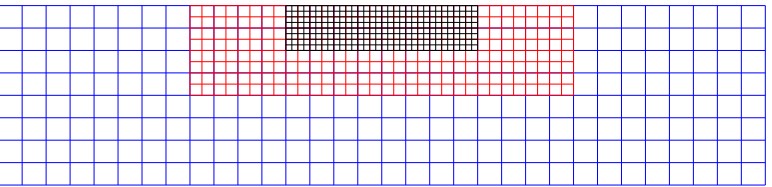

**Appendix 1—figure 1.** Schematic view of a nested finite difference grid for the agar region $\Omega_2$.
DOI: https://doi.org/10.7554/eLife.41093.029

## Growth rate update

Once the nutrient concentration field $C(\vec{r}, t)$ is updated, we can define the spatially and temporally varying local mass growth rate $\lambda = \lambda(\vec{r}, t)$ by

$$\lambda(\vec{r}, t) = \lambda_S \frac{C(\vec{r}, t)}{C(\vec{r}, t) + K_S}. \tag{A1.2.8}$$

This is first defined at each grid point $\vec{r}$ and then at any other point in the colony region by interpolation.

## A1.3 Cell Growth, Division, and Movement

We model an underlying *E. coli* bacterial cell as a sphero-cylinder; cf. *Figure 12A* in the main text. We denote by $\vec{p}$ and $\vec{q}$ the centers of the two hemispheres. We call $\ell = \|\vec{p} - \vec{q}\|$ the cell cylindrical length which can vary with time $t$. We also denote by $\vec{n} = (\vec{q} - \vec{p})/\ell$ the unit vector pointing from one center of hemisphere $\vec{p}$ to the other $\vec{q}$, and call it the direction of the cell. Note that the center of mass of the cell is $\vec{r}_c = (\vec{p} + \vec{q})/2$. We denote by $w_0$ the diameter of each of the hemispheres. The volume and mass of the cell are given by

$$V_{\text{cell}} = \frac{1}{4}\pi w_0^2 \ell + \frac{1}{6}\pi w_0^3 \qquad \text{and} \qquad M_{\text{cell}} = \rho_{\text{cell}} V_{\text{cell}}, \qquad (A1.3.1)$$

respectively, where $\rho_{\text{cell}}$ is the constant cell mass density introduced in *Equation (A1.2.1)*. We shall assume that all cells have the same diameter $w_0$ of hemispheres, independent of time.

### Cell growth

With our assumption, a cell grows only in its cylindrical length $\ell = \ell(t)$ but not in its diameter of hemispheres. The growth of the cell is governed by the growth equation for the cell cylindrical length

$$\frac{d\ell(t)}{dt} = \tilde{\lambda}(\vec{r}_c, t)\ell(t). \qquad (A1.3.2)$$

Here, $\tilde{\lambda}(\vec{r}_c, t)$ is the cell elongation rate evaluated at the cell center $\vec{r}_c$. The elongation rate $\tilde{\lambda}$ is proportional to the mass growth rate $\lambda$ defined in *Equation (A1.2.8)*: $\tilde{\lambda}(\vec{r}, t) = \sigma\lambda(\vec{r}, t)$. This effective relation between the two growth rates will be discussed after we describe the cell division. Numerically, we use the first-order approximation to obtain the cell cylindrical length at time $t + \Delta t$ by

$$\ell(t + \Delta t) = \ell(t) + \tilde{\lambda}(\vec{r}_c, t)\Delta t,$$

where $\Delta t$ is the simulation time step. Initially at $t = 0$, all the cells start with a constant cylindrical length $\ell_0$.

### Cell division

When a cell of centers of hemispheres $\vec{p}$ and $\vec{q}$ grows long enough, with its cylindrical length $\ell \geq \ell_{\text{div}}$ for some critical value $\ell_{\text{div}} > w_0$, it divides into two daughter cells of cylindrical lengths $\ell_1$ and $\ell_2$, respectively; cf. *Figure 12B* in the main text. The two centers of hemispheres of the mother cell become the centers of hemispheres of the daughter cells. The lengths $\ell_1$ and $\ell_2$ of these daughter cells are given by

$$\ell_1 = \frac{1}{2}\ell_{\text{div}} - \frac{w_0}{2} + \eta\ell_{\text{ran}} \qquad \text{and} \qquad \ell_2 = \frac{1}{2}\ell_{\text{div}} - \frac{w_0}{2} - \eta\ell_{\text{ran}},$$

where $\eta \in (-0.5, 0.5)$ is a random variable and $\ell_{\text{ran}}$ is an input positive number representing the maximum fluctuation of cell cylindrical length during the cell division; cf. *Supplementary file 1*-Table S3. In average, all new born cells have the same cylindrical length $\ell_0$; and we take it to be the same for all the initial cells. This implies that $\ell_{\text{div}} = 2\ell_0 + w_0$; cf. *Figure 12B* in the main text. Note that before division, the mother cell has volume $\pi w_0^2 \ell_{\text{div}}/4 + \pi w_0^3/6$; after division, the total volume of two daughter cells is $\pi w_0^2 \ell_{\text{div}}/4 + \pi w_0^3/12$. There is $\pi w_0^3/12$ volume loss every time due to division. For a fixed cell aspect ratio $\ell_{\text{div}} : w_0 = 3 : 1$, which is what we have in our simulations, the volume loss is about 9%. Note also that we have used a constant dividing length $\ell_{\text{div}}$ independent of the batch culture growth rate $\lambda_S$. The effect of this simplification is discussed in Appendix A2.2.

The centers of hemispheres of the two daughter cells are given by (cf. *Figure 12B* in the main text)

$$\vec{p}_1 = \vec{p}, \qquad \vec{q}_1 = \vec{p} - \frac{\ell_1}{\ell}(\vec{p} - \vec{q}), \qquad \vec{p}_2 = \vec{q} + \frac{\ell_2}{\ell}(\vec{p} - \vec{q}), \qquad \vec{q}_2 = \vec{q}.$$

Moreover, these daughter cells inherit the velocity from their mother cell. But the angular velocities of these two daughter cells are set to be

$$\vec{\omega}_1 = (0,0,0) \quad \text{and} \quad \vec{\omega}_2 = \omega_{\text{ran}}(0,0,\xi),$$

respectively, where $\xi \in (-0.5, 0.5)$ is a random variable and $\omega_{\text{ran}}$ is an input positive number that is the maximum fluctuation of the angular velocity; cf. *Supplementary file 1*-Table S3.

In the cell division, each of the two daughter cells also experiences the rotational fluctuation. Let us fix the center of mass $\vec{r}_c$ of a daughter cell. We rotate the vector $\vec{p} - \vec{r}_c$ with $\vec{p}$ the center of a hemisphere of this daughter cell. (For notational simplicity, we use $\vec{r}_c$ and $\vec{p}$ for this daughter cell; and they are different from those of the mother cell.) To do so, we construct a local Cartesian coordinate system with the origin at $\vec{r}_c$ and the three unit coordinate vectors $\vec{e}_1$, $\vec{e}_2$, and $\vec{e}_3$ by $\vec{e}_3 = (\vec{p} - \vec{r}_c)/\|\vec{p} - \vec{r}_c$,

$$\vec{e}_1 = \frac{(\vec{p} - \vec{r}_c) \times \arg\min_{\vec{i} \in \{\vec{e}_x, \vec{e}_y, \vec{e}_z\}} (\vec{p} - \vec{r}_c) \cdot \vec{i}}{\|(\vec{p} - \vec{r}_c) \times \arg\min_{\vec{i} \in \{\vec{e}_x, \vec{e}_y, \vec{e}_z\}} (\vec{p} - \vec{r}_c) \cdot \vec{i}\|},$$

and $\vec{e}_2 = \vec{e}_3 \times \vec{e}_1$, where $\vec{e}_x$, $\vec{e}_y$, $\vec{e}_z$ are the unit coordinate vectors in the original coordinate system. Then we rotate the vector $\vec{p} - \vec{r}_c$ by an angle of $\varphi$ around the axis $\vec{e}_2$, and $\theta$ around $\vec{e}_3$ successively. Here, $\varphi$ is a random variable in the range $[0, \varphi_{\text{ran}}\pi]$, where $\varphi_{\text{ran}}$ is a constant that sets the magnitude of fluctuations of $\varphi$ (cf. *Supplementary file 1*-Table S3) and $\theta \in [0, 2\pi)$ is a random number. If we denote by $\vec{p}_{\text{new}}$ the new center of hemisphere after the rotation, then

$$\vec{p}_{\text{new}} - \vec{r}_c = \|\vec{p} - \vec{r}_c\|(\sin\varphi\cos\theta\,\vec{e}_1 + \sin\varphi\sin\theta\,\vec{e}_2 + \cos\varphi\,\vec{e}_3).$$

This equation allows us to find the coordinates of $\vec{p}_{\text{new}}$ in the original coordinate system. The coordinates of the other center of hemisphere $\vec{q}_{\text{new}}$ are given by $\vec{q}_{\text{new}} = 2\vec{r}_c - \vec{p}_{\text{new}}$.

## The two growth rates

We assume that, in the life span of a cell, the cell mass growth rate $\lambda$ and the cell elongation rate $\tilde{\lambda}$ stay constant. The growth of mass satisfies the equation $M(t) = M(0)e^{\lambda t}$. If the mass doubling time is $t_d$, then $M(t_d) = 2M(0)$ and hence $\lambda t_d = \ln 2$. On the other hand, the doubling time is the time that a new born cell, which in average has the cylindrical length $\ell_0$, grows as its cylindrical length reaches the dividing length $\ell_{\text{div}}$. Note that $\ell_{\text{div}} = 2\ell_0 + w_0$. From the growth equation *Equation (A1.3.2)*, we have then $\ell_{\text{div}} = l_0 e^{\tilde{\lambda} t_d}$. Therefore,

$$\tilde{\lambda} t_d = \ln\frac{\ell_{\text{div}}}{\ell_0} = \ln\frac{2\ell_{\text{div}}}{\ell_{\text{div}} - w_0}.$$

Finally, $\tilde{\lambda} = \sigma\lambda$, where

$$\sigma = \frac{\tilde{\lambda}}{\lambda} = \frac{\ln(\ell_{\text{div}}/\ell_0)}{\ln 2} = \frac{\ln(2\ell_{\text{div}}/(\ell_{\text{div}} - w_0))}{\ln 2}.$$

For the fixed dividing cell aspect ratio $\ell_{\text{div}} : w_0 = 3 : 1$, we have $\sigma = \ln 3 : \ln 2$.

## Cell movement

Consider a cell at some time instant. Let us denote its centers of hemispheres by $\vec{p}_{\text{old}}$ and $\vec{q}_{\text{old}}$, its cylindrical length by $\ell_{\text{old}} = \|\vec{p}_{\text{old}} - \vec{q}_{\text{old}}\|$, its direction by $\vec{n}_{\text{old}} = (\vec{q}_{\text{old}} - \vec{p}_{\text{old}})/\ell_{\text{old}}$, and its mass by $M_{\text{old}}$. Let us also denote by $\vec{v}_{\text{old}}$ and $\vec{\omega}_{\text{old}}$ the velocity and angular velocity, respectively, at the center of the cell. We apply the velocity-Verlet algorithm to update the cell position, velocity, and angular velocity with the simulation time step $\Delta t$.

We first calculate the force $\vec{F}_{\text{half}}$ and torque $\vec{T}_{\text{half}}$ of the cell. Details of such calculations are given below in section A1.4. We then calculate the velocity $\vec{v}_{\text{half}}$ and angular velocity $\vec{\omega}_{\text{half}}$ for a half time step:

$$\vec{v}_{\text{half}} = \vec{v}_{\text{old}} + \frac{\Delta t}{2} \frac{\vec{F}_{\text{half}}}{M_{\text{old}}},$$

$$\vec{T}_{\text{half,n}} = (\vec{T}_{\text{half}} \cdot \vec{n}_{\text{old}})\vec{n}_{\text{old}},$$

$$\vec{T}_{\text{half,t}} = \vec{T}_{\text{half}} - \vec{T}_{\text{half,n}},$$

$$\vec{\omega}_{\text{half}} = \vec{\omega}_{\text{old}} + \frac{\Delta t}{2}\left(\frac{\vec{T}_{\text{half,n}}}{I_{\text{old,n}}} + \frac{\vec{T}_{\text{half,t}}}{I_{\text{old,t}}}\right),$$

where $I_{\text{old,n}}$ and $I_{\text{old,t}}$ are the moments of inertia of the cell along the directions $\vec{n}_{\text{old}}$ and its orthogonal, respectively. By direct calculations and using the constant density $\rho_{\text{cell}}$ in the place of the mass density of an underlying cell, we have

$$I_{\text{old,n}} = \rho_{\text{cell}} \pi w_0^4 \left(\frac{1}{32}\ell_{\text{old}} + \frac{1}{60}w_0\right),$$

$$I_{\text{old,t}} = \frac{1}{480}\rho_{\text{cell}} \pi w_0^2 \left(4w_0^3 + 15\ell_{\text{old}}w_0^2 + 20\ell_{\text{old}}^2 w_0 + 10\ell_{\text{old}}^3\right).$$

We now update the cell positions for the entire time step $\Delta t$ by updating the centers of hemispheres

$$\vec{p}_{\text{new}} = \vec{p}_{\text{old}} + \Delta t\left(\vec{v}_{\text{half}} + \vec{\omega}_{\text{half}} \times \frac{\vec{p}_{\text{old}} - \vec{q}_{\text{old}}}{2}\right),$$

$$\vec{q}_{\text{new}} = \vec{q}_{\text{old}} + \Delta t\left(\vec{v}_{\text{half}} + \vec{\omega}_{\text{half}} \times \frac{\vec{q}_{\text{old}} - \vec{p}_{\text{old}}}{2}\right).$$

We update the force and torque of the new cell with the centers of hemispheres $\vec{p}_{\text{new}}$ and $\vec{q}_{\text{new}}$ by the procedure of force calculations described below in section A1.4 to get $\vec{F}_{\text{new}}$ and $\vec{T}_{\text{new}}$. Finally, we update the velocity and angular velocity for the second half time step to get

$$\vec{v}_{\text{new}} = \vec{v}_{\text{half}} + \frac{\Delta t}{2} \frac{\vec{F}_{\text{new}}}{M_{\text{new}}},$$

$$\vec{T}_{\text{new,}n} = (\vec{T}_{\text{new}} \cdot \vec{n}_{\text{new}})\vec{n}_{\text{new}},$$

$$\vec{T}_{\text{new,}t} = \vec{T}_{\text{new}} - \vec{T}_{\text{new,}n},$$

$$\vec{\omega}_{\text{new}} = \vec{\omega}_{\text{half}} + \frac{\Delta t}{2}\left(\frac{\vec{T}_{\text{new,}n}}{I_{\text{new,}n}} + \frac{\vec{T}_{\text{new,}t}}{I_{\text{new,}t}}\right),$$

where $M_{\text{new}}$, $\vec{n}_{\text{new}} = (\vec{q}_{\text{new}} - \vec{p}_{\text{new}})/\ell_{\text{new}}$, and $\ell_{\text{new}} = \|\vec{p}_{\text{new}} - \vec{q}_{\text{new}}\|$ are the mass, direction, and cylindrical length, respectively, of the new cell with the centers of hemispheres $\vec{p}_{\text{new}}$ and $\vec{q}_{\text{new}}$, and the moments of inertia $I_{\text{new,}n}$ and $I_{\text{new,}t}$ can be calculated similarly using the cylindrical length $\ell_{\text{new}}$.

## A1.4 Force Calculations

Mechanical forces exerted on a cell include the elastic and dissipative forces from the cell-cell mechanical interactions, the elastic and dissipative forces from the cell-agar interaction if the cell is in contact with the agar, the surface tension force if the cell is on the top of the colony, and the viscous force from the interaction between the cell and the surrounding liquid. For a

given cell, we shall denote these forces by $\vec{F}_{\text{cc}}$, $\vec{F}_{\text{ca}}$, $\vec{F}_{\text{surf}}$, and $\vec{F}_{\text{visc}}$, respectively. So, the total force acting on the cell is

$$\vec{F} = \vec{F}_{\text{cc}} + \vec{F}_{\text{ca}} + \vec{F}_{\text{surf}} + \vec{F}_{\text{visc}}.$$

Note that most cells are in the interior of the colony; and they only experience the forces from the cell-cell and cell-liquid interactions.

## (a) Cell-cell interaction force

When two cells are in direct contact, they generate the cell-cell interaction force. As in **Volfson et al. (2008)**, we describe such forces using a standard model in granular solids (**Brilliantov et al., 1996**; **Cundall and Strack, 1979**; **Kuwabara and Kono, 1987**; **Makse et al., 2004**; **Shäfer et al., 1996**), with some adjustment based on experimental considerations on bacterial cells. An important part of our force scheme is a detailed treatment of the cell-cell frictional force, which together with the cell-agar friction are responsible for crucial mechanical behaviors such as buckling of the bacterial colony. While a similar grain-grain friction is commonly included in models of granular solids, however, friction in cellular models is often described to only include a purely viscous force (**Farrell et al., 2013**; **Ghosh et al., 2015**).

Let us fix a cell in the colony centered at $\vec{r}_c$, and call it a primary cell. Let us also fix a neighboring cell centered at $\vec{r}_{c'}$. We denote by $d$ the minimal distance between the central cylindrical line segments of the two cells, and by $\vec{a}$ and $\vec{a}'$ the points on these line segments, respectively, that reach this minimal distance:

$$d = \|\vec{a} - \vec{a}'\|;$$

cf. **Figure 13A** in the main text. We will describe an algorithm of finding the minimum distance $d$ and these two points $\vec{a}$ and $\vec{a}'$ at the end of this part. We denote by $\vec{n}_{\text{cc}}$ the unit vector along the direction from $\vec{a}'$ to $\vec{a}$, by $\vec{r}_{\text{cc}}$ the center of these two points, and by $\delta_{\text{cc}}$ the indentation size (i.e., the amount of overlap) between the two cells, respectively:

$$\vec{n}_{\text{cc}} = \frac{1}{d}(\vec{a} - \vec{a}'); \quad \vec{r}_{\text{cc}} = \frac{1}{2}(\vec{a} + \vec{a}'); \quad \delta_{\text{cc}} = \begin{cases} w_0 - d & \text{if } w_0 - d > 0, \\ 0 & \text{if } w_0 - d \leq 0. \end{cases}$$

Let $\vec{v}$, $\vec{v}'$ and $\vec{\omega}$, $\vec{\omega}'$ be the velocities and angular velocities of the primary and neighboring cells, respectively. The velocities $\vec{V}$ and $\vec{V}'$ of the two cells at the midpoint $\vec{r}_{\text{cc}}$ are then given by

$$\vec{V} = \vec{v} + \vec{\omega} \times (\vec{r}_{\text{cc}} - \vec{r}_c) \quad \text{and} \quad \vec{V}' = \vec{v}' + \vec{\omega}' \times (\vec{r}_{\text{cc}} - \vec{r}_{c'}),$$

respectively. Denote the difference of these velocities by

$$\vec{v}_{\text{cc}} = \vec{V} - \vec{V}'.$$

We call the unit vector $\vec{n}_{\text{cc}}$ the normal direction in the interaction of these two cells. We specify the tangential direction to be the unit vector

$$\vec{\tau}_{\text{cc}} = \frac{\vec{v}_{\text{cc}} - (\vec{v}_{\text{cc}} \cdot \vec{n}_{\text{cc}})\vec{n}_{\text{cc}}}{\|\vec{v}_{\text{cc}} - (\vec{v}_{\text{cc}} \cdot \vec{n}_{\text{cc}})\vec{n}_{\text{cc}}\|},$$

if the denominator is nonzero, that is the relative velocity $\vec{v}_{\text{cc}}$ has a nonzero tangential component. (Otherwise, $\vec{\tau}_{\text{cc}}$ can be any unit vector orthogonal to $\vec{n}_{\text{cc}}$.) Note that the tangential direction $\vec{\tau}_{\text{cc}}$ depends on the direction of the relative velocity $\vec{v}_{\text{cc}}$.

Let us assume now that these two cells are in direct contact with each other, that is $\delta_{\text{cc}} > 0$. The total interaction force $\vec{F}_{\text{cc}}$ between these two cells exerted at the center $\vec{r}_{\text{cc}}$ and the corresponding torque $\vec{T}_{\text{cc}}$ about the axis $\vec{r}_{\text{cc}} - \vec{r}_c$, are given, respectively, by

$$\vec{F}_{\text{cc}} = \vec{F}_{\text{cc},n} + \vec{F}_{\text{cc,t}},$$

$$\vec{T}_{\text{cc}} = (\vec{r}_{\text{cc}} - \vec{r}_{\text{c}}) \times \vec{F}_{\text{cc}}.$$

Here, $\vec{F}_{\text{cc,n}}$ is the normal force in the direction $\vec{n}_{\text{cc}}$ and $\vec{F}_{\text{cc,t}}$ the tangential force in the direction $\vec{\tau}_{\text{cc}}$. They are defined by

$$\vec{F}_{\text{cc,n}} = \left[ \frac{2}{3} k_{\text{cc}} \sqrt{w_0} \delta_{\text{cc}}^{3/2} - \gamma_{\text{cc},n} M_{\text{eff}} \delta_{\text{cc}} (\vec{v}_{\text{cc}} \cdot \vec{n}_{\text{cc}}) \right] \vec{n}_{\text{cc}}, \tag{A1.4.1}$$

$$\vec{F}_{\text{cc,t}} = -\min \left\{ \gamma_{\text{cc},t} M_{\text{eff}} \delta_{\text{cc}}^{1/2} | \vec{v}_{\text{cc}} \cdot \vec{\tau}_{\text{cc}} |, \frac{2}{3} \mu_{\text{cc}} k_{\text{cc}} \sqrt{w_0} \delta_{\text{cc}}^{3/2} \right\} \vec{\tau}_{\text{cc}}. \tag{A1.4.2}$$

Note that the final form of the force $\vec{F}_{\text{cc}}$ is the same as that in the main text; cf. **Equations 7a and b** there.

The first part in the normal force $\vec{F}_{\text{cc,n}}$ is the Hertz contact force resulting from the elastic collision of the two cells and pointing from the neighboring cell to the primary cell. Here, we approximate the cells as spheres of radius $w_0/2$. In such a case, the classical Hertz contact force is $(4/3) k_{\text{cc}} \sqrt{w_0/4} \delta_{\text{cc}}^{3/2}$, which is exactly what we have, where $k_{\text{cc}}$ is the reduced (or effective) elastic constant and $w_0/2$ is the reduced radius. If $E$ and $\nu$ are Young's module and Poisson's ratio of cells, then $k_{\text{cc}} = E/(2(1 - \nu^2))$ (cf. **Johnson, 1985**; **Landau and Lifshitz, 1986**; **Popov, 2010**). The second part in the normal force $\vec{F}_{\text{cc,n}}$ is the friction force in the normal direction $\vec{n}_{\text{cc}}$ due to the relative motion of the two cells, where $\gamma_{\text{cc},n}$ is the (normal) static friction coefficient and

$$M_{\text{eff}} = \rho_{\text{cell}} \frac{V_{\text{cell}} V'_{\text{cell}}}{V_{\text{cell}} + V'_{\text{cell}}}$$

is the reduced mass.

This form of the normal force **Equation (A1.4.1)** has been used in **Volfson et al. (2008)**. Similar forms of such a normal force can be found in the literature of granular solids (cf. **Brilliantov et al., 1996**; **Herrmann and Luding, 1998**; **Kuwabara and Kono, 1987**; **Makse et al., 2004**; **Shäfer et al., 1996**; **Silbert et al., 2001**). In particular, **Kuwabara and Kono (1987)** and **Brilliantov et al. (1996)** derived the normal force between two spherical granular grains, assuming such grains are viscoelastic. The frictional part in their derived normal force is proportional to $\sqrt{R_{\text{red}}} \sqrt{\delta} v$, where $R_{\text{red}}$ is the reduced radius and equals $R/2$ if both spheres have the same radius $R$, is the amount of overlap of the grains (same as our $\delta_{\text{cc}}$), and $v$ is the relative speed of the head-on collision of the grains (same as our $|\vec{v}_{\text{cc}} \cdot \vec{n}_{\text{cc}}|$). Since we have always used a fixed radius of the spherical caps of a cell in our simulations, with or without the factor $\sqrt{R_{\text{red}}}$ does not change noticeably our simulation results. However, unlike in granular material simulations where the grain size is fixed, a cell in the bacterial colony can have different mass and size at different stages. So, including the dependence on the effective mass is reasonable.

The magnitude of the tangential force $\vec{F}_{\text{cc,t}}$ in **Equation (A1.4.2)**, where $\gamma_{\text{cc},t}$ and $\mu_{\text{cc}}$ are the (tangential) static and dynamic friction constants, respectively, is the minimum of a tangential friction force that depends on the relative velocity $\vec{v}_{\text{cc}}$ and the dynamic friction force that is proportional to the Hertz contact force which is part of the normal force $\vec{F}_{\text{cc,n}}$ in **Equation (A1.4.1)**. This is different from that used in **Volfson et al. (2008)**: we only include the Hertz contact force here, not the entire normal force, as the elastic collision force can be dominated in cell-cell interactions.

To understand the form of the tangential component of the friction force, let us first consider the friction for two solid objects in contact. Coulomb's law for such friction states that the static tangential friction force $\vec{F}_{\text{static}}$ required to move the objects relative to each other and the dynamic tangential friction force $\vec{F}_{\text{dynamic}}$ between the objects to maintain such motion once it is initiated are both proportional to the normal force $F_{\text{normal}}$ pressing the objects together (**Popov, 2010**):

$$\vec{F}_{\text{static}} = \mu_s \vec{F}_{\text{normal}} \quad \text{and} \quad \vec{F}_{\text{dynamics}} = \mu_d \vec{F}_{\text{normal}},$$

where $\mu_s$ and $\mu_d$ are the static and dynamics friction coefficients, respectively. In general, $\mu_d < \mu_s$ and $\mu_d \approx \mu_s$; cf. **Appendix 1—figure 2A**. Here we assume for simplicity that $\mu_d = \mu_s$. In addition, we assume that the friction force is proportional to the relative velocity of the two cells in the tangential direction, with the proportion constant $\gamma$, when the tangential relative velocity is small; cf. **Appendix 1—figure 2B**. This allows the inclusion in the tangential friction of the dependence of the tangential velocity when it is small in magnitude and can be damped away quickly in the dynamics. Such an assumption is supported by our experimental observation that the radial velocity of colony expansion is linear in the batch culture growth rate but is independent of the individual cell velocity, as the 'buckling length' is independent of such local cell velocity. Our specific form of the tangential friction for small value of relative tangential velocity, that is the first part in the minimum of the tangential force $\vec{F}_{cc,t}$ in **Equation (A1.4.2)**, is the same as that in **Volfson et al. (2008)**. It is different from a form, popularly used in simulations of granular solids, that only includes the linear dependence on the relative tangential velocity but not the effective mass and the overlap distance $\delta_{cc}$ (cf. **Shäfer et al., 1996**) and the references therein. Moreover, for simplicity of simulations, we have also neglected the history dependence in the tangential velocity (**Cundall and Strack, 1979**; **Shäfer et al., 1996**).

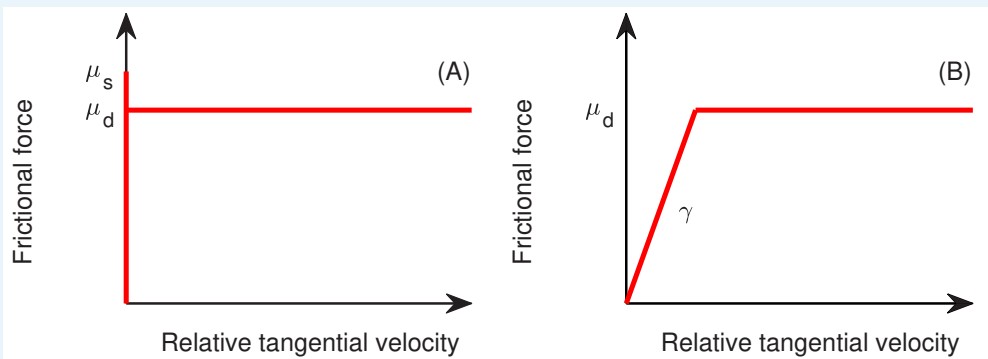

**Appendix 1—figure 2.** The standard (**A**) and modified (**B**) model for friction between two objects, as a function of tangential relative velocity between the two objects, where $\mu_s$ and $\mu_d$ are the static and dynamic friction constants, respectively.
DOI: https://doi.org/10.7554/eLife.41093.030

We end this part with a method of computing the minimum distance $d$ between the central cylindrical line segments $\gamma_1$ and $\gamma_2$ of the two cells and the corresponding points on these segments that reach this minimum distance. For notational convenience, let us denote by $\vec{p}_i$ and $\vec{q}_i$ the position vectors of the centers of hemispherical caps of the cell $i$ with $i = 1$ and 2, respectively. Let $\vec{u}_i = \vec{q}_i - \vec{p}_i$ We parameterize the central cylindrical line segments $\gamma_i$ by $\vec{r}_i(t) = \vec{p}_i + t\vec{u}_i$ for $0 \leq t \leq 1$ Denote $\vec{p}_0 = \vec{p}_1 - \vec{p}_2$ and define the distance-square function

$$f(s,t) = \|\vec{r}_1(s) - \vec{r}_2(t)\|^2 = \|s\vec{u}_1 - t\vec{u}_2 + \vec{p}_0\|^2.$$

Clearly, $f$ is minimized in $[0,1] \times [0,1]$ by some point $(s_0, t_0) \in [0,1] \times [0,1]$. The minimum distance $d > 0$, and the two points $\vec{a}_1$ and $\vec{a}_2$ on the two line segments reaching this distance are then given by

$$d^2 = f(s_0, t_0) = \min_{s,t \in [0,1]} f(s,t) \quad \text{and} \quad \vec{a}_1 = \vec{r}_1(s_0), \quad \vec{a}_2 = \vec{r}_2(t_0). \quad \text{(A1.4.3)}$$

We first assume that the two lines are not parallel, that is $\vec{u}_1 \times \vec{u}_2 \neq \vec{0}$. In this case, $f(s,t)$ is a strictly convex and quadratic function with a constant symmetric positive definite Hessian matrix. By setting $\partial_s f(s,t) = 0$ and $\partial_t f(s,t) = 0$, we find that $f$ is minimized in $\mathbb{R}^2$ at $(s_0, t_0)$ with

$$s_0 = \frac{(\vec{p}_0 \times \vec{u}_2) \cdot (\vec{u}_1 \times \vec{u}_2)}{\|\vec{u}_1 \times \vec{u}_2\|^2} \quad \text{and} \quad t_0 = \frac{(\vec{p}_0 \times \vec{u}_1) \cdot (\vec{u}_1 \times \vec{u}_2)}{\|\vec{u}_1 \times \vec{u}_2\|^2}.$$

If $(s_0, t_0) \in [0,1] \times [0,1]$, then we obtain $d$ and $\vec{a}_1$, by **Equation (A1.4.3)**. Otherwise, we compute the minimum value of $f$ on each of the four sides of the square $[0,1] \times [0,1]$ and compare these values to find the global minimum points $(s_0, t_0) \in [0,1] \times [0,1]$ and the corresponding minimum value of $f$ on this square. Consider, for instance, the side $s = 0$ and $0 \leq t \leq 1$. The function $f(0,t)$ for all $t \in \mathbb{R}$ is found to be minimized at $t_1 = (\vec{a}_0 \cdot \vec{u}_2)/\|\vec{u}_2\|^2$. If $t_1 \in [0,1]$, then the minimum value of $f$ on this side is $f(0, t_1)$. Otherwise, this value will be the minimum of $f(0,0)$ and $f(0,1)$.

We now assume that the two line segments $\gamma_1$ and $\gamma_2$ are parallel. In this case, the minimum distance $d$ is achieved by the minimum of the distance from $\vec{p}_1$ to the second line segment $\gamma_2$ and that from $\vec{q}_1$ to $\gamma_2$:

$$d = \min\{\text{dist}\,(\vec{p}_1, \gamma_2), \text{dist}\,(\vec{q}_1, \gamma_2)\}.$$

Each distance from a point to a line segment can be found by minimizing a convex quadratic function on $[0,1]$, similar to and simpler than the previous case.

### (b) Cell-agar interaction force

When a cell is in contact with the agar surface (including the case of the cell overlaps partially with the agar region), a cell-agar interaction force $\vec{F}_{ca}$ is generated and can be modeled similar to the cell-cell interaction force. Assume one end of the cell dips into the agar region; cf. **Figure 13B** in the main text. Let $\vec{p} = (x_a, y_a, z_a)$ be the center of the spherical cap corresponding to that end of the cell. We denote by $\delta_{ca}$ the indentation depth: $\delta_{ca} = w_0/2 - z_a$. We also denote $\vec{r}_{ca} = (x_a, y_z, z_a - w_0/2)$, which is the midpoint of the line segment along the vertical line passing through the point $\vec{p}$ between $z = 0$ and $z = -\delta_{ca}$. As before, we denote by $\vec{v}_{ca}$ the relative velocity at the center $\vec{r}_{ca}$. It is given by

$$\vec{v}_{ca} = \vec{v} + \vec{\omega} \times (\vec{r}_{ca} - \vec{r}_c),$$

where $\vec{v}$ is the velocity of the cell at its center $\vec{r}_c$ and $\vec{\omega}$ is the angular velocity of the cell. The normal direction is now the positive $z$-direction; we denote the unit vector along this direction by $\vec{n}_{ca} = (0,0,1)$. The tangential direction is defined through the relative velocity $\vec{v}_{ca}$ by

$$\vec{\tau}_{ca} = \frac{\vec{v}_{ca} - (\vec{v}_{ca} \cdot \vec{n}_{ca})\vec{n}_{ca}}{\|\vec{v}_{ca} - (\vec{v}_{ca} \cdot \vec{n}_{ca})\vec{n}_{ca}\|},$$

if the denominator is nonzero. (Otherwise, $\vec{\tau}_{ca}$ can be any unit vector orthogonal to $\vec{n}_{ca}$.)

Similar to the cell-cell interaction, the total cell-agar interaction force exerted on the midpoint $\vec{r}_{ca}$ and the corresponding torque are given by

$$\vec{F}_{ca} = \vec{F}_{ca,n} + \vec{F}_{ca,t},$$

$$\vec{T}_{ca} = (\vec{r}_{ca} - \vec{r}_c) \times \vec{F}_{ca}.$$

Here $\vec{F}_{ca,n}$ and $\vec{F}_{ca,t}$ are the forces normal and tangential to the mean colony-agar interface $z = 0$. They are given by

$$\vec{F}_{ca,n} = \left[\frac{2\sqrt{2}}{3} k_{ca} \sqrt{w_0} \delta_{ca}^{3/2} - \gamma_{ca,n} M_{eff} \delta_{ca} (\vec{v}_{ca} \cdot \vec{n}_{ca})\right] \vec{n}_{ca}, \tag{A1.4.4}$$

$$\vec{F}_{ca,t} = -\min\left\{\gamma_{ca,t} M_{cell} \delta_{ca}^{1/2} |\vec{v}_{ca} \cdot \vec{\tau}_{ca}|, \frac{2\sqrt{2}}{3} \mu_{ca} k_{ca} \sqrt{w_0} \delta_{ca}^{3/2}\right\} \vec{\tau}_{ca}. \tag{A1.4.5}$$

where $k_{ca}$ is an elastic constant in the Hertzian stress, $\gamma_{ca,n}$ and $\gamma_{ca,t}$ are static friction constants, $\mu_{ca}$ is the dynamic friction constant, and $M_{cell} = \rho_{cell} V_{cell}$ is the cell mass with $V_{cell}$ the cell

volume. Note that the factor $\sqrt{2}$ in the Hertz contact force part is different from those for the cell-cell interaction case. Here, the mean agar surface can be treated as a sphere of radius $\infty$ which leads to the reduced radius of the cell-agar system to be $w_0/2$.

In our numerical implementation, we do not decide which end of the cell dips into the agar region. Instead, we compute the corresponding forces at both ends and add them together.

## (c) Surface tension

Bacterial cells are hydrophilic. They are coated with water molecules. Once a bacterial cell sticks out of the colony surface, the tension between the air and water surface generates the surface tension force that brings down the cell. Such surface tension has long been recognized as a critical component in colony growth. Existing models of surface tension for colony, however, rarely treat the cells and the surrounding liquid as distinct media. Rather, the surface tension is frequently treated as a property of a composite fluid of cells plus liquid (**Grimson and Barker, 1993**; **Zhang et al., 2008**). Alternatively, the liquid phase is ignored and surface tension is assumed to arise from attractive interactions between the cells themselves (**Farrell et al., 2013**). In both cases, the surface tension scales with the macroscopic curvature of the colony. Here, we endeavor to model the surface tension force as a boundary force between the discrete cells and the continuum liquid.

The surface tension force can be calculated using the virtual work principle, $dW = \gamma_{\text{surf}}\, dA$, where $W$ is the work done by the water surface, $\gamma_{\text{surf}}$ is the water-vapor surface tension, and $dA$ is the change in water area $A$ as the cell is raised with an additional $dh$; cf. **Appendix 1—figure 3**. Here, we approximate a cell by a sphere of diameter $w_0$. (The notation for the radius is $R$ here.) The surface area is $A = 2\pi R h$ and hence $dA = 2\pi R dh$. (In the case of a disk in the two-dimensional setting, the change in area is $dA = 2\pi r R d\theta = 2\pi R^2 \sin\theta d\theta$. Note that $h = R\sin\theta$ and $dh = R\sin\theta d\theta$. Hence $dA = 2\pi R dh$.) As a result, the surface tension force is

$$F = \frac{dW}{dh} = 2\pi \gamma_{\text{surf}} R = \pi w_0 \gamma_{\text{surf}}.$$

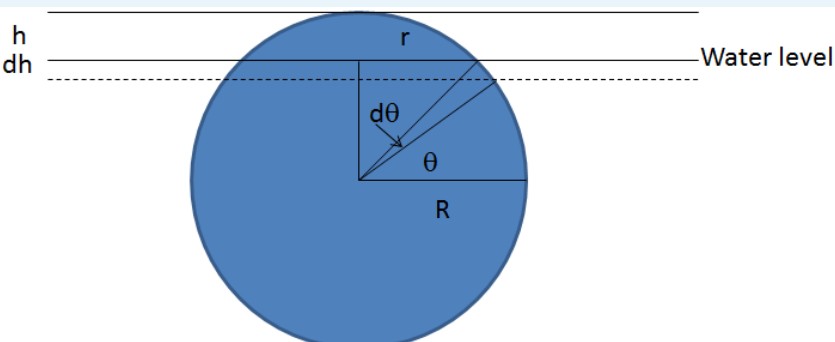

**Appendix 1—figure 3.** Schematic of the derivation of the surface tension force.
DOI: https://doi.org/10.7554/eLife.41093.031

Therefore, the surface tension is a constant force when a cell sticks out of the water. However, when a cell is below the water level, it should not experience any surface tension force. As a result, the surface tension force of a cell is discontinuous across the water surface. To reconcile this discontinuity, we make an approximation that the surface tension force increases linearly with the height until the height reaches some critical value $h_c \approx 0.1 w_0$, when it saturates to its maximum value of $\pi w_0 \gamma_{\text{surf}}$.

Specifically, let us consider a cell on top of the colony. Let $\vec{r}_c$ denote the center, and $\vec{p}$ and $\vec{q}$ the two centers of hemispheres of such a cell; cf. **Figure 14C** in the main text. We define the total surface tension force $\vec{F}_{\text{surf}}$ and torque $\vec{T}_{\text{surf}}$ on this cell to be

$$\vec{F}_{\text{surf}} = \vec{F}_{\text{surf},\vec{p}} + \vec{F}_{\text{surf},\vec{q}},$$

$$\vec{T}_{\text{surf}} = (\vec{p} - \vec{r}_{\text{c}}) \times \vec{F}_{\text{surf},\vec{p}} + (\vec{q} - \vec{r}_{\text{c}}) \times \vec{F}_{\text{surf},\vec{q}},$$

where

$$\vec{F}_{\text{surf},\vec{i}} = \pi w_0 \gamma_{\text{surf}} \min \left\{ \frac{\max\{0, (h_{\vec{i}} - z_{\vec{i}})\vec{n}_{\text{s}} \cdot \vec{n}_z + \delta h\}}{0.1 w_0}, 1 \right\} \vec{n}_{\text{s}}, \quad \vec{i} = \vec{p} \text{ or } \vec{q}.$$

Here, $h_{\vec{i}}$ is the water level at $\vec{i}$ or $\vec{q}$), $\vec{n}_z = (0,0,1)$ is the unit vector along the $z$ direction, $z_{\vec{i}} = \vec{i} \cdot \vec{n}_z$ is the $z$-component of $\vec{i}$, $\vec{n}_{\text{s}}$ is the unit vector of the colony surface, and $\delta h$ is a constant determining how tightly the surface tension holds the cells. The water level is coarse-grained. First, on each horizontal grid square $\mathcal{B}_{i,j} = (i\Delta x, (i+1)\Delta x) \times (j\Delta y, (j+1)\Delta y)$, we set the water level, $h_{i,j}$, at the center of this grid square to be the maximum of the $z$-coordinate of the two centers of hemispherical caps of cells whose centers are in the grid column $\mathcal{B}_{i,j} \times [0, b]$. (Recall that $z = b$ is the top of our computational box; cf. **Figure 11** in the main text.) Then we construct the coarse-grained water level everywhere by a continuous and piecewise linear interpolation at all $h_{i,j}$.

## (d) Viscous force

The cells in the colony experience a drag force—the Stokes drag force—due to their interactions with the surrounding liquid. Such viscous force $\vec{F}_{\text{visc}}$ exerted at a cell by the surrounding liquid and the corresponding torque $T_{\text{visc}}$ are given by (note that $w_0$ is the diameter)

$$\vec{F}_{\text{visc}} = -3\pi\mu_{\text{liq}} w_0 \vec{v} \quad \text{and} \quad \vec{T}_{\text{visc}} = -\pi\mu_{\text{liq}} w_0^3 \vec{\omega},$$

respectively, where $\vec{v}$ and $\vec{\omega}$ are the velocity and angular velocity, respectively, at the center of mass of the cell, and $\mu_{\text{liq}}$ is the liquid viscosity.

We finish our description of forces with a remark on the static and dynamic frictions. A static friction is proportional to the cell speed, while a dynamic friction is the same as the static friction for small speed but saturates after the speed is large; cf. **Appendix 1—figure 2**. In our tangential friction forces that arise from the cell-cell and cell-agar interactions, the saturation is controlled by capping through the elastic force; cf. **Equation (A1.4.2)** and **Equation (A1.4.5)**. To compare our dynamic friction forces with static friction forces alone, we shall consider a static friction model where the tangential friction forces in **Equation (A1.4.2)** and **Equation (A1.4.5)** are replaced by the following static frictions:

$$\vec{F}_{\text{cc,t}} = -\gamma_{\text{cc},t} M_{\text{eff}} \delta_{\text{cc}}^{1/2} |\vec{v}_{\text{cc}} \cdot \vec{\tau}_{\text{cc}}| \vec{\tau}_{\text{cc}} \quad \text{and} \quad \vec{F}_{\text{ca,t}} = -\gamma_{\text{ca},t} M_{\text{cell}} \delta_{\text{ca}}^{1/2} |\vec{v}_{\text{ca}} \cdot \vec{\tau}_{\text{ca}}| \vec{\tau}_{\text{ca}}, \qquad \text{(A1.4.6)}$$

respectively. The difference between the static and dynamic friction models is shown in **Figure 8F**, and the related discussions are given in Discussion in the main text.

## A1.5 Coarse-Grained Variables

To better present our simulation results, we need to coarse-grain the cell volume fraction $\phi$, pressure field $P$, velocity field $\vec{v}$, and director field $\vec{n}$ over a subregion $\mathcal{G}$ of the colony region. Examples of a subregion $\mathcal{G}$ include:

- The union of a few grid boxes for coarse-graining in the entire colony;
- A small box at the agar-colony interface $(i\Delta x, (i+1)\Delta x) \times (j\Delta y, (j+1)\Delta y) \times (0, \Delta z)$ for some $i, j$ for coarse-graining around such interface;
- A small box in a vertical layer $(-\Delta x, \Delta x) \times (j\Delta y, (j+1)\Delta y) \times (k\Delta z, (k+1)\Delta z)$ or $(i\Delta x, (i+1)\Delta x) \times (-\Delta y, \Delta y) \times (k\Delta z, (k+1)\Delta z)$ for some $i, j$, and $k$ for coarse-graining around the cross-section $x = 0$ or $y = 0$, respectively; and

- A cylindrical 'cube' $(i\delta r, (i+1)\delta r) \times (j\delta\theta, (j+1)\delta\theta) \times (k\Delta z, (k+1)\Delta z)$ in the cylindrical coordinates $(r, \theta, z)$ for some small $\delta r > 0$, $\delta\theta > 0$, and $\Delta z > 0$, and for some $i$, $j$, and $k$, for azimuthal coarse-graining.

Now let us fix a subregion $\mathcal{G}$ in the colony. Let $f$ denote the pressure $P$ or a component of the velocity field $\vec{v}$, and denote by $f_i$ such a quantity at the center of cell $i$. We define the coarse-grained average of $f$ over $\mathcal{G}$ to be

$$\bar{f}(\mathcal{G}) = \frac{\sum_{\vec{r}_{c_i} \in \mathcal{G}} f_i}{\sum_{\vec{r}_{c_i} \in \mathcal{G}} 1},$$

where $\vec{r}_{c_i}$ is the center of mass for the cell $i$. Similarly, we define the coarse-grained volume fraction over the subregion $\mathcal{G}$ to be

$$\bar{\phi}(\mathcal{G}) = \frac{\sum_{\vec{r}_{c_i} \in \mathcal{G}} V_i}{\mathrm{Vol}(\mathcal{G})},$$

where $V_i$ is the volume of cell $i$ and $\mathrm{Vol}(\mathcal{G})$ is the volume of the subregion $\mathcal{G}$. Note that, if $\mathcal{G}$ is a grid box, then $\bar{\phi}(\mathcal{G})$ is the same as the volume fraction $\phi$ on that box; cf. section A1.2. Note also that the cell density $\rho$ can be coarse-grained following its relation with the volume fraction $\phi$; cf. **Equation (A1.2.1)**.

The director field $\vec{n}$ cannot be coarse-grained simply by summing over the directors in a subregion, since the director of a given cell can have two possible directions and the sum can lead to artificial cancellations. Here, we define the coarse-grained director field $\vec{n}(\mathcal{G})$ over a given subregion $\mathcal{G}$ to be a maximizer of the maximization problem

$$\max_{\vec{n}} \sum_{\vec{r}_{c_i} \in \mathcal{G}} |\vec{n} \cdot \vec{n}_i|^2 \quad \text{subject to } \|\vec{n}\| = 1.$$

By the Lagrange multiplier method, this leads to an eigenvalue problem for a 3-by-3 matrix, with the maximizer $\vec{n}(\mathcal{G})$ being a unit eigenvector that corresponds to the largest eigenvalue.

Let $\delta r > 0$ be small. Let $N_\theta \geq 2$ be an integer and define $\delta\theta = 2\pi/N_\theta$. We denote $\mathcal{G}_{i,j,k} = (i\delta r, (i+1)\delta r) \times (j\delta\theta, (j+1)\delta\theta) \times (k\Delta z, (k+1)\Delta z)$ in the cylindrical coordinates $(r, \theta, z)$, We define the azimuthal average of a scalar field $f$ by

$$\tilde{f}(i\delta r, k\Delta z) = \frac{1}{N_\theta} \sum_{j=0}^{N_\theta - 1} \bar{f}(\mathcal{G}_{i,j,k}),$$

where $\bar{f}(\mathcal{G}_{i,j,k})$ is the the coarse-grained average of $f$ over $\mathcal{G}_{i,j,k}$. The azimuthal average of a vector field can be defined componentwise.

Given any point in the agar-colony interface with the polar coordinates $(r, \theta)$, we define

$$\Delta r = r - R(\theta), \quad \text{where } R(\theta) = \max_{\rho(r',\theta) > 0} r', \tag{A1.5.1}$$

where $\rho(r', \theta)$ is the local cell density projected onto the colony-agar interface. This is the negative distance between this point and the colony edge along the ray of angle $\theta$. For each integer $j$ with $0 \leq j \leq N_\theta$, we denote by $i_j$ the largest integer not exceeding $(R(j\delta\theta) + \Delta r)/\delta r$. We then define the azimuthal average of the radial component $V_r$ of the velocity field $\vec{v}$ by

$$V_r(\Delta r) = \frac{1}{N_\theta} \sum_{j=0}^{N_\theta - 1} \bar{\vec{v}}(\mathcal{G}_{i_j, j, 0}) \cdot \vec{r}_j, \tag{A1.5.2}$$

where $\vec{r}_j = (\cos(j\delta\theta), \sin(j\delta\theta))$. Other azimuthal-averaged quantities in terms of $\Delta r$ can also be defined similarly.

## Appendix 2

DOI: https://doi.org/10.7554/eLife.41093.028

# Additional Results of Simulation and Analysis

Unless otherwise stated, all the notations and terms are the same as defined in the main text and Appendix 1.

## A2.1 Constant Density and Volume Fraction

For each of the batch culture growth rates $\lambda_S = 1\,h^{-1}$ and $0.5\,h^{-1}$, we simulated the growth of bacterial colony. **Figure 3—figure supplement 1** in the main text shows our simulation results on the volume fractions and their course-grained values. We observe that the volume fraction of cells in the colony has the mean value around 0.68 with the standard deviation around 0.03 for both of the batch culture growth rates $\lambda_S = 1\,h^{-1}$ and $0.5\,h^{-1}$. This suggests that we can treat the cell volume fraction as a constant inside the colony. Hence, we can also approximate well the density $\rho$ by a constant $\rho_0$; cf. **Equation (A1.2.1)** in Appendix 1.

## A2.2 Effect of $\lambda_S$-Dependence on Cell Dividing Length

It has been known that cell dividing length $\ell_{\mathrm{div}}$ may vary with the batch culture growth rate $\lambda_S$ according to the relation (cf. **Donachie, 1968**; **Jun and Taheri-Araghi, 2015**; **Wallden et al., 2016**)

$$V_{\mathrm{div}} = V_0 e^{\lambda_S/\lambda_{S0}}, \tag{A2.2.1}$$

where $V_{\mathrm{div}}$ is the corresponding dividing volume of the cell, $V_0$ is a constant, and $\lambda_{S0} = 1.0\,h^{-1}$. In our simplified computational model, we have not included such variations. Here, we study the effect of the $\lambda_S$-dependence on the cell dividing length $\ell_{\mathrm{div}}$ by computer simulations.

Based on recent experimental observations on the size on *E. coli* cell (**Jun and Taheri-Araghi, 2015**; **Si et al., 2017**), we have the ratio

$$\ell_{\mathrm{div}} : w_0 = 3 : 1 \tag{A2.2.2}$$

for all the batch culture growth rate $\lambda_S$. This and **Equation (A2.2.1)**, together with the formula for cell volume (cf. **Equation (A1.3.1)** in Appendix 1), then imply that the cell dividing length should be given by

$$\frac{11}{324}\pi\ell_{\mathrm{div}}^3 = V_0 e^{\lambda_S/\lambda_{S0}}.$$

Setting $\lambda_S = \lambda_{S0} = 1\,h^{-1}$ in **Equation (A2.2.1)** and using the assumption **Equation (A2.2.2)**, we obtain also that

$$\frac{11}{324}\pi\ell_{\mathrm{div},0}^3 = V_0 e,$$

where $\ell_{\mathrm{div},0}$ is the cell dividing length for $\lambda_S = \lambda_{S0}$. We now assume that this cylindrical length is $\ell_{\mathrm{div},0} = 3\,\mu m$ (**Jun and Taheri-Araghi, 2015**; **Si et al., 2017**). The above two equations then provide us with the cell dividing length

$$\ell_{\mathrm{div}} = 27 e^{\lambda_S/\lambda_{S0}-1}. \tag{A2.2.3}$$

We chose $\lambda_S = 0.1\,h^{-1}, 0.2\,h^{-1}, \dots, 1.0\,h^{-1}$. For each of these values of $\lambda_S$, we ran simulations with the variable dividing length $\ell_{\mathrm{div}}$ determined by **Equation (A2.2.3)**. We also ran simulations with a fixed diving length $\ell_{\mathrm{div}} = 3\,\mu m$; cf. **Supplementary file 1**-Table S3. In **Figure 5C** in the main text, we plot the (constant) vertical ascending speed $V_H$ vs. $\lambda_S$ for both

fixed (open circles) and variable (filled circles) dividing lengths. We observe that the results from a fixed dividing length are consistent with those from a variable dividing length. In **Figure 8F** in the main text, we see that the (constant) radial velocity $V_R$ obtained with a fixed dividing length is close to that with a variable dividing length, but the discrepancy is more significant than the case of $V_H$. In **Appendix 2—figure 1** below, we plot the velocities $V_R$ and $V_H$ for variable dividing length, and fit the simulation data with $\lambda_S \geq 0.5\,\mathrm{h}^{-1}$. We observe that the straight line that fits simulated $V_R$ intersects the $x$-axis at $\sim 0.2\,\mathrm{h}^{-1}$. This agrees well with the experimental result plotted in **Figure 1H** in the main text. Hence, the inclusion of $\lambda_S$-dependence on the cell dividing length can better describe experiment.

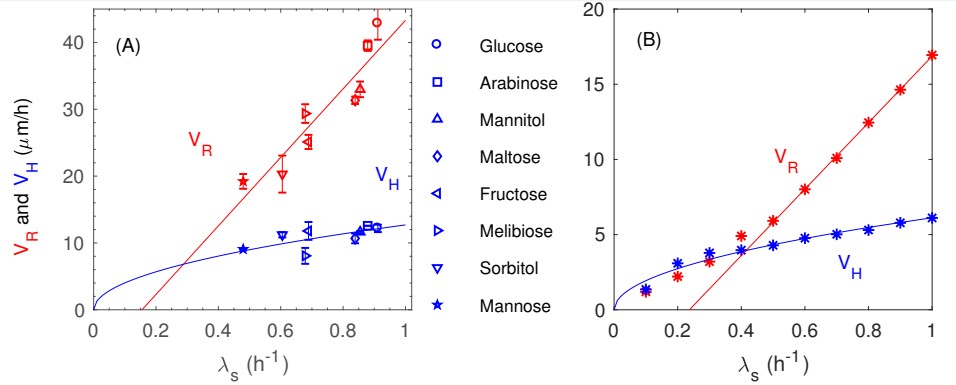

**Appendix 2—figure 1.** Experimental measurement on $V_R$ and $V_H$ with various batch culture growth rates. Data are fitted with the stright line $V_R = 51.27\lambda_S - 7.96$ and the squre root curve $V_H = 12.7\sqrt{\lambda_S}$ for the redial and vertical speeds of expansion $V_R$ and $V_H$ respectively. (B) Simulation results on $V_R$ and $V_H$ with various batch culture growth rates and with a variable diviting length $l_{\mathrm{div}}$. data are fitted for $\lambda_S \geq 0.5\,\mathrm{h}^{-1}$ using the straight line $V_R = 22.1\lambda_S - 5.24$ the squre-root curve $V_H = 6.12\sqrt{\lambda_S}$, respectively.
DOI: https://doi.org/10.7554/eLife.41093.033

We now show that using a fixed cell dividing length $\ell_{\mathrm{div}}$ independent of the batch culture growth rate $\lambda_S$ is a reasonable simplification for our simulations. We fixed the batch culture growth rate $\lambda_S = 1.0\,\mathrm{h}^{-1}$ and the constant concentration in the boundary condition $C_s = 2.0\,\mathrm{mM}$. We then distributed randomly 625 cells at time $t = 0\,\mathrm{h}$. At $t = 9.0\,\mathrm{h}$, there are around 0.16 million cells in the colony. We picked randomly $2,000$ of them from the bottom layer, and then tracked the local mass growth rate of each of these cells from $t = 9.0\,\mathrm{h}$ to $t = 15\,\mathrm{h}$. If a cell divides during this time period, we kept track one of its two daughter cells. We found that the local growth rates of about 80% of these cells change from high values, larger than 90% of $\lambda_S$, to low values, smaller than 10% of $\lambda_S$, during this time period of colony growth. This indicates that the majority of the cells go through a complete transition in local growth rates.

We now consider those 80% cells that experience the transition from high to low growth rates. **Appendix 2—figure 2A** is the histogram of the number of cell generations (i.e., the number of divisions) of these cells. It is clear that during the transitioning time period, most of these cells did not divide or only divided once, signaling the sharpness of the high-to-low growth rate transition. To better understand such sharpness, we selected randomly 100 cells which complete the high-to-low transition, and tracked the growth rate of each of these cells during the time period from $t = 9.0\,\mathrm{h}$ to $t = 16\,\mathrm{h}$. In **Appendix 2—figure 2B**, we plot the local growth rate vs. shifted time for each of these 100 cells. For a given cell, we shifted the time so that the growth rate of $\lambda_S/2$ was reached at the shifted time 0. All these indicate that cells pass the transitioning region in a short time, and it is therefore reasonable to assume a fixed $l_{\mathrm{div}}$ and $w_0$ for the entire simulation.

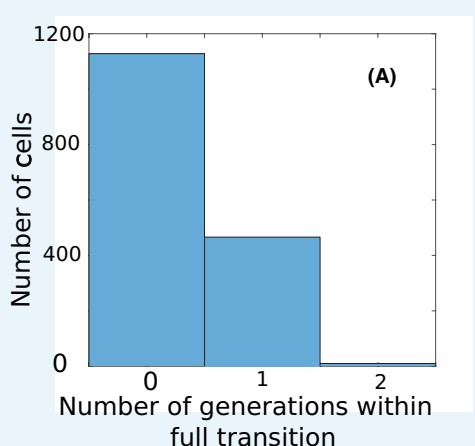
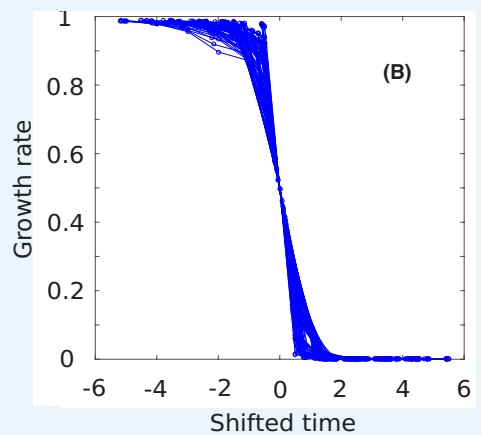

**Appendix 2—figure 2.** Statistics of cells undergoing growth transition. (**A**) The histogram of the number of generations of the cells that experienced high-to-low growth rate transition.(**B**) Curves of growth rates vs. shifted time for 100 cells randomly selected from those 80% cells.
DOI: https://doi.org/10.7554/eLife.41093.034

### A2.3 A One-Dimensional Model for Nutrient Penetration

The vertical ascending velocity and the pattern formation of the colony depend largely on how deep the nutrient from agar can penetrate into the interior of colony. Near the agar surface, cells grow fast with abundant nutrient supply. Away from the agar surface, the growth of cells in the colony is much limited due to the lack of nutrient. In this section, we construct a simplified 1D model of nutrient diffusion and reaction, and show that the nutrient concentration decays quadratically in the region where it is above the Monod constant $K_S$ but decays exponentially in the region where it is below $K_S$. We introduce the nutrient penetration level $H_S$ to be the height (i.e., the $z$ coordinate) at which the nutrient concentration takes the value $K_S$, and analyze how $H_S$ affects the colony growth.

Since the colony is rather thin, the variations of the nutrient concentration in the $x$ and $y$ directions are rather small compared with that in the $z$ direction. Therefore, we can assume that the steady-state concentration $C = C(z)$ depends only on the $z$ variable, and approximate the full Laplacian $\Delta C$ in the colony region by the 1D Laplacian $\partial_{zz}C$ in the $z$ variable. Consequently, we consider the following 1D model for the nutrient concentration $C = C(z)$ :

$$D_+ C'' = \frac{\rho_0 \lambda_S}{Y} \frac{C}{C + K_S} \quad \text{for } z > 0,$$

$$C(0) = C_0 \quad \text{and} \quad C(\infty) = 0,$$

where a prime denotes the spatial derivative and $C_0 > 0$ is a controlling parameter that describes the amount of nutrient concentration from the agar substrate. Note that we have replaced the local cell density $\rho$ by the constant density $\rho_0$; cf. the description of nutrient update in Appendix A1.2. Setting

$$\tilde{C} = \frac{C}{K_S} \quad \text{and} \quad \tilde{z} = \frac{z}{\beta} \quad \text{with} \quad \beta = \left( \frac{D_+ K_S Y}{\rho_0 \lambda_S} \right)^{1/2},$$

we obtain the non-dimensionalized form of the model

$$\tilde{C}'' = \frac{\tilde{C}}{\tilde{C} + 1} \quad \text{for } \tilde{z} > 0, \tag{A2.3.1}$$

$$\tilde{C}(0) = \tilde{C}_0 \quad \text{and} \quad \tilde{C}(\infty) = 0, \tag{A2.3.2}$$

where $\tilde{C}_0 = C_0/K_S$.

We observe that there is a unique solution $\tilde{C} = \tilde{C}(\tilde{z})$ to this boundary-value problem that is nonnegative for all $\tilde{z} \geq 0$. This solution is the unique minimizer of the convex functional

$$I[u] = \int_0^\infty \left[ \frac{1}{2} u'^2 + u - \ln(1+u) \right] d\tilde{z}$$

of all nonnegative functions $u$ such that both $u$ and $u'$ are square-integrable on $(0, \infty)$, and $u(0) = \tilde{C}(0)$ and $u(\infty) = 0$. Note that the solution $\tilde{C} = \tilde{C}(\tilde{z})$ is a monotonically decreasing function of $\tilde{z} > 0$. For otherwise, $\tilde{C}$ would reach a local maximum at some $\tilde{z}_m > 0$ with $\tilde{C}''(\tilde{z}_m) < 0$ but $\tilde{C}(\tilde{z}_m) \geq 0$. This is impossible by **Equation (A2.3.1)**. We also observe that $\tilde{C}'(\infty) = 0$, for otherwise $\tilde{C}'$ would be negative and stay strictly away from 0 as $\tilde{C}'' \geq 0$, leading eventually to $\tilde{C}(\infty) = -\infty$ which would contradict the fact that $\tilde{C}(\infty) = 0$.

Now, multiplying both sides of **Equation (A2.3.1)** by $\tilde{C}'$, we get

$$\frac{1}{2} \frac{d}{d\tilde{z}} (\tilde{C}')^2 = \frac{d}{d\tilde{z}} \left[ \tilde{C} - \ln(\tilde{C} + 1) \right].$$

Integrating both sides of this equation from 0 to $\tilde{z} > 0$, we obtain

$$\frac{1}{2} (\tilde{C}'(\tilde{z}))^2 - \frac{1}{2} (\tilde{C}'(0))^2 = \left[ \tilde{C}(\tilde{z}) - \ln(\tilde{C}(\tilde{z}) + 1) \right] - \left[ \tilde{C}_0 - \ln(\tilde{C}_0 + 1) \right].$$

Sending $\tilde{z} \to \infty$, we get

$$-\frac{1}{2} (\tilde{C}'(0))^2 = -\left[ \tilde{C}_0 - \ln(\tilde{C}_0 + 1) \right].$$

The combination of the above two equations leads to

$$\frac{1}{2} (\tilde{C}'(\tilde{z}))^2 = \tilde{C}(\tilde{z}) - \ln(\tilde{C}(\tilde{z}) + 1) \quad \forall \tilde{z} > 0. \tag{A2.3.3}$$

We now study how fast $\tilde{C} = \tilde{C}(\tilde{z})$ decays. Assume $\tilde{C}(z_1) \leq 1$ for some $\tilde{z}_1 \geq 0$. Then $\tilde{C}(z) \leq 1$ for all $\tilde{z} \geq \tilde{z}_1$. Noting that

$$x - \ln(x + 1) \geq x^2/3 \quad \text{if } 0 \leq x \leq 1,$$

and that $\tilde{C}' \leq 0$, we have by **Equation (A2.3.3)** that $-\tilde{C}' \geq \sqrt{2/3}\tilde{C}$ for all $\tilde{z} \geq \tilde{z}_1$. This leads to the exponential decay

$$\tilde{C}(\tilde{z}) \leq \tilde{C}(z_1) e^{-\sqrt{2/3}(\tilde{z} - \tilde{z}_1)} \quad \forall \tilde{z} \geq \tilde{z}_1. \tag{A2.3.4}$$

If $\tilde{C}_0 = \tilde{C}(0) \leq 1$, then we can have $\tilde{z}_1 = 0$ in **Equation (A2.3.4)** and the concentration $\tilde{C} = \tilde{C}(\tilde{z})$ decays exponentially.

Suppose now $\tilde{C}_0 > 1$. Since $\tilde{C} = \tilde{C}(\tilde{z})$ decreases monotonically and $\tilde{C}(\infty) = 0$, there exists a unique $\tilde{z}_0 = z_0/\beta > 0$ such that $\tilde{C}(\tilde{z}_0) = 1$ (i.e., $C(z_0/\beta) = K_S$) and $\tilde{C}(\tilde{z}) \leq 1$ for all $\tilde{z} \geq \tilde{z}_0$. Thus the inequality (**Equation (A2.3.4)**) holds with $\tilde{z}_1 = \tilde{z}_0$ and $\tilde{C}(\tilde{z}_1) = \tilde{C}(\tilde{z}_0) = 1$. We show now that the concentration $\tilde{C} = \tilde{C}(\tilde{z})$ decays quadratically in $[0, \tilde{z}_0]$. Precisely, we shall prove that

$$\left[ \max \left( \sqrt{\tilde{C}_0} - \frac{1}{\sqrt{2}} \tilde{z}, 0 \right) \right]^2 \leq \tilde{C}(\tilde{z}) \leq \left( \sqrt{\tilde{C}_0} - \sqrt{\frac{\ln(e/2)}{2}} \tilde{z} \right)^2 \quad \forall \tilde{z} \in [0, \tilde{z}_0]. \tag{A2.3.5}$$

By **Equation (A2.3.3)** and the fact that $\tilde{C}' \leq 0$, we have $-\tilde{C}' \leq \sqrt{2\tilde{C}}$ in $[0, \tilde{z}_0]$. This leads to $(\sqrt{\tilde{C}})' \geq -1/\sqrt{2}$. Integrating both sides of this inequality from 0 to $\tilde{z}$, we obtain the first inequality in **Equation (A2.3.5)**. Note that

$$(\ln 2)x - \ln(x+1) \geq 0 \qquad \forall x \geq 1.$$

Thus, by **Equation (A2.3.3)** and the fact that $\tilde{C}' \leq 0$, we obtain that $-\tilde{C}' \geq \sqrt{2\ln(e/2)\tilde{C}}$, and further that $(\sqrt{\tilde{C}})' \leq -\sqrt{(\ln(e/2))/2}$ in $[0, \tilde{z}_0]$. An integration of both sides of this inequality from 0 to $\tilde{z}$ then leads to the second inequality in **Equation (A2.3.5)**.

In **Figure 5—figure supplement 1** in the main text, we see that the nutrient concentration $C_{\text{ctr}}$ along the $z$-axis is described well by a quadratic function for $C_{\text{ctr}} \geq K_S$ (i.e., $\tilde{C} > 1$) and that the nutrient concentration decays exponentially for $C_{\text{ctr}} \leq K_S$ (i.e., $\tilde{C} \leq 1$).

The position $\tilde{z}_0 > 0$, defined by $\tilde{C}(\tilde{z}_0) = 1$, is the (rescaled) vertical level across which the nutrient concentration transitions from the quadratic decay described in **Equation (A2.3.5)** to the exponential decay described by **Equation (A2.3.4)** with $\tilde{z}_1 = \tilde{z}_0$. The nutrient penetration level we have defined is exactly $H_S = \beta\tilde{z}_0$. Setting $\tilde{z} = \tilde{z}_0$ in **Equation (A2.3.5)**, we see that $\tilde{z}_0 = O(\sqrt{\tilde{C}_0})$ if $\tilde{C}_0 \gg 1$. Therefore,

$$H_S = \beta\tilde{z}_0 = O\left(\sqrt{\frac{D_+ C_0 Y}{\rho_0 \lambda_S}}\right) = O\left(\frac{1}{\sqrt{\lambda_S}}\right).$$

Since the nutrient decays exponentially above $H_S$, only those cells below the height $H_S$ grow with the maximum growth rate $\lambda_S$. Therefore, the speed of vertical expansion $V_H$ is given by $V_H \propto H_S \lambda_S \propto \sqrt{\lambda_S}$; see the discussion in the main text: Vertical rise—quantitative analysis, in Simulation Results and Analysis.

