## [Decision Letter]

Thank you for submitting your article "Spatiotemporal establishment of growing bacterial colonies" for consideration by *eLife*. Your article has been reviewed by three peer reviewers, including Richard Neher as the Reviewing Editor and Reviewer #1, and the evaluation has been overseen by Naama Barkai as the Senior Editor.

The reviewers have discussed the reviews with one another and the Reviewing Editor has drafted this decision to help you prepare a revised submission.

Summary:

All reviewers agreed that your analysis of bacterial colony growth is an important step towards a quantitative understanding of this important aspect of microbiology. The manuscript clearly shows which mechanisms and forces are important in different locations and stages. Friction causes cells in the center of the colony to verticalize leading to vertical growth of the colony. This growth is restricted to a thin layer close to agar due to nutrient depletion. Radial growth occurs in a concentric ring at the periphery of the colony where cells are oriented horizontally. Your work nicely integrates observations, simulations, and analysis. However, there are a number of points that we would like to see addressed.

Essential revisions:

1) Verticalization of cells in the center of the colony is plausible and compelling in the simulation and the model, but strong evidence for this process in experiments is lacking. More direct evidence for cell orientation is particularly important as others (Su et al., 2012) have observed horizontal layers of cells. These authors used a constrained 2d geometry, but it is conceivable that surface tension has a similar effect. Enos-Berlage et al., 2000 provide some evidence in this direction. On a related semantic note, "buckling" is often more narrowly understood as the failure of a continuous material under compression – "verticalization" might be a better term.

2) A recent paper (Beroz et al., 2018) studies a very similar problem with interesting parallels and differences. The results you report should be compared to those by Beroz et al.

3) You stress the importance of incorporating both static and dynamic friction. To support this conclusion, we would like to see more explicitly that other/simpler models of friction lead to qualitatively different conclusions at odds with the experimental observations. An extended discussion of the implications of different forms of friction for colony or biofilm growth would be welcome.

4) The differences between dense homogeneous colonies and biofilms should be articulated more clearly. Many biofilms contain widely separated bacteria connected by polymeric substances. What concrete lessons can be learned from colony growth about biofilm growth?

5) The source code should be made available. Ideally on a repository like github.

---

## [Author Response]

Essential revisions:1) Verticalization of cells in the center of the colony is plausible and compelling in the simulation and the model, but strong evidence for this process in experiments is lacking. More direct evidence for cell orientation is particularly important as others (Su et al., 2012) have observed horizontal layers of cells. These authors used a constrained 2d geometry, but it is conceivable that surface tension has a similar effect. Enos-Berlage et al., 2000 provide some evidence in this direction. On a related semantic note, "buckling" is often more narrowly understood as the failure of a continuous material under compression – "verticalization" might be a better term.

At a first glance, the formation of horizontally oriented cell layers observed by Su et al., 2012, is in apparent contradiction to vertically oriented cells in the interior of a colony predicted by our model. However, Su et al.’s images were taken for very thin colonies (no more than a few layers in thickness). For such thin colonies, our model actually predicts cells to lie mostly horizontally; see Author response image 1 and B. According to our model, appreciable verticalization of cells occurs only after the colony thickness reaches certain height of several layers of cells; see the inset of Author response image 1.

**Author response image 1. respfig1:** Lack of verticalization for thin colonies. (**A**) Reproduction of Figure 2A from (Su et al., 2012). (**B**) Bottom view of a simulated colony at t=7h. (**C**) The fraction of vertically oriented cells (defined by the angle with the z-axis less than 45 degrees) vs. height from the simulation.

We have added Figure 4—figure supplement 2, which reproduces Author response image 1, and also the following sentence in “Vertical rise – a pictorial view” in the main text: (See also our response to the next comment.)

“However, the internal verticalization took some time to develop (Figure 4—figure supplement 2); appreciable fraction of cells (50%) picked up vertical orientation only when the radius reached 250μm.”

We thank the reviewer for pointing out the very interesting study by Enos-Berlage et al. which we were not aware of previously. These authors showed very clear confocal images of cells standing up. However, it is very difficult for us to obtain images of similar qualities for our colonies. Before we go into our results, let us explain the experimental issues involved since some of the reviewers are theorists.

1) They put a slide on top of the colony, which enabled them to use a (probably) water objective to get better resolution. If we do that to our colony, our colony collapses. Why the difference? Their colonies have EPS, which tends to make cells stick together more. (They say it has good structural integrity, because of EPS).

2) Because our colonies produce no EPS, our colonies are much more densely packed and appear opaque. It is therefore more difficult to distinguish between different cells.

3) We were only able to take high resolution image of our colonies from the bottom, through a layer of agar. To do so, we had to use long working distance objectives, which compromises z-resolution. Also, we had to significantly reduce the agar thickness, with possible physiological consequences.

Despite the above difficulties, we attempted to directly visualize cellular orientations in the interior of our colonies. As seen in Author response image 2, cells near the periphery of the colony clearly have elongated shape while cells in the interior have round shape. This is consistent with the predicted vertical orientation in the interior. However, as mentioned above, in order to take confocal images at single-cell resolution, we had to use very thin agar plates (~0.3mm thickness). This significantly perturbs the colony growth characteristics, e.g., the radial and vertical growth, from the bulk of the results described in this study (which was done with 7mm thick agar). Many parameters can be affected for such different experimental settings, agar dryness, surface roughness, nutrient depletion, etc. Consequently, we do not feel comfortable to include these images in the manuscript, even though they support our prediction. Instead, we prefer to leave the result of verticalization in compact colony as a prediction, to be validated experimentally in details in a better designed future study.

**Author response image 2. respfig2:** Experimental results on the confocal images of a bacterial colony at different positions, from the center to the edge. The bacterial strain and the growth medium are the same as described in “Experimental Methods” in the main text, except that the thickness of agar dish is ~0.3mm. The picture was taken at roughly 24 hours after the initial inoculation.

Finally, we thank the reviewer for suggesting the term “verticalization”. We now refer to it for the microscopic cell flow in the colony. However, we keep the term “buckling” to refer to the mechanical event itself, and still use “buckling width” to refer to the single layered, concentric ring region in the colony periphery.

2) A recent paper (Beroz et al., 2018) studies a very similar problem with interesting parallels and differences. The results you report should be compared to those by Beroz et al.

We believe the driving force for verticalization in our model is the same as that in the model of Beroz et al. It is a buckling instability driven by in-plane mechanical compression fueled by cell growth. The barrier for verticalization is however different. In Beroz et al. it is a cell-surface adhesion. In our case, it is the surface tension that leads to an effective force driving the colony into the agar. The two different mechanisms lead to quantitative differences in the process of verticalization. While Beroz et al. found cells to stand up even within a mono-layer, our model shows substantial verticalization only when the colony grows to a sufficient thickness, which is consistent with experimental observations by us and by Su et al.; see response to Essential revision 1 and Author response image 1.

With that said, however, we note that while our model applies directly to the compact colony cluster studied in our experiments, the experimental system that Beroz et al. modeled was actually a loosely packed colony. As far as we understand, cell-cell adhesion not modeled in Beroz et al. played a big role as described in Yan et al., 2016. It seems to us that additional assumptions are needed to justify the application of Beroz et al.’s model to their experimental systems.

We further note that, while the main focus of Beroz’s paper is on verticalization, the main focus of our study is to characterize and describe the temporal dynamical development of simple bacterial colonies in 3D. In particular, we focus on how radial expansion and vertical growth depend on various environmental parameters. We regard our key finding to be the elucidation of the origin of the linear radial expansion, which was long thought to be due to nutrient limitation. Instead, we show here that it is mechanical in origin, resulting from an interplay of colony surface tension and cell-agar friction.

We have added the following sentences in the section Discussion in the main text to describe the similarities between our work and that by Beroz et al. on the cell verticalization:

“Cell verticalization has been observed experimentally for *Vibrio parahaemolyticus* (Enos-Berlage and McCarter, 2000) and for Vibrio Cholerae (Beroz et al., 2018; Yan et al., 2016). […] Due to the different energy barriers against verticalization, the length scales of verticalization between our model and that of (Beroz et al., 2018) are very different: The colonies in Beroz et al. spread very slowly radially (~3μm/h), and verticalization occurs at a colony radius of 5-10μm. Colonies in our model spread much faster (~14μm/h), and substantial verticalization occurs at a radius of ~250μm; see Figure 4—figure supplement 2.”

3) You stress the importance of incorporating both static and dynamic friction. To support this conclusion, we would like to see more explicitly that other/simpler models of friction lead to qualitatively different conclusions at odds with the experimental observations. An extended discussion of the implications of different forms of friction for colony or biofilm growth would be welcome.

Introducing the dynamic friction was motivated by the strong dependence of the radial expansion speed on the growth rate observed (Figure 1H of the main text, reproduced as Author response image 3). As can be seen from our simulation results (filled circles in Figure 8F of the revised manuscript, reproduced as Author response image 3), the model with dynamic friction captures the experimental observations well. If we adopt a model with the static friction alone, we would instead obtain results shown in Author response image 3 (filled circles), which is the same as blue triangles in the new Figure 8F in the main text, with a much weaker growth-rate dependence.

**Author response image 3. respfig3:** The dynamic and static friction models distinguished by the growth-rate dependence of radial expansion speed. (**A**) From main text Figure 1F. Experimental results on the radial speed VR and the vertical speed VH with respect to various batch culture growth rates. (**B**) From Figure 8F in the main text (new version). Simulation results of radial speed VR of the colony, where both static and dynamic frictions are included in simulations. (**C**) New simulation results of radial speed VR of the colony, where only the static friction is included.

The reason that dynamic friction is called for here is simple. A proportional relation between the radial expansion speed VR and growth-rate λs is obtained if the buckling width is the same in different growth conditions. (It is strictly a proportional relation for cells with a fixed division length, as shown by the open circles in Figure 8F in the revised version, because the relation becomes superlinear when the growth-rate dependence of the cell division length itself is taken into account, as shown by the filled circles.) The latter is produced by the dynamic friction, since the frictional force depends only on the normal force, rather than on the relative speed between cell and agar. In contrast with the static friction, the faster cells grow, the faster is the relative speed between cell and agar, hence stronger the frictional force. This leads to reduced dependence of VR on the growth-rate (compare Author response image 3 and Author response image 3).

We have added the blue triangles in Figure 8F, same as Author response image 3. In addition, we have added the following sentence in the section “Radial expansion – quantitative analysis” in “Simulation Results and Analysis”:

“In contract, static friction leads to a much weaker dependence of VR on λS (blue triangles in Figure 8F).”

We have also added the following sentence in Conclusion (in the paragraph where we discuss the fiction forces) in the main text:

“Indeed, in a model with static friction alone, a much weaker growth-rate dependence of radial expansion speed was obtained (Figure 8F blue triangles).”

4) The differences between dense homogeneous colonies and biofilms should be articulated more clearly. Many biofilms contain widely separated bacteria connected by polymeric substances. What concrete lessons can be learned from colony growth about biofilm growth?

Due to complex EPS, biofilm is notoriously difficult to model. In this work, we focus on bacterial colonies without EPS, in order to confront predictions based on model with key known ingredients (growth rate, nutrient diffusion, elastic cell-cell interactions, and friction forces) with quantitative experiments. Key results we learned from this study all shed light on the more complex biofilm dynamics:

1) One key finding is that radial growth of our colonies is not limited by nutrient as commonly believed, but by the interplay of surface tension and cell-agar friction. Given that biofilms have typically much lower bacterial densities, nutrient limitation will be even less of a problem. Also, the EPS secreted by the bacteria could modify both the surface tension and cell-agar friction to control the radial expansion speed.

2) Nutrient supply is limiting for the vertical growth of our colonies. This becomes less of a problem for the loosely packed biofilms. Moreover, biofilms are said to form channels in their interior, which will further alleviate the supply of nutrient, thereby allowing for faster vertical expansion.

3) Verticalization of cells in the interior, which is important for vertical growth but occurs at rather large colony sizes according to our model (Author response image 1), also occurs in biofilms but at much smaller colony sizes as described by Yan et al., 2016, and Beroz et al., 2018. While the precise nature of the forces driving verticalization may be different in the two cases, the underlying origin appear to be similar – mechanical instability due to in-plane compression resulting from colony expansion and cell-agar friction.

In light of the above, we can see that the additional ingredients in biofilms provide the colonies with the means to expand even faster both horizontally and vertically.

We have added the following in the section Discussion (second to the last paragraph) of the main text:

“While our work is exclusively on bacterial colonies without EPS, key results we learned from this study shed light on the more complex dynamics of heterogeneous biofilms. […] In light of these comparisons, we see that the additional ingredients provided by biofilms enable the colonies to expand faster both horizontally and vertically.”

5) The source code should be made available. Ideally on a repository like github.

Our computer codes are developed using C++ with OpenMP parallelization. We have placed the major and basic parts of our codes in the repository GitHub. The address is: https://github.com/huiprobable/CellsMD3D